Manuscript prepared for The Cryosphere
with version 2015/04/24 7.83 Copernicus papers of the LATEX class copernicus.cls.
Date: 23 June 2016

# An ice sheet wide framework for englacial attenuation from ice penetrating radar data

T. M. Jordan[1], J. L. Bamber[1], C. N. Williams[1], J. D. Paden[2], M. J. Siegert[3],
P. Huybrechts[4], O. Gagliardini[5], and F. Gillet-Chaulet[6]

[1]Bristol Glaciology Centre, School of Geographical Sciences, University of Bristol, Bristol, UK.
[2]Center for Remote Sensing of Ice Sheets, University of Kansas, Lawrence, USA.
[3]Grantham Institute and Earth Science and Engineering, Imperial College, University of London, London, UK.
[4]Earth System Science and Departement Geografie, Vrije Universiteit Brussel, Brussels, Belgium.
[5]Le Laboratoire de Glaciologie et Geophysique de l'Environnement, University Grenoble Alpes, Grenoble, France.
[6]Le Laboratoire de Glaciologie et Geophysique de l'Environnement, Centre National de la Recherche Scientifique, Grenoble, France.

*Correspondence to:* T. M. Jordan (tom.jordan@bris.ac.uk) or J. L. Bamber (j.bamber@bris.ac.uk)

**Abstract.** Radar-inference of the bulk properties of glacier beds, most notably identifying basal melting, is, in general, derived from the basal reflection coefficient. On the scale of an ice sheet, unambiguous determination of basal reflection is primarily limited by uncertainty in the englacial attenuation of the radio wave, which is an Arrhenius function of temperature. Existing bed-returned power algorithms for deriving attenuation assume that the attenuation rate is regionally constant which is not feasible at an ice sheet wide scale. Here we introduce a new semi-empirical framework for deriving englacial attenuation, and, to demonstrate its efficacy, we apply it to the Greenland Ice Sheet. A central feature is the use of a prior Arrhenius temperature model to estimate the spatial variation in englacial attenuation as a first guess input for the radar algorithm. We demonstrate regions of solution convergence for two input temperature fields, and for independently analysed field campaigns. The coverage achieved is a trade-off with uncertainty and we propose that the algorithm can be 'tuned' for discrimination of basal melt (attenuation loss uncertainty $\sim 5$ dB). This is supported by our physically realistic ($\sim 20$ dB) range for the basal reflection coefficient. Finally, we show that the attenuation solution can be used to predict the temperature bias of thermomechanical ice sheet models, and is in agreement with known model temperature biases at the Dye 3 ice core.

## 1 Introduction

Ice Penetrating Radar (IPR) data provide valuable insights into several physical properties of glaciers and their beds including: ice thickness (e.g. Bailey et al. (1964); Evans and Robin (1966)), bed roughness (e.g. Berry (1973); Siegert et al. (2005); Rippin (2013)), basal material properties (e.g. Oswald and Gogineni (2008); Jacobel et al. (2009); Fujita et al. (2012); Schroeder et al. (2016)), internal

layer structure (e.g. Fujita et al. (1999); Bentley et al. (1998); Peters et al. (2005); Matsuoka et al. (2010a); Macgregor et al. (2015a)), basal melting or freezing (e.g. Fahnestock et al. (2001); Catania et al. (2010); Bell et al. (2011)), and englacial temperature (Macgregor et al., 2015b). In recent years, there has been a substantial increase in radar track density in Greenland and parts of Antarctica, which has lead to the development of new ice sheet wide data products for bed elevation and ice thickness (Fretwell et al., 2013; Bamber et al., 2013; Morlighem et al., 2014). These data products provide essential boundary conditions for numerical models of ice sheets (e.g. Gillet-Chaulet et al. (2012); Cornford et al. (2015)), and enable investigation of a diversity of topics related to ice sheet dynamics. By contrast, despite many notable regional studies (e.g. Oswald and Gogineni (2008); Jacobel et al. (2009); Fujita et al. (2012); Schroeder et al. (2016)), ice sheet wide data products for bulk basal material properties, such as quantifying regions of basal melt do not exist. As contemporary models of ice sheet dynamics have been demonstrated to be highly sensitive to basal traction (Price et al., 2011; Nowicki et al., 2013; Ritz et al., 2015), the poorly constrained basal interface poses a problem for their predictive accuracy. Additionally, ice sheet wide evaluation of englacial temperature from IPR data over the full ice column has yet to be realised, with recent advances focusing primarily on the isothermal regime (Macgregor et al., 2015b).

Bulk material properties of glacier beds can, in principle, be identified from their basal (radar) reflection coefficient (Oswald and Robin, 1973; Bogorodsky et al., 1983a; Peters et al., 2005; Oswald and Gogineni, 2008). The basal reflection coefficient is predicted to vary over a $\sim 20$ dB range for different subglacial materials, with water having a $\sim 10$ dB higher value than the most reflective frozen bedrock (Bogorodsky et al., 1983a). Relative basal reflection values can be fairly well constrained in the interior of ice sheets where the magnitude and spatial variation in the attenuation rate is expected to be low (Oswald and Gogineni, 2008, 2012). However, toward the margins of ice sheets unambiguous radar-inference of basal melt from bed reflections is limited primarily by uncertainty in the spatial variation of englacial attenuation (Matsuoka, 2011; MacGregor et al., 2012). Arrhenius models, where the attenuation rate is an exponential function of inverse temperature (Corr et al., 1993; Wolff et al., 1997; MacGregor et al., 2007; Macgregor et al., 2015b), predict that the depth-averaged attenuation rate varies by a decibel range of $\sim 5$-40 dB km$^{-1}$ over the Antarctic Ice Sheet (Matsuoka et al., 2012a). These models are, however, strongly limited by both inherent uncertainty in model parameters ($\sim 20$-25% fractional error) (MacGregor et al., 2007, 2012; Macgregor et al., 2015b), including a potential systematic underestimation of attenuation at the frequency of the IPR system (Macgregor et al., 2015b). Additionally Arrhenius models are highly sensitive to the input temperature field, which itself is poorly constrained. Despite this evidence for spatial variation in attenuation radar-algorithms, which use the relationship between bed-returned power and ice thickness to identify an attenuation trend, make the assumption that the attenuation rate is locally constant (e.g. Gades et al. (2000); Winebrenner et al. (2003); Jacobel et al. (2009); Fujita et al. (2012)). Due to this constancy assumption these radar algorithms are suspected to yield erroneous

values (Matsuoka, 2011; Schroeder et al., 2016). Moreover, these radar algorithms and are not tuned for automated application over the scale of an ice sheet.

In this study we introduce a new ice sheet wide framework for the radar-inference of attenuation and apply it to IPR data from the Greenland Ice Sheet (GrIS). A central feature of our approach is to firstly estimate the spatial variation in the attenuation rate using an Arrhenius model, which enables us to modify the empirical bed-returned power method. Specifically, the estimate is used to: (i) constrain a moving window for the algorithm sample region, enabling a formally regional

method to be applied on a ice sheet wide scale; (ii) to standardise the power for local variation in attenuation within each sample region when deriving attenuation using bed-returned power. We demonstrate regions of algorithm solution convergence for two different input temperature fields and for independently analysed IPR data. The coverage provided by the algorithm is a trade-off with solution accuracy, and we suggest that the algorithm can be 'tuned' for basal melt discrimination

in restricted regions, primarily in the southern and eastern GrIS. This is supported by the decibel range for the basal reflection coefficients ($\sim 20$ dB for converged regions). Additionally, we show that the attenuation rate solution can be used to infer bias in the depth-averaged temperature field of thermomechanical ice sheet models.

## 2    Data and methods

### 2.1    Ice penetrating radar data


The airborne IPR data used in this study were collected by the Center for Remote Sensing of Ice Sheets (CReSIS) within the Operation IceBridge project. Four field seasons from 2011-2014 (months March-May) have been analysed in this proof of concept study. These field seasons are the most spatially comprehensive to date, with coverage throughout all the major drainage basins of the GrIS

and relatively dense across-track spacing toward the ice margins (Fig. 1). The radar instrument, the Multi-Channel Coherent Radar Depth Sounder (MCoRDS), has been installed on a variety of platforms and has a programmable frequency range. However, for the data used in this study, it is always operated on the NASA P-3B Orion aircraft and uses a frequency range from 180 MHz to 210 MHz, which, after accounting for pulse shaping and windowing, corresponds to a depth-range resolution

in ice of 4.3 m (Rodriguez-Morales et al., 2014; Paden, 2015). The data processing steps to produce the multi-looked Synthetic Aperture Radar (SAR) images used in this work, are described in Gogineni et al. (2014). The along-track resolution after SAR processing and multilooking depends on the season and is either $\sim 30$ m or $\sim 60$ m with a sample spacing of $\sim 15$ m or $\sim 30$ m respectively. The radar's dynamic range is controlled using a waveform playlist which allows low and high gain

channels to be multiplexed in time. The digitally recorded gain for each channel allows radiometric calibration and, in principle, enables power measurements from different flight tracks and field

seasons to be combined. This is in contrast to pre 2003 CReSIS Greenland datasets, which used a manual gain control that was not recorded in the data stream.

## 2.2 Overview of algorithm

A flow diagram for the separate components of the radar algorithm is shown in Fig. 2. The along-track processing of the IPR data (Sect. 2.3) is an adaptation of the method developed by Oswald and Gogineni (2008, 2012), and is particularly suited to evaluation of bulk material properties via the reflection coefficient. The Arrhenius model estimation of the attenuation rate, (Sect. 2.4), uses the framework developed by MacGregor et al. (2007); Macgregor et al. (2015b) and assumes tempera-

ture fields from the GISM (Greenland Ice Sheet Model) (Huybrechts, 1996; Shapiro and Ritzwoller, 2004; Goelzer et al., 2013), and SICOPOLIS (SImulation COde for POLythermal Ice Sheets) (Greve, 1997) thermomechanical models. The Arrhenius model is used to firstly constrain the sample region for the algorithm (Sect. 2.5), and then to correct for local attenuation variation within each region when inferring the attenuation rate. Sections 2.5 and 2.6 represent the central original method contri-

butions in this study. They both address how the regional bed-returned power method for attenuation evaluation (which assumes local constancy) can be modified for spatial variation. Algorithm quality control is then implemented, by testing for regions where the attenuation solution is marked by strong correlation between bed-returned power and ice thickness, (Sect. 2.7). Finally, maps are produced for the radar-inferred attenuation rate, the two-way attenuation loss, and the basal reflection

coefficient, (Sect. 2.8).

## 2.3 Waveform processing

The processing of the IPR data, based upon the method developed by Oswald and Gogineni (2008, 2012), uses an along-track (phase-incoherent) average of the basal waveform and a depth aggregated/integrated definition of the bed-returned power. The advantage of using this definition, com-

pared with the conventional peak power definition, is that the variance due to variable bed roughness (e.g. Berry (1973); Peters et al. (2005)) is reduced. This reduction in variance is thought to occur because, based on conservation of energy principles, the aggregated definition of bed-returned power for a diffuse surface is more directly related to the predicted (specular) reflection coefficients than equivalent peak power values (Oswald and Gogineni, 2008). In our study we make two important

modifications to this method, which are described here, along with an overview of the key processing steps. The first modification corresponds to defining a variable window size for the along-track averaging of the basal waveform (which enables us to optimise the effective data resolution in thin ice), and the second corresponds to the implementation of an automated waveform quality control procedure.

Using the waveform processing method of Oswald and Gogineni (2008, 2012), the along-track
waveform averaging window is set using the first return radius

$$r = \sqrt{p\left(s + \frac{h}{\sqrt{\epsilon_{ice}}}\right)}, \tag{1}$$

where $p$=4.99 m is the (pre-windowed) radar pulse half-width in air (Rodriguez-Morales et al., 2014),
$s$ is the height of the radar sounder above the ice surface, $h$ is the ice thickness and, $\epsilon_{ice} = 3.15$ is
the real part of the relative dielectric permittivity for ice. For a flat surface, $r$, corresponds to the
radius of the circular region illuminated by the radar pulse such that it extends the initial echo re-
turn by <50% (Oswald and Gogineni, 2008). Additionally, if adjacent waveforms within this region
are stacked about their initial returns and arithmetically averaged, they represent a phase-incoherent
average where the effects of power fluctuations due to interference are smoothed (Oswald and Gogi-
neni, 2008; Peters et al., 2005). Oswald and Gogineni (2008, 2012) considered the northern interior
of the GrIS where $h \sim 3000$ m, and subsequently $r$ and the along-track averaging interval were ap-
proximated as being constant. Since our study considers IPR data from both the ice margins and the
interior, we use Eq. (1) to define a variable size along-track averaging window. For the typical flying
height of $s$=480 m, $r$ ranges from $\sim 55$ m in thin ice ($h$=200 m) to $\sim 105$ m in thick ice ($h$=3000 m),
though can be higher during plane maneuvers. The number of waveforms in each averaging window
is then obtained by dividing $2r$ by the along-track resolution.

The incoherently averaged basal waveforms range from sharp pulse-like returns associated with
specular reflection, to broader peaks associated with diffuse reflection (refer to Oswald and Gogineni
(2008) for a full discussion). An example of an incoherently averaged waveform is shown in Fig. 3a,
in units of linear power, $P$, versus depth-range index $D_i$. The plot shows the upper and lower limits
of the power depth integral, $D_{lower}$ and $D_{upper}$. These limits are symmetric about the peak power
value, with $(D_{upper} - D_{lower}) = 2r$ (in units of the depth-range index); a range motivated by the
observed fading intervals described in (Oswald and Gogineni, 2008). Subsequently, as is the case for
the along-track averaging bin, the power integral limits vary over the extent of the ice sheet and are
of greater range in thicker ice. The aggregated (integrated) power is then defined by

$$P_{agg} = \sum_{Di=D_{lower}}^{Di=D_{upper}} P(D_i). \tag{2}$$

Waveform quality control, was implemented by testing if the waveform decays to a specified fraction
of the peak power value within the integral limits $D_{lower}$ and $D_{upper}$. This effectively provides a
test that the SAR beamwidth is large enough to include all of the scattered energy, which was argued
to be the general case by Oswald and Gogineni (2008). Decay fractions of 1%, 2% and 5% were
considered, and 2% was established to give the best coverage, whilst excluding obvious waveform
anomalies. The waveform in Fig. 3a is an example that satisfies the quality control measure, whereas
the waveform shown in Fig. 3b does not. The relative decibel power for each waveform is then

defined by

$$[P] = 10 \log_{10} P_{agg}, \tag{3}$$

where the decibel notation $[X] = 10 \log_{10} X$ is used. Finally, the relative power is corrected for the effects of geometrical spreading using

$$[P^C] = [P] - [G], \tag{4}$$

where

$$[G] = 20 \log_{10} \frac{g \lambda_0}{8 \pi \left( s + \frac{h}{\sqrt{\epsilon_{ice}}} \right)}, \tag{5}$$

(Bogorodsky et al., 1983b) with $g = 4$ the antenna gain (corresponding to 11.8 dBi) (Paden, 2015), and $\lambda_0$=1.54 m the central wavelength of the radar pulse (Rodriguez-Morales et al., 2014).

### 2.4 Arrhenius temperature model for attenuation

It is well established that the dielectric conductivity and radar attenuation rate in glacier ice is described by an Arrhenius relationship where there is exponential dependence upon inverse temperature and a linear dependence upon the concentration of soluble ionic impurities (Corr et al., 1993; MacGregor et al., 2007; Stillman et al., 2013; Macgregor et al., 2015b). The Arrhenius modelling framework introduced by Macgregor et al. (2015b) for the GrIS, which we adopt here, includes three soluble ionic impurities: hydrogen/acidity ($H^+$), chlorine/sea salt ($Cl^-$), and ammonium ($NH_4^+$). Our Arrhenius model assumes uniform, depth-averaged, molar concentrations: $c_{H^+}$=0.8 $\mu$M, $c_{Cl^-}$=1.0 $\mu$M and $c_{NH_4^+}$=0.4 $\mu$M ($M$ = mol $L^{-1}$), which are derived from GRIP core data (Macgregor et al., 2015b). A decomposition of the temperature dependence for the attenuation rate for pure ice and the different ionic species is shown in Fig. 4. Use of layer stratigraphy for the concentration of the ionic species (rather than depth-averaged values) is discussed in detail in MacGregor et al. (2012); Macgregor et al. (2015b). The equations and parameters for the model calculation of the attenuation rate, $\hat{B}$ (dB $km^{-1}$), the depth-averaged attenuation rate, $< \hat{B} >$ (dB $km^{-1}$), and the two-way attenuation loss, $[\hat{L}]$ (dB), are outlined in Appendix A. Throughout this manuscript we use $\hat{X}$ notation to distinguish Arrhenius model estimates from the radar derived values, and $< X >$ to indicate depth-averages. For brevity we often refer to the depth-averaged attenuation rate as the attenuation rate.

The Arrhenius relationship is empirical and the dielectric properties of impure glacier ice, (pure ice conductivity, molar conductivities of soluble ionic impurities, and activation energies), need to be measured with respect to a reference temperature and frequency. Two Arrhenius models for the dielectric conductivity and the attenuation rate were applied to the GrIS by Macgregor et al. (2015b): the W97 model introduced by Wolff et al. (1997), and the M07 model introduced by MacGregor et al. (2007). For equivalent temperature and chemistry the W97 model produces conductivity/attenuation

rate values at $\sim 65$ % of the M07 model (Macgregor et al., 2015b). In Appendix A we describe these models in more detail, along with an empirical correction to the W97 model (from herein referred to as W97C), which accounts for a proposed frequency dependence of the dielectric conductivity between the radar frequency (195 MHz) and the reference frequency of the Arrhenius model (300 kHz). In Appendix A we propose a test, based upon the thickness correlation for the estimated values of the basal reflection coefficient, for how well tuned each model is for estimating the conductivity/attenuation at the radar frequency. From this test we conclude that the M07 model provides a suitable estimate for our algorithm, and unless stated we use it an all further attenuation estimates.

The temperature fields for GISM and SICOPOLIS were used to estimate the spatial variation in the depth-averaged attenuation rate for the GrIS and were interpolated at 1 km grid resolution. Both the GISM and SICOPOLIS models provide temperature profiles as a function of relative depth, and these were vertically scaled using the 1 km Greenland Bedmap 2013 ice thickness data product (Bamber et al., 2013). For the SICOPOLIS temperature field it is necessary to convert the (homologous) temperature values from degrees below pressure melting point to units of K (or °C) using a depth correction factor of -0.87 K km$^{-1}$ (Price et al., 2015). For both temperature fields, the attenuation rate is predicted to vary extensively over the GrIS, with minimum values in the interior ($\sim 7$ dB km$^{-1}$) and maximum values for the south western margins of $> 35$ dB km$^{-1}$ (shown for GISM in Fig. 5a and SICOPOLIS in Fig. 5b). Toward the ice sheet margins GISM generally has lower temperature and therefore lower attenuation rate than SICOPOLIS (Fig. 5c). The GISM vertical temperature profiles are in better overall agreement with the temperature profiles at the deep ice core sites shown in Fig. 1b (refer to Macgregor et al. (2015b) for summary plots of the core temperature profiles).

## 2.5 Constraining the algorithm sample region

Radar-inference of the depth-averaged attenuation rate, using the relationship between bed-returned power and ice thickness, requires sampling IPR data from a local region of the ice sheet (Gades et al., 2000; MacGregor et al., 2007; Jacobel et al., 2009; Fujita et al., 2012; Matsuoka et al., 2012b). An implicit assumption of the method is that the depth-averaged attenuation rate is constant across the sample region (Layberry and Bamber, 2001; Matsuoka et al., 2010a). However, as was shown in Sect. 2.4, the depth-averaged attenuation rate is predicted to have pronounced spatial variation, and therefore an ice sheet wide radar attenuation algorithm must take this into account. In our development of an automated framework we use the spatial distribution of $< \hat{B} >$ (the prior Arrhenius model estimate) to constrain the size and shape of the sample region as a function of position (a 'moving target window') by estimating regions where the attenuation rate is constant subject to a specified tolerance. The most general, but computationally expensive, approach to defining the sample region would be to define an irregular contiguous region about each window centre where the

attenuation rate is less than a tolerance criteria (such as an absolute difference). Here, motivated by computational efficiency, we have developed a 'segmentation approximation' for defining the anisotropic sample region window. This approach uses local differences in the estimated $<\hat{B}>$ field along 8 grid directions, and is similar in its representation of anisotropy to numerical gradient operators defined on an orthogonal grid. Below we describe the key conceptual steps to our method with the further details in Appendix B.

Fig.6a illustrates an example of the anisotropy that can occur in the spatial distribution of $<\hat{B}>$ for a 120 km$^2$ region of the GrIS. The target window is divided into eight segments, (notated by $S_n$ with n=1,2,...,8), in a plane-polar coordinate system about a central point $(x_0, y_0)$, (Fig. 6b), with the ultimate goal to produce a variable radial length of the target window by interpolating with respect to angle. The size of each segment is defined by its central radius vector, $R_n$, for angles $\theta_n = \frac{(n-1)\pi}{8}$, with $R_1 = R_5$, $R_2 = R_6$, $R_3 = R_7$, $R_4 = R_8$. The estimate $<\hat{B}>$ is then approximated in the plane-polar coordinate system by defining the attenuation rate in each segment to have the same radial dependence as along the direction of the central radius vector: $<\hat{B}(r)>=<\hat{B}(r_n, \theta_n)>$ with $r = \sqrt{(x - x_0)^2 + (y - y_0)^2}$ (Fig. 6c). The Euclidean distance of $<\hat{B}>$ from $(x_0, y_0)$ is then used to define a tolerance metric, shown for $\sqrt{(<\hat{B}(x,y)> - <\hat{B}(x_0,y_0)>)^2}$ in Fig. 6d and $\sqrt{(<\hat{B}(r_n,\theta_n)> - <\hat{B}(x_0,y_0)>)^2}$ (the segment approximation) in Fig. 6e respectively. Finally, the boundaries of the target window are defined by linear interpolation along a circular arc (Fig. 6f). Note that the target window boundaries are largest in the direction approximately parallel to the contours of constant $<\hat{B}>$ in Fig. 6a.

A primary consideration for the moving target window is that the dimensions, $R_n$, are smoothly varying in space. If the converse were true then there would be a sharp discontinuity in the IPR data that is sampled. It was established that, rather than use of a simple maximum Euclidean distance criteria to define $R_n$, a Root Mean Square (RMS) integral measure produces greater spatial continuity (described fully in Appendix B). The spatial distribution of the target window radius vectors $R_1$, $R_2$, $R_3$, $R_4$ using GISM temperature field are shown in Fig. 7. All four plots have the general trend that the target window radi are larger in the interior of the ice sheet corresponding to where the $<\hat{B}>$ field is more slowly varying. The dependence of $R_1$, $R_2$, $R_3$, $R_4$ upon the anisotropy of the $<\hat{B}>$ field in Fig. 5 is also evident, with larger radi approximately parallel to contours of constant $<\hat{B}>$ and smaller radi approximately perpendicular. This target windowing approach is sensitive to the input temperature field and repeat plots for the SICOPOLIS temperature field are shown in the supplementary material (Fig. S2). Finally, we note that the segmentation approach is sensitive to the horizontal gradient/local difference in $<\hat{B}>$ (and therefore the horizontal gradient of depth-averaged temperature). Hence systematic biases in the model temperature fields are less important.

## 2.6 Radar-inference of attenuation rate

The method of using the relationship between ice thickness and bed-returned power to infer the radar-attenuation rate and basal reflection coefficient has been employed many times to local regions of ice sheets (Gades et al., 2000; Winebrenner et al., 2003; MacGregor et al., 2007; Jacobel et al., 2009; Fujita et al., 2012). An explanation for how this method works, begins with the radar power equation

$$[P^C] = [R] - [L], \tag{6}$$

where $[R]$ is the basal reflection coefficient, $[L]$ is the total (two way) power loss (Matsuoka et al., 2010a). This version of the radar power equation neglects instrumental factors, which here we assume to be a constant for each field campaign. In our study $[P^C]$ is the aggregated geometrically corrected power, as defined by Eqs. (2)-(4), whereas in the majority of other studies $[P^C]$ is the geometrically corrected peak-power of the basal echo. Equation (6) does not include additional loss due to internal scattering, which can occur when the glacial ice has crevasses and is not well stratified as is often the case for fast flowing regions near the ice sheet margin (Matsuoka et al., 2010a; MacGregor et al., 2007). Expressing the total loss in terms of the depth averaged attenuation rate as $[L] = 2 < B > h$, and then considering the variation in Eq. (6) with respect to ice thickness gives

$$\frac{\delta[P^C]}{\delta h} = \frac{\delta[R]}{\delta h} - 2 < B >, \tag{7}$$

(Matsuoka et al., 2010a). If $\frac{\delta[R]}{\delta h} << \frac{\delta[P^C]}{\delta h}$, (refer to Sect. 2.7 for the algorithm quality control measures that test for this), then

$$< B > \approx -\frac{1}{2} \frac{\delta[P^C]}{\delta h}. \tag{8}$$

Subsequently, radar-inference of the attenuation rate is achieved via linear regression of Eq. (8), the total loss can be calculated from $[L] = 2 < B > h$, and the basal reflection coefficients can be calculated from Eq. (6).

As discussed here and in Sect. 2.5, in applying this linear regression approach, it is assumed that the regression gradient (i.e. the depth-averaged attenuation rate) is constant throughout the sample region which can lead to erroneous slope estimates (Matsuoka, 2011). In practice, however, the sample region must necessarily include ice with a range of thicknesses, and therefore a range of temperatures and attenuation rates. In our modification to the basic method, the Arrhenius model is used to 'standardise' bed-returned power for local attenuation variation, using the central point of each target window as a reference point. This is achieved via the power correction

$$[P^C]_i \rightarrow [P^C]_i + 2 \left( < \hat{B}(x_i, y_i) > - < \hat{B}(x_0, y_0) > \right) h_i, \tag{9}$$

where $(x_i, y_i)$ corresponds to the position of the $i$th data point within the target window and $(x_0, y_0)$ corresponds to the central point. This power correction represents an estimate of the difference in attenuation loss between an ice column of the actual measurement (loss estimate $2 < \hat{B}(x_i, y_i) > h_i$),

and a hypothetical ice column with the same thickness as the measurement but with the attenuation rate of the central point (loss estimate $2 < \hat{B}(x_0, y_0) > h_i$).

An example of a $[P^C]$ versus $h$ regression plot pre- and post- power correction, Eq. (9), is shown in Fig. 8. In this example, ice columns that are thinner than the central point have $(< \hat{B}(x_i, y_i) > - < \hat{B}(x_0, y_0) >) > 0$ and the power values are increased by Eq. (9), whereas ice columns that are thicker than the central point have $(< \hat{B}(x_i, y_i) > - < \hat{B}(x_0, y_0) >) < 0$ and the power values are decreased. Subsequently, the power correction acts to enhance the linear correlation between power and ice thickness, (as demonstrated by the increase in the $r^2$ value in Fig. 8), and enables the underlying attenuation trend to be better discriminated. It follows that, for this situation described, failing to take into account the spatial variation in attenuation rate in the linear regression procedure results in a systematic underestimation of the attenuation rate. The difference in radar-inferred attenuation rate pre- and post-power correction depends upon the distribution of IPR flight track coverage within the sample region and the size of the sample region, and is typically $\sim$ 1-4 dB km$^{-1}$. Equation (9) represents our central modification to the bed-returned power method for deriving attenuation. We anticipate that, if a temperature model is available, this correction for local attenuation variation could be applied in future regional studies (even if the windowing methods describe in Sect. 2.5 are not).

When applying the linear regression approach described in this section, IPR data from each field season were considered separately. To ensure that there was sufficiently dense data within each sample region a minimum threshold of 20 measurements was enforced, where each 'measurement' corresponds to a separate along-track averaged waveform as described in Sect. 2.3. Additionally, target window centres that were more than 50 km from the nearest IPR data point were excluded.

### 2.7 Quality control

The accuracy of the radar-inferred attenuation rate solution from Eq. (8) depends upon: (i) a strong correlation between bed-returned power and ice thickness, $\frac{\delta[P^C]}{\delta h}$, (ii) a weak correlation between reflectivity and ice thickness, $\frac{\delta[R]}{\delta h}$, relative to $\frac{\delta[P^C]}{\delta h}$. To make a prior estimate of the correlation for $\frac{\delta[R]}{\delta h}$ we use the prior Arrhenius model estimate of the basal reflection coefficient governed by

$$[\hat{R}] = [\hat{L}] + [P^C] = 2 < \hat{B} > h + [P^C], \tag{10}$$

and consider the correlation and linear regression model for $\frac{\delta[\hat{R}]}{\delta h}$. The joint quality control threshold:

$$r^2_{[P^C]} \quad > \quad \alpha, \tag{11}$$

$$r^2_{ratio} \quad = \quad \frac{r^2_{[P^C]}}{r^2_{[P^C]} + r^2_{[\hat{R}]}} > \beta, \tag{12}$$

is then enforced where $r^2_{[P^C]}$ and $r^2_{[\hat{R}]}$ are $r^2$ correlation coefficients for the $\frac{\delta[P^C]}{\delta h}$ and $\frac{\delta[\hat{R}]}{\delta h}$ linear regression models, and $0 \geq \alpha \geq 1$, $0 \geq \beta \geq 1$ are threshold parameters. The first thresholding criteria,

Eq. (11) tests for strong absolute correlation in $\frac{\delta[P^C]}{\delta h}$, and the second thresholding criteria, Eq. (12), tests for strong relative correlation in $\frac{\delta[P^C]}{\delta h}$ with respect to $\frac{\delta[\hat{R}]}{\delta h}$. The name for the $r^2_{ratio}$ parameter represents that it is 'correlation ratio'. Both quality measures are designed with attenuation rate/loss accuracy in mind, (rather than directly constraining the distribution of relative reflection). Unlike the use of the Arrhenius model attenuation estimate in Sect. 2.5 and Sect. 2.6, which uses the local

difference in the $< \hat{B} >$ field, in Eq. (10) the absolute value of $< \hat{B} >$ is used. A justification for the use of the absolute value here, is that it is used only as a quality control measure and does not directly enter the calculation of the radar-inferred attenuation rate.

In general, $r^2_{[\hat{R}]}$ can be high (or equivalently $r^2_{ratio}$ can be low) due to: (i) there being a true correlation in the basal reflection coefficient with thickness, (ii) there being a correlation due to

additional losses other than attenuation such as internal scattering, (iii) the Arrhenius model estimate of the attenuation rate being significantly different from the true attenuation rate. Whilst the first two reasons are both desirable for quality control filtering, the third reason is an erroneous effect. However, as the dual threshold filters out all three classes of sample region, this erroneous effect simply reduces the coverage of the algorithm.

## 2.8 Gridded maps

The attenuation rate solution from the radar algorithm, $< B >$, is at a 1 km grid resolution and arises as a consequence of the scan resolution of the moving target window described in Sect. 2.5. It is defined on the same polar-stereographic coordinate system as in Fig. 1 and the gridded thickness data from Bamber et al. (2013). Subsequently, a gridded data set for the two way loss can be calculated

using $[L] = 2 < B > h$. For grid cells that contain IPR data, the mean $[P^C]$ value is calculated, and using Eq. (6) an along-track map for the gridded relative reflection coefficient, $[R]$, is obtained. Due to the definition of relative power in Eqs. (3) and (4), the values of $[R]$ are also relative. As described in Sect. 2.3 the averaging procedure for the basal waveforms means that the effective resolution of the processed IPR data varies over the extent of the ice sheet. Consequently, the number of data points

that are arithmetically averaged in each grid cell varies according to both this resolution variation and the orientation of the flight tracks relative to the coordinate system. For a single flight line, (i.e. no intersecting flight tracks), the number of points in a grid cell typically ranges from $\sim 4$ in thick ice to $\sim 16$ in thin ice. Initially, maps for the four field seasons were independently processed, which enables cross over analysis for the uncertainty estimates. Joint maps were then produced by

averaging values where there were grid cells with coverage overlap.

## 3 Results and discussion

With a view toward identifying regions of the GrIS where the radar attenuation algorithm can be applied, we firstly consider ice sheet wide properties for the linear regression correlation parameters

(Sect. 3.1). We then demonstrate that, on the scale of a major drainage basin, basin 4 in Fig. 1b (SE

Greenland), the attenuation solution converges for the two input temperature fields (Sect. 3.2). We go

on to show that the converged attenuation solution produces a physically realistic range and spatial

distribution for the basal reflection coefficient (Sect. 3.3). The relationship between algorithm cov-

erage and uncertainty is then outlined (Sect. 3.4). Finally, we consider how the attenuation solution

can be used to predict temperature bias in thermomechanical ice sheet models (Sect. 3.5).

**3.1   Ice sheet wide properties**

Ice sheet wide maps for the linear regression correlation parameters are shown in Fig. 9a-c using the

GISM temperature field as an input. As discussed in Sect. 2.6 and Sect. 2.7, the radar algorithm re-

quires: (i) a strong correlation between bed-returned power and ice thickness (high $r^2_{[PC]}$), (ii) a weak

correlation between basal reflection and ice thickness (low $r^2_{[\hat{R}]}$ and high $r^2_{ratio}$). In general, $r^2_{[PC]}$

has stronger correlation values in southern Greenland (typically $\sim 0.7$-$0.9$). These regions of higher

correlation correspond to where there is higher variation in ice thickness due to basal topography,

and are correlated with regions of higher topographic roughness (Rippin, 2013). Correspondingly,

in the northern interior of the ice sheet where the topographic roughness is lower there are weaker

correlation values for $r^2_{[PC]}$ (typically $\sim 0.2$-$0.3$). The correlation values for $r^2_{[PC]}$ in the northern

interior can also, in part, be explained by the lower absolute values for the depth-averaged attenua-

tion rate as predicted in Fig. 5. The correlation values for $r^2_{[\hat{R}]}$ are generally much lower than $r^2_{[PC]}$

and more localised. As discussed in Sect. 2.7, regions where $r^2_{[\hat{R}]}$ is high can arise due to both true

target-window scale variation in the basal reflector or due to a significant bias in the Arrhenius model

estimation, $[\hat{R}]$. The values for $r^2_{ratio}$, are largely correlated with $r^2_{[PC]}$.

Examples of algorithm coverage for three different sets of $(\alpha,\beta)$ quality control thresholds, Eqs.

(11) and (12), are shown in Fig. 9d. These are chosen such that each successively higher quality

threshold region is contained within the lower threshold region. In Sect. 3.4 we discuss how the

coverage regions relate to uncertainty in the radar-inferred attenuation rate and two-way attenua-

tion loss, and the central problem of the radar-inference of the basal material properties. For the

discussion here, it is simply important to note that algorithm coverage is fairly continuous for a

significant proportion of the southern ice sheet, (corresponding to large regions of major drainage

basins 4,5,6,7), and toward the margins of the other drainage basins. The spatial distribution of the

radar-inferred attenuation rate, $< B(T_{\mathrm{GISM}}) >$, is shown in Fig. 9e and the radar-inferred attenuation

loss , $[L(T_{\mathrm{GISM}})]$, is shown in Fig. 9f, both for threshold $(\alpha, \beta) = (0.6, 0.8)$. Note that the ice sheet

wide properties for $< B(T_{\mathrm{GISM}}) >$ are similar to the Arrhenius model predictions (Fig. 5a) with

higher values ($\sim 15$-$30$ dB km$^{-1}$) toward the ice margins and lower values ($\sim 7$-$10$ dB km$^{-1}$) in the

interior.

The ice sheet wide properties of the algorithm are preserved using the SICOPOLIS temperature

field as an input (refer to Supplemental Material for a repeat plot of Fig. 9). Notably, the ice sheet

wide distribution for $r^2_{[PC]}$ is similar, and for equivalent choices of threshold parameters there is better coverage for the southern GrIS than for the northern interior.

## 3.2 Attenuation solution convergence

To demonstrate the convergence of the attenuation solution for different input temperature fields (convergence is defined here as a normally distributed difference distribution about zero), we com-

pare the solution differences for the (input) Arrhenius models, $< \hat{B}(T_{\text{GISM}}) > - < \hat{B}(T_{\text{SIC}}) >$ and $[\hat{L}(T_{\text{GISM}})] - [\hat{L}(T_{\text{SIC}})]$, with the corresponding (output) radar-inferred solution differences, $< B(T_{\text{GISM}}) > - < B(T_{\text{SIC}}) >$ and $[L(T_{\text{GISM}})] - [L(T_{\text{SIC}})]$. As $[L] = 2 < B > h$, it is necessary to consider the thickness dependence of the solution differences and the consequences for a thickness correlated bias in basal reflection values. We focus on the southeast GrIS, corresponding to target window cen-

tres that are located in drainage basin 4 Fig. 1a. This region is selected post ice sheet wide processing, and the IPR data from neighboring drainage basins are incorporated in the linear regression plots for the target windows that lie close to the basin boundaries. We consider an attenuation rate solution for fixed threshold parameters $(\alpha, \beta)$=(0.6,0.8). These are chosen to achieve a solution uncertainty deemed to approach the accuracy required to discriminate basal melt (discussed fully in Sect. 3.4).

The inset region we consider is shown in (Fig. 10a). The prior Arrhenius model solution difference for the attenuation rate, $< \hat{B}(T_{\text{GISM}}) > - < \hat{B}(T_{\text{SIC}}) >$, is strongly negatively biased (Fig. 10b). If the solution difference is aggregated over all grid cells that contain IPR data the mean and standard deviation, $\mu \pm \sigma$, is -2.42 $\pm$ 0.88 dB km$^{-1}$ (Fig. 10d). Note, that $\sigma$ does not represent an uncertainty for the Arrhenius modeled attenuation rate. It is a measure of the spread of the two different input

attenuation rate fields. On the scale of the drainage basin, this solution bias is approximately constant with ice thickness (Fig. 10e). By contrast, the radar algorithm solution difference, $< \hat{B}(T_{\text{GISM}}) > - < \hat{B}(T_{\text{SIC}}) >$, fluctuates locally between regions of both small positive and negative bias (Fig. 10c). The aggregated radar solution bias is approximately normally distributed about zero, $\mu \pm \sigma$=-0.18 $\pm$ 1.53 dB km$^{-1}$ (Fig. 10d), and approximately constant with ice thickness (Fig. 10e).

Corresponding difference distributions for the attenuation loss are shown in Fig. 10f and Fig. 10g. These represent a rescaling of the distributions in Fig. 10d and Fig. 10e, by the factor $2h$ and do not take thickness uncertainty into account. The Arrhenius model solution difference is weakly negatively correlated with thickness ($r^2$=0.09), and from Eq. (6) results in a thickness correlated bias for the basal reflection coefficient. As the attenuation loss solution bias can be > 10 dB for thick

ice ($h \sim 2000$ m or greater), this would potentially result in a different diagnosis of thawed and dry glacier beds using the different temperate fields in the Arrhenius model. Again, the radar-inferred solution difference is approximately normally distributed about zero ($\mu \pm \sigma$=-0.56 $\pm$ 5.19 dB). The radar-inferred difference is also uncorrelated with ice thickness ($r^2$=0.00) which is highly desirable for unambiguous radar-inference of basal material properties on an ice sheet wide scale.

If a similar analysis for the attenuation solution differences is applied to drainage basins 3,5,6 (southern and eastern Greenland) we observe algorithm solution convergence, (in the sense of a normally distributed difference centred on zero), and an associated reduction in the solution bias from the Arrhenius model input. In drainage basins 1,2,7,8 (northern and western Greenland) we do not observe analogous solution convergence for the radar-inferred values. We do, however, typically

see a reduction in the mean systematic bias for the attenuation rate/loss solution relative to the Arrhenius model input. In the supplementary material we provide additional plots and discuss the potential reasons for the algorithm non-convergence, which are thought to relate primarily to the more pronounced temperature sensitivity of the algorithm target windows in the northern GrIS.

### 3.3    Attenuation rate and basal reflection maps

For regions of the GrIS where the attenuation rate solution converges and there is algorithm coverage overlap for the different temperature field inputs, it is possible to define the mean radar-inferred attenuation rate solution

$$< \bar{B} >= \frac{1}{2} \left( < B(T_{\text{SIC}}) > + < B(T_{\text{GISM}}) > \right). \tag{13}$$

Note, that the explicit temperature dependence for the mean value is dropped as, for the regions of

convergence, it represents a solution that is (approximately) independent of the input temperature field. Within the drainage basins where the solution converges and where only one of $< B(T_{\text{SIC}}) >$ or $< B(T_{\text{GISM}}) >$ is above the coverage threshold, we use the single values to define the mean $< \bar{B} >$ field. A justification for this approach is that regions where only one temperature field has coverage are most likely an instance of where the other temperature field has erroneous estimates for $\frac{\delta[\hat{R}]}{\delta h}$ as

discussed in Sect. 2.7. Hence, for a given $(\alpha, \beta)$ threshold, the coverage region for $< \bar{B} >$ is slightly larger than for $< B(T_{\text{SIC}}) >$ and $< B(T_{\text{GISM}}) >$. A map for the converged attenuation rate solution using Eq. (13) is shown in Fig. 11 for coverage threshold $(\alpha, \beta)$=(0.60,0.80). This field is generally smoothly varying, as would be expected given its primary dependence upon temperature.

    Inset maps for the depth-averaged attenuation rate and basal reflection coefficient are compared

with balance velocity (Bamber et al., 2000) in Fig. 11b-d. Following the naming convention in Bjørk et al. (2015), this region is upstream from the Apuseeq outlet glacier. Balance velocities rather than velocity measurements are used due to incomplete observations in the region of interest (Joughin et al., 2010). The correspondence between the fast flowing region (approximately > 120 m a$^{-1}$) and the nearcontinuous regions of higher attenuation rate (approximately > 18 dB km$^{-1}$) and higher

basal reflection values (approximately > 8 dB) is evident. This supports the view that the fast flowing region corresponds to relatively warm ice, and is underlain by a predominately thawed bed which acts to enhance basal sliding.

    The probability distribution for the relative basal reflection coefficient, $[R]$, over the converged region is shown in Fig. 11e. The distribution is self-normalised by setting the mean value to equal

zero. The decibel range is $\sim 20$ dB which is consistent with the predicted decibel range for sub-glacial materials (Bogorodsky et al., 1983a), and our estimate of the loss uncertainty ($\sim 5$ dB), discussed in more detail in Sect. 3.4. Since our definition of the basal reflection coefficient is based upon the aggregated definition of the bed-returned power, Eqs. (2) and (3), the overall range will be less than using the conventional peak power definition.

## 3.4 Relationship between uncertainty and coverage

There are two metrics, both as a function of the quality threshold parameters $(\alpha, \beta)$, that we propose can be used to quantify the uncertainty of the radar algorithm. The first metric is the standard deviation of the attenuation solution differences for different input temperature fields as previously described in Sect. 3.2. This metric assesses solution variation due to the target windowing and the lo-
cal correction to the power within the target window described in Sect. 2.5 and Sect. 2.6 respectively. The second metric is to consider the standard deviation of the attenuation solution differences for independently analysed field seasons for a fixed input temperature field. This metric provides a test that the waveform-processing and system performance is consistent between different field seasons. Furthermore, it provides a test if different flight track distributions and densities in the same target
window, produce a similar radar-inferred attenuation rate.

Attenuation rate and loss solution difference distributions for three $(\alpha, \beta)$ coverage thresholds for the different temperature field inputs (the first uncertainty metric) are shown in Fig. 12a and Fig. 12b respectively, along with corresponding coverage regions in Fig. 12c. As in Sect. 3.2, these distributions are for grid cells that contain IPR data within drainage basin 4. It is clear that the
standard deviation of the difference distribution is related to how strict the coverage threshold is, with the strictest coverage threshold having the smallest standard deviation value (refer to plots for values). Subsequently, we suggest that the coverage of the algorithm is a trade-off with uncertainty. The systematic bias for the strictest coverage threshold, $(\alpha, \beta) = (0.80, 0.90)$, is thought to arise due to sampling an insufficiently small region of the ice sheet. The standard deviation values in Fig. 12
for drainage basin 4 are similar in the other drainage basins where there is solution convergence. For example, for $(\alpha, \beta) = (0.60, 0.80)$, $\sigma \sim 1.5$ dB km$^{-1}$ for the attenuation rate difference distribution.

A similar relationship between the choice of $(\alpha, \beta)$ threshold parameters and solution accuracy arises for independently analysed field campaign data and a full data table is supplied in the supplementary material. The attenuation solution difference distributions are close to being normally
distributed about zero, with small systematic biases ($\sim 0.1$-$0.7$ dB km$^{-1}$) for the attenuation rate. For the same choice of $(\alpha, \beta)$ threshold parameters, the attenuation rate solution standard deviations are of similar order to the equivalent temperature field difference distributions. For example, for $(\alpha, \beta) = (0.60, 0.80)$, $\sigma$ is in the range 0.98-1.71 dB km$^{-1}$ for the different field season pairs.

Since for both uncertainty metrics, the solution differences are a function of $(\alpha, \beta)$, we suggest
that the coverage region can be 'tuned' to a desired accuracy. For the problem of basal melt discrim-

ination, where the reflection coefficient difference between water and frozen bedrock is $\sim$ 10-15 dB (Bogorodsky et al., 1983b), we suggest that standard deviation values for the attenuation loss of $\sim$ 5 dB approaches the required accuracy. If this is rescaled by the ice thickness for a typical sample region (ice thickness $\sim$ 1500-2000 m) this results in a desired attenuation rate accuracy $\sim$ 1-1.5 dB km$^{-1}$. For both uncertainty metrics this corresponds to approximately $(\alpha, \beta) = (0.6, 0.8)$. This interpretation of uncertainty is consistent with the $\sim$ 20 dB decibel range for the basal reflection coefficients in Fig. 11. Throughout the algorithm development, we continually considered both uncertainty metrics. Of particular note, if the Arrhenius model is used to constrain the target window dimensions (Sect. 2.5), but not to make a power correction within each target window (Sect. 2.6), there are more pronounced systematic biases present for both uncertainty metrics.

The recent study by Macgregor et al. (2015b) also produced a GrIS wide map for the radar-inferred attenuation rate. This study used returned power from internal layers in the glacier ice to infer the attenuation rate (Matsuoka et al., 2010b), and the values are therefore only for some fraction of the ice column (roughly corresponding to the isothermal region of the vertical temperature profiles). The uncertainty was quantified using the attenuation rate solution standard deviation ($\sigma$=3.2 dB km$^{-1}$) at flight transect crossovers. A direct comparison between their uncertainty estimate and ours is not possible, as we use a different definition of cross-over point (i.e. all grid-cells that contain IPR data in a mutual coverage region), and we can tune the coverage of our algorithm for a desired solution accuracy. Additionally, whereas each value using the internal layer method is spatially independent, the moving target-windowing approach of our algorithm means each radar-inferred value is dependent upon neighboring estimates.

### 3.5  Evaluation of temperature bias of ice sheet models

The evaluation of the temperature bias of a thermomechanical ice sheet model using attenuation rates inferred from IPR data was recently considered for the first time by Macgregor et al. (2015b); in this case the ISSM model described by Seroussi et al. (2013). For the internal layer method used by Macgregor et al. (2015b) the attenuation rate inferred from the IPR data represents a truly independent test of temperature bias. For our method, which uses ice sheet model temperature fields as an input, this is not necessarily the case, and we only consider regions where the radar-inferred values tend to converge for different input temperature fields (the map in Fig. 11a). The inversion of the Arrhenius relations (solving for a depth-averaged temperature given a depth-averaged attenuation rate) is both a non-linear and non-unique problem. We leave this problem, which is potentially more complex for the full ice column than the depth section where internal layers are present (which is closer to being isothermal), for future work. Instead we estimate temperature bias using the Arrhenius model-radar algorithm solution differences for the depth-averaged attenuation rate: $< \hat{B}(T_{\mathrm{GISM}}) > - < \bar{B} >$ and $< \hat{B}(T_{\mathrm{SIC}}) > - < \bar{B} >$. These differences can only give a broad indication regarding the horizontal distribution of depth-averaged temperature bias, and will not hold exactly if ionic concentrations or

the shape of the vertical temperature profiles differ substantially over the region. In order to illustrate the sensitivity of our results, and the evaluation of model temperature fields in general, to the choice of conductivity model, we use the W97C model alongside the M07 model.

Arrhenius model-radar algorithm attenuation solution differences are shown for the M07 model (GISM Fig. 13a, SICOPOLIS Fig. 13b) and W97C model (GISM Fig. 13c, SICOPOLIS Fig.13d). The frequency correction parameter for W97C corresponds to $\sigma_{195\mathrm{MHz}}/\sigma_{300\mathrm{kHz}}$=1.7 (the ratio of the dielectric conductivity at the IPR system frequency relative to the reference frequency of the Arrhenius model), and is described in detail in Appendix A. Dye 3 is the only ice core within the coverage

region and the model and core temperature profiles are shown in Fig. 13e. For the M07 model $< \hat{B}(T_{\mathrm{GISM}}) >$-$< \bar{B} >$ is negative in the region of the Dye 3 core (suggestive of negative temperature bias), whereas $< \hat{B}(T_{\mathrm{SIC}}) >$-$< \bar{B} >$ is positive (suggestive of positive temperature bias) which is in agreement with the known model temperature biases Fig. 13e. Arrhenius model attenuation rate values at the core are $< \hat{B}(T_{\mathrm{GISM}}) >$=12.8 dB km$^{-1}$ and $< \hat{B}(T_{\mathrm{SIC}}) >$=16.7 dB km$^{-1}$ and the radar

inferred value is $< \bar{B} >$= 15.8 dB km$^{-1}$. The W97C model (which estimates attenuation rate values $\sim$ 10-15 % higher than the M07 model) is also consistent with this attenuation rate/temperature bias hierarchy, with $< \hat{B}(T_{\mathrm{SIC}}) >$= 18.7 dB km$^{-1}$ and $< \hat{B}(T_{\mathrm{GISM}}) >$= 14.3 dB km$^{-1}$. It is also possible to use the ice core temperature profile at Dye 3 in the Arrhenius model to predict depth-averaged attenuation rate values. This gives $< \hat{B}(T_{\mathrm{CORE}}) >$=13.9 dB km$^{-1}$ for the M07 model and

$< \hat{B}(T_{\mathrm{CORE}}) >$=15.8 dB km$^{-1}$ for the W97C model. These values are both consistent with the radar-inferred value subject to the original uncertainty estimate of the M07 model ($\sim$ 5 dB km$^{-1}$ when the temperature field is known (MacGregor et al., 2007)).

     A final caveat to our approach here is that it does not include layer stratigraphy in the Arrhenius model. The analysis in Macgregor et al. (2015b) predicts that, throughout the GrIS, radar-inferred

temperatures that incorporate layer stratigraphy are generally systematically lower (correspondingly depth-averaged attenuation rates are systematically higher). This deficit is predicted to be most pronounced in southern and western Greenland, due to the higher fraction of Holocene ice in these regions which has higher acidity than the depth-averaged values at GRIP (Macgregor et al., 2015a).

## 4   Conclusions

In this study, we considered the first application of a 'bed-returned power' radar algorithm for englacial attenuation over the extent of an ice sheet. In developing our automated, ice sheet wide, approach we made various refinements to previous regional versions of the algorithm (Gades et al., 2000; MacGregor et al., 2007; Jacobel et al., 2009; Fujita et al., 2012; Matsuoka et al., 2012b). These included using a waveform processing procedure that is specifically tuned for evaluation of

bulk material properties, incorporating a prior Arrhenius model estimate for the spatial variation in attenuation to constrain the sample area, standardising the power within each sample area, and

introducing an automated quality control approach based upon the underlying radar equation. We demonstrated regions of attenuation solution convergence for two different input temperature fields and for independently analysed field seasons. A feature of the algorithm is that the uncertainty, as measured by standard deviation of the attenuation solution difference distribution for different input temperature fields and separate field seasons, is tunable. Subsequently, we suggested that the algorithm could be used for the discrimination of bulk material properties over selected regions of ice sheets. Notably, assuming a total loss uncertainty of $\sim 5$ dB to be approximately sufficient for basal melt discrimination, we demonstrated that, on the scale of a major drainage basin, the attenuation solution produces a physically realistic ($\sim 20$ dB) range for the basal reflection coefficient.

The converged radar algorithm attenuation solution provides a means of assessing the bias of forward Arrhenius temperature models. Where temperature fields are poorly constrained, and where the algorithm has good coverage, we suggest that it is preferable to using a prior Arrhenius model calculation. With this in mind, the potential problems with using a forward Arrhenius model for attenuation were illustrated (Sect. 3.2). Notably, we demonstrated that even a small regional bias in attenuation rate (this could arise either due to temperature bias or due a systematic bias in the Arrhenius model parameters) leads to thickness-correlated errors in attenuation losses and therefore the basal reflection coefficients. These thickness-correlated errors persist regardless of whether the regional bias is with respect to the 'true' value or to another modelled value. We hypothesise that the algorithm convergence for different input temperature fields occurs because the local differences in the Arrhenius model attenuation rate field that are used as an algorithm input (i.e. $< \hat{B}(x,y) > - < \hat{B}(x_0,y_0) >$) are more robust than the absolute values. This is broadly equivalent to saying that the horizontal gradients in the depth-averaged temperature field of the ice sheet models are more robust than the absolute values of the depth-averaged temperature. Similarly, our use of local differences for the attenuation rate estimate is also robust to systematic biases in the Arrhenius model.

We have yet to consider an explicit classification of the subglacial materials and quantification of regions of basal melting. In future work, we aim to combine IPR data from preceding CReSIS field campaigns to produce a gridded data product for  basal reflection values and basal melt. It is anticipated that, as outlined by Oswald and Gogineni (2008, 2012); Schroeder et al. (2013), the specularity properties of the basal waveform, and how this relates to basal melt detection, could also be incorporated in this analysis. As the regions of algorithm coverage are sensitive to uncertainty, we suggest that these data products could have spatially varying uncertainty incorporated. Additionally, for the basal reflection and basal melt data sets, uncertainty in the measurements of $[P^C]$ will have to be incorporated in the uncertainty estimate for $[R]$. Establishing a procedure for the interpolation of these data sets where either: (i) the algorithm coverage is poor due to low attenuation solution accuracy, or (ii) the IPR data are sparse, will form part of this framework. Regions of lower solution accuracy, generally correspond to the interior of the ice sheet where spatial variation in the attenuation rate is much less pronounced (primarily the northern interior). Due to this lower spatial variability, (and

despite the caveats in the paragraph above), these regions could potentially have their basal reflection values derived by using a forward Arrhenius temperature model for the attenuation.

Finally, we envisage that the framework introduced in this paper could be used for radar-inference of radar-attenuation, basal reflection and basal melt for the Antarctic Ice Sheet. Given that for high solution accuracy the radar algorithm requires high topographic roughness and relatively warm ice we suggest that IPR data in rougher regions toward the margins should be analysed first (refer to Siegert et al. (2005) for an overview of topographic roughness in East Antarctica). Additionally, the prediction of the model temperature field bias using the attenuation rate solution could be extended to the Antarctic Ice Sheet.

## Appendix A: Additional information for Arrhenius model

### A1  Model equations

In ice, a low loss dielectric, the radar attenuation rate, $\hat{B}$ (dB km$^{-1}$) is linearly proportional to the high frequency limit of the electrical conductivity, $\sigma_\infty$ ($\mu$S m$^{-1}$), following the relationship

$$\hat{B} = \frac{10\log_{10}e}{1000\epsilon_0 c\sqrt{\epsilon_{ice}}}\sigma_\infty, \tag{A1}$$

where $c$ is the vacuum speed of the radio wave (Winebrenner et al., 2003; MacGregor et al., 2012). For $\epsilon_{ice} = 3.15$, as is assumed here, $\hat{B} = 0.921\sigma_\infty$. The Arrhenius relationship describes the temperature dependence of $\sigma_\infty$ for ice with ionic impurities present, and is given by

$$\begin{aligned}
\sigma_\infty = {} & \sigma_{pure}\exp\left\{\frac{E_{pure}}{k_B}\left(\frac{1}{T_r} - \frac{1}{T}\right)\right\} \\
& + \mu_{\text{H}^+}c_{\text{H}^+}\exp\left\{\frac{E_{\text{H}^+}}{k_B}\left(\frac{1}{T_r} - \frac{1}{T}\right)\right\} \\
& + \mu_{\text{Cl}^-}c_{\text{Cl}^-}\exp\left\{\frac{E_{\text{Cl}^-}}{k_B}\left(\frac{1}{T_r} - \frac{1}{T}\right)\right\} \\
& + \mu_{\text{NH}_4^+}c_{\text{NH}_4^+}\exp\left\{\frac{E_{\text{NH}_4^+}}{k_B}\left(\frac{1}{T_r} - \frac{1}{T}\right)\right\},
\end{aligned} \tag{A2}$$

where $T$ (K) is the temperature, $T_r$ is a reference temperature, $K_B = 1.38\times10^{-23}$ J K$^{-1}$ is the Boltzmann constant, and $c_{\text{H}^+}$, $c_{\text{Cl}^-}$ and $c_{\text{NH}_4^+}$ are the molar concentrations of the chemical constituents ($\mu$M) (MacGregor et al., 2007; Macgregor et al., 2015b). The model parameters are summarised in tabular form by Macgregor et al. (2015b) for both the M07 model and W97 model.

Following the assumptions in Sect. 2.4 for the GrIS temperature field, ionic concentrations, and ice thickness data set, it is possible to obtain the spatial dependence of the attenuation rate, $\hat{B}(x,y,z)$, where $(x,y)$ are planar coordinates and $z$ is the vertical coordinate. The two-way attenuation loss for a vertical column of ice, $[\hat{L}(x,y)]$ (dB), is then obtained via the depth integral

$$[\hat{L}] = 2\int_0^h \hat{B}(z)dz. \tag{A3}$$

Finally, the depth averaged (one-way) attenuation rate, $< \hat{B}(x,y) >$ (dB km$^{-1}$) is calculated from

$$< \hat{B} >= [\hat{L}]/2h. \tag{A4}$$

## A2 Frequency dependence and empirical correction

Both the W97 model and the M07 model assume that the dielectric conductivity/attenuation rate is
frequency independent between the medium frequency, MF; 0.3-3 MHz, (the range that the Arrhenius model parameters are measured) and the very high frequency, VHF; 30-300 MHz, (the range encompassing the frequency of IPR systems) (Macgregor et al., 2015b). The W97 model is derived using the dielectric profiling method at GRIP core and is referenced to 300 kHz (Wolff et al., 1997), whereas the M07 model is derived from a synthesis of prior measurements and is not referenced to a specific frequency (MacGregor et al., 2007). The empirical frequency correction to the W97 model between the MF and VHF, W97C, was motivated by an inferred systematic underestimation in the attenuation rate at the GrIS ice cores. This analysis was based upon using reflections from internal layers to derive attenuation rate values and then inverting the Arrhenius relations to estimate englacial temperature. The frequency corrected model represents a departure from the classical (frequency independent) Debye model for dielectric relaxations under an alternating electric field. The physical basis for the frequency dependence is related to the presence of a log-normal distribution for the dielectric relaxations (Stillman et al., 2013).

For the MCoRDS system that is considered in this study and by Macgregor et al. (2015b), the empirical frequency correction to $\sigma_\infty$ in Eq. (A2) is given by

$$\sigma_\infty \longrightarrow \left(\frac{\sigma_{195\text{MHz}}}{\sigma_{300\text{kHz}}}\right)\sigma_\infty, \tag{A5}$$

where $\sigma_{195\text{MHz}}/\sigma_{300\text{kHz}}$ is the ratio of the conductivity at the central frequency of the radar system to the W97 model frequency. A ratio $\sigma_{195\text{MHz}}/\sigma_{300\text{kHz}} = 2.6$ was inferred by Macgregor et al. (2015b), from minimising the difference between radar-inferred temperatures and borehole temperatures. This value was thought to potentially represent an overestimate due to unaccounted biases in the internal layer method (e.g. non-specularity of internal reflections, volume scattering). Additionally, Paden et al. (2005) observed a $8 \pm 1.2$ dB increase in signal loss from the bed at NGRIP between 100 and 500 MHz. If this is interpreted as being entirely to the frequency dependence of the conductivity then this implies $\sigma_{195\text{MHz}}/\sigma_{300\text{kHz}} = 1.7$ (Macgregor et al., 2015b).

## A3 Test for model bias and model selection

The W97C model with $\sigma_{195\text{MHz}}/\sigma_{300\text{kHz}} = 2.6$ calculates attenuation rate values at $\sim 170$ % of the M07 model, whereas the W97C model with $\sigma_{195\text{MHz}}/\sigma_{300\text{kHz}}=1.7$ calculates conductivity/attenuation rate values at $\sim 115$ % of the M07 model. To date, neither of these frequency-corrected models have been used to calculate full ice column losses or basal reflection coefficients for MCoRDS IPR

data. In order to inform our choice of conductivity model, we considered the decibel range of the estimated reflection coefficient, $[\hat{R}]$, as a function of ice thickness. Whilst it is not strictly necessary that this distribution is invariant with ice thickness (there may be an overall thickness dependence to the distribution of thawed/frozen beds), a thickness-invariant distribution over an extended region serves as an indirect test of the validity the conductivity models. We consider northern Greenland

(drainage basin 1 in Fig. 1) as a trial region since the attenuation rate/temperature is low compared to southern Greenland with less spatial variation (Fig. 5). Initially, the GISM temperature field is used as it is closer to the NEEM and Camp Century core profiles (see supplementary material).

    A prior estimate for the basal reflection coefficient, $[\hat{R}]$, as a function of ice thickness for four conductivity models is shown in Fig. 14: (a) W97 (uncorrected), (b) M07, (c) W97C ( $\sigma_{195\text{ MHz}}/\sigma_{300\text{ kHz}}$=1.7)

(the inferred value from Paden et al. (2005)), (d) W97C ($\sigma_{195\text{ MHz}}/\sigma_{300\text{ kHz}}$=2.6) (the inferred value from Macgregor et al. (2015a)). The W97 (uncorrected) model has negative correlation with ice thickness, (-6.03 dB km$^{-1}$, $r^2$=0.29), the M07 model is near invariant with ice thickness (-0.29 dB km$^{-1}$, $r^2$=0.0009), the W97C model with $\sigma_{195\text{ MHz}}/\sigma_{300\text{ kHz}}$=1.7 has a minor positive correlation (1.86 dB km$^{-1}$, $r^2$=0.03), and the W97C model with $\sigma_{195\text{ MHz}}/\sigma_{300\text{ kHz}}$=2.6 has a strong positive cor-

relation (12.02 dB km$^{-1}$, $r^2$=0.49). The negative correlation for W97 is consistent with the conclusion by Macgregor et al. (2015b) that the model is an underestimate of the conductivity at frequency of the radar system. The reasoning behind this is that, since $[\hat{L}] = 2 <\hat{B}> h$, a systematic underestimate in the attenuation rate results in an underestimation of the loss that increases with ice thickness, and from Eq. (10) a negative thickness gradient results for the basal reflection coefficient. The op-

posite is true for W97C with $\sigma_{195\text{ MHz}}/\sigma_{300\text{ kHz}}$=2.6, where the strong positive correlation indicates that the attenuation rate is significantly overestimated. Since both the M07 model and W97C with $\sigma_{19\,5\text{MHz}}/\sigma_{300\text{ kHz}}$=1.7 are close to being thickness invariant, we infer that the conductivity models are better tuned for estimating the attenuation rate at the radar frequency. Repeat analysis for other regions of the GrIS and using the SICOPOLIS temperature field confirm these general conclusions.


## Appendix B: Additional information for constraining the algorithm sample region

In this Appendix we describe the RMS integral measure that we use to define the sample region boundaries, as described conceptually in Sect. 2.5. The RMS measure, which is similar to the RMS integral measure for a continuous-time function, is defined for each segment by

$$\text{RMS}(R_n) = \sqrt{\frac{2}{R_n^2} \int_0^{R_n} (<\hat{B}(r_n,\theta_n)> - <\hat{B}(x_0,y_0)>)^2 r_n dr_n}. \tag{B1}$$

Specifying a value of $\text{RMS}(R_n)$, then enables radius vectors $R_n$ to be derived from evaluating the integral, Eq. (B1). It was further established that smoother windowing occurs if the constraints

$R_1 = R_5$, $R_2 = R_6$, $R_3 = R_7$, $R_4 = R_8$, are applied and the joint integral

$$\text{RMS}(R_n) = \frac{1}{2}\sqrt{\frac{2}{R_n^2}\int_0^{R_n}(<\hat{B}(r_n,\theta_n)> - <\hat{B}(x_0,y_0)>)^2 r_n dr_n}$$

$$+ \frac{1}{2}\sqrt{\frac{2}{R_n^2}\int_0^{R_n}(<\hat{B}(r_m,\theta_m)> - <\hat{B}(x_0,y_0)>)^2 r_m dr_m}, \tag{B2}$$

with index pairs $(n,m)$=(1,5), (2,6), (3,7) and (4,8) is used to solve for $R_n$.

Tuning the RMS tolerance, Eq. (B2), is discussed in the supplementary material. Briefly, the chosen value (RMS=1 dB km$^{-1}$) is a balance between being large enough to ensure that there is an adequate spread in ice thickness, whilst being sufficiently small to ensure that attenuation rate values are sufficiently close to the central point of the target window. It is shown in this study that in central Greenland, this condition is generally not satisfied because the gradient in ice thickness with distance is too small. The segmentation approximation and RMS tolerance measure is just one possible approach to constraining the sample region and incorporating anisotropy. For example, we could have considered an ovular or ellipsoidal shape region.

*Acknowledgements.* This study was supported by UK NERC grant NE/M000869/1. We would like to thank our reviewers J. Macgregor and M. Wolovick for their valuable comments which greatly improved this manuscript, and K. Matsuoka for handling our manuscript submission. We would also like to thank J. Macgregor for supplying ice core temperature profiles.

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

**Table 1.** List of principal symbols.

| Symbol | Units | Description | Equation(s) |
|---|---|---|---|
| $[P^C]$ | dB | Aggregated and geometrically corrected bed-returned power | (2)-(5) |
| $h$ | km | Thickness of ice column | |
| $\hat{B}$ | dB km$^{-1}$ | Arrhenius model estimate for attenuation rate | (A1), (A2) |
| $[\hat{L}]$ | dB | Arrhenius model estimate for two-way attenuation loss | (A3) |
| $<\hat{B}>$ | dB km$^{-1}$ | Arrhenius model estimate for depth-averaged attenuation rate | (A4) |
| $[\hat{R}]$ | dB | Arrhenius model estimate for basal power reflection coefficient | (10) |
| $R_n$ | km | Radius vectors for sample regions with $n$=1,2,3,4 | |
| RMS($R_n$) | dB km$^{-1}$ | Root mean square tolerance measure for sample regions | (B2) |
| $<B>$ | dB km$^{-1}$ | Radar-inferred value for depth-averaged attenuation rate | (8) |
| $[L]$ | dB | Radar-inferred value for two-way attenuation loss | |
| $[R]$ | dB | Radar-inferred value for basal power reflection coefficient | |
| $r^2_{[P^C]}$ | | $r^2$ correlation coefficient for $[P^C]$ versus $h$ | |
| $r^2_{[\hat{R}]}$ | | $r^2$ correlation coefficient for $[\hat{R}]$ versus $h$ | |
| $r^2_{ratio}$ | | Correlation ratio of $r^2_{[PC]}$ to $(r^2_{[PC]} + r^2_{[\hat{R}]})$ | (12) |
| $\alpha$ | | Quality control threshold for $r^2_{[PC]}$ | (11) |
| $\beta$ | | Quality control threshold for $r^2_{ratio}$ | (12) |

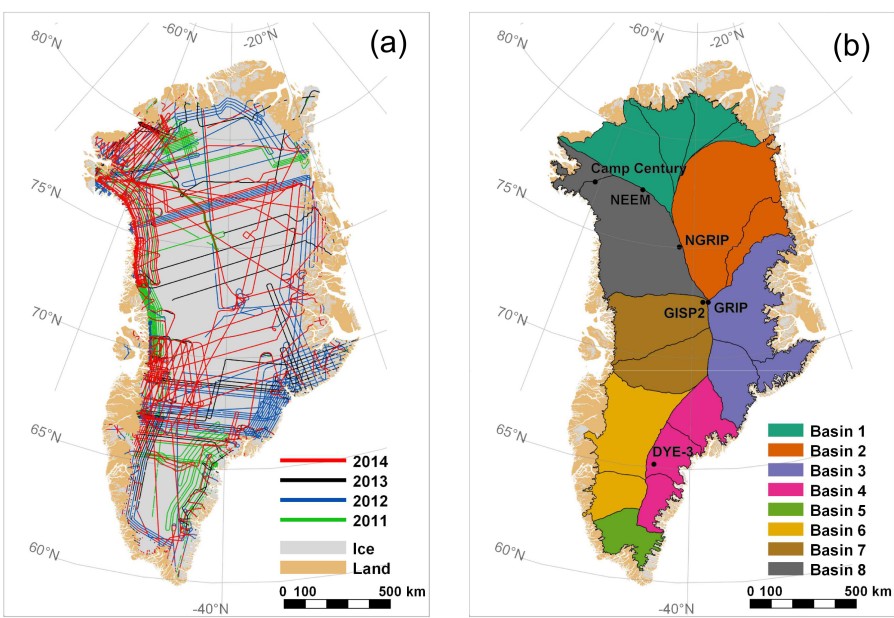

**Figure 1.** (**a**) Source map for CReSIS flight tracks. (**b**) Ice core locations and GrIS drainage basins (Zwally et al., 2012). The coordinate system, used throughout this study, is a polar-stereographic projection with reference latitude 71° N and longitude 39° W. The land-ice-sea mask is from Howat et al. (2014).

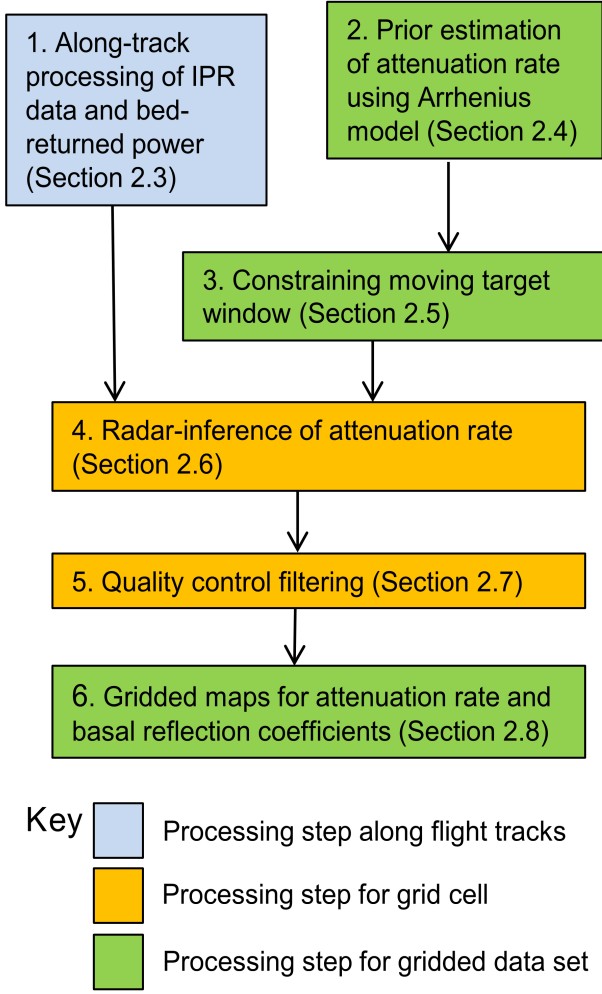

**Figure 2.** Flow diagram for the components of the radar algorithm.

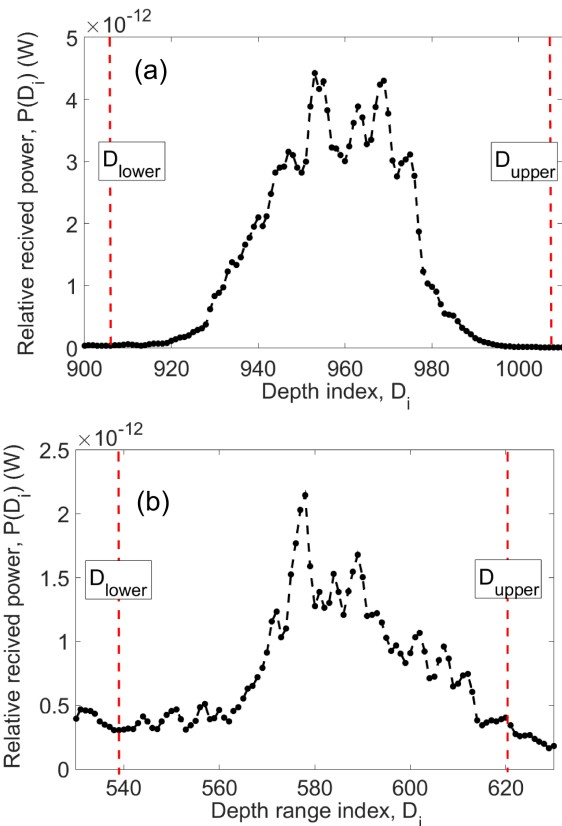

**Figure 3.** Waveform processing using the power depth-integral method, Eq. (2). **(a)** A waveform that satisfies the quality control criteria (decays to 2% of peak power within integral bounds). **(b)** A waveform that does not satisfy the quality control criteria.

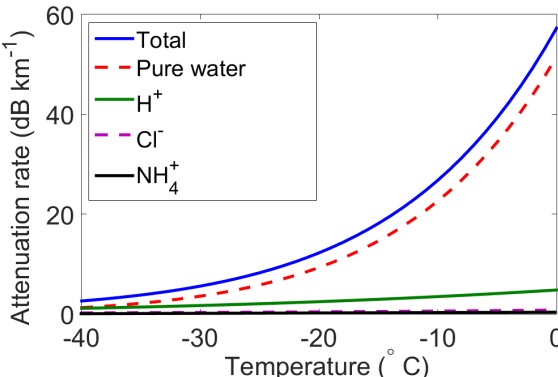

**Figure 4.** Temperature dependence of estimated attenuation rate, $\hat{B}$, assuming depth-averaged chemical concentrations at GRIP core and the Arrhenius model, M07, in MacGregor et al. (2007).

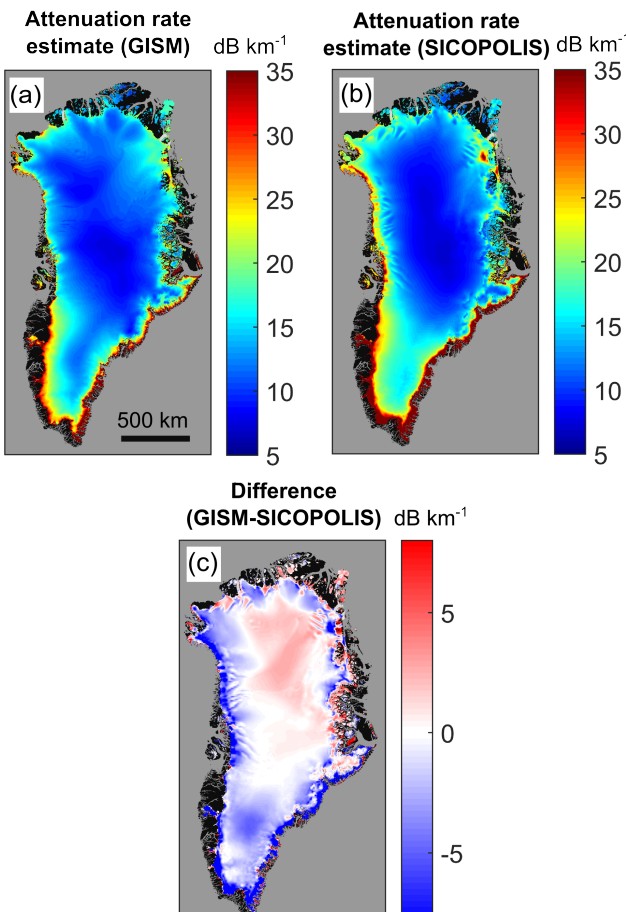

**Figure 5.** Estimated spatial dependence of depth-averaged attenuation rate for the GrIS using Arrhenius model. (a) GISM temperature field, $< \hat{B}(T_{\text{GISM}}) >$. (b) SICOPOLIS temperature field, $< \hat{B}(T_{\text{SIC}}) >$. (c) Attenuation rate difference plot for GISM-SICOPOLIS, $< \hat{B}(T_{\text{GISM}}) > - < \hat{B}(T_{\text{SIC}}) >$.

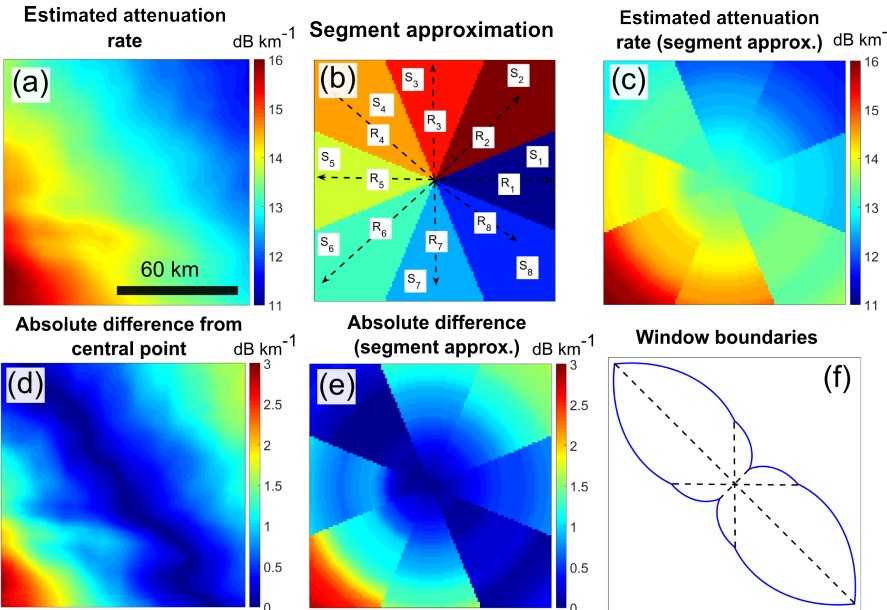

**Figure 6.** Constraining the target window boundaries. (**a**) Estimated attenuation rate, $< \hat{B}(x,y) >$. (**b**) Segment approximation: segments $S_n$=1,...,7,8, radi $R_n$=1,...,7,8 with $n$=1,...,7,8. (**c**) Segment approximation for the attenuation rate, $< \hat{B}(r) >=< \hat{B}(r_n, \theta_n) >$. (**d**) Local tolerance/absolute difference, $\sqrt{(< \hat{B}(x,y) > - < \hat{B}(x_0,y_0) >)^2}$. (**e**) Segment approximation for tolerance, $\sqrt{(< \hat{B}(r_n, \theta_n) > - < \hat{B}(x_0,y_0) >)^2}$. (**f**) Target window boundaries.

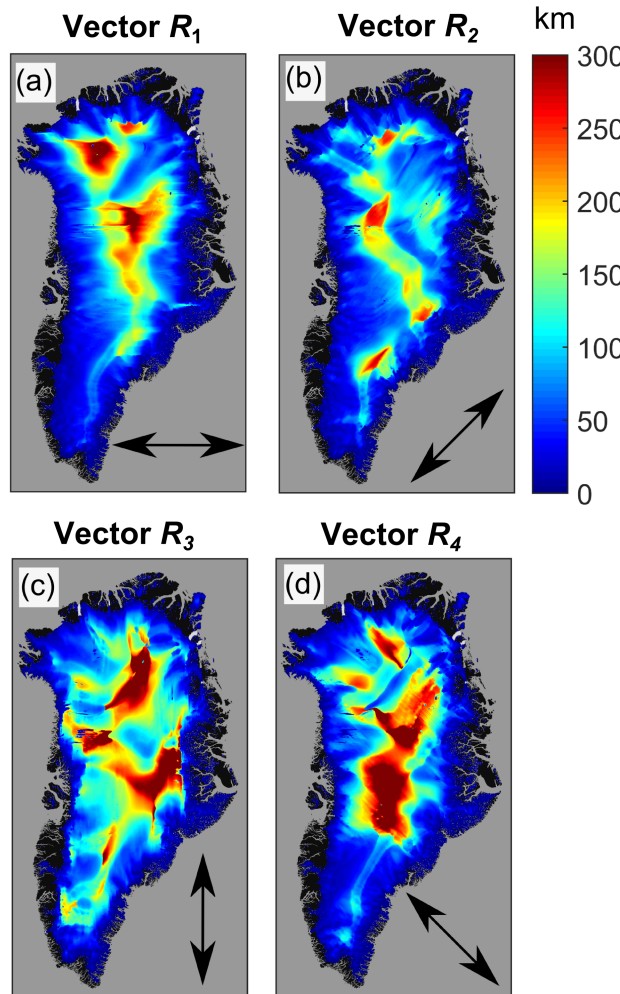

**Figure 7.** Maps for target window radi vector length using the GISM temperature field. **(a)** Vector $R_1$, **(b)** Vector $R_2$, **(c)** Vector $R_3$, **(d)** Vector $R_4$. The orientation of each radi vector is shown in each subplot.

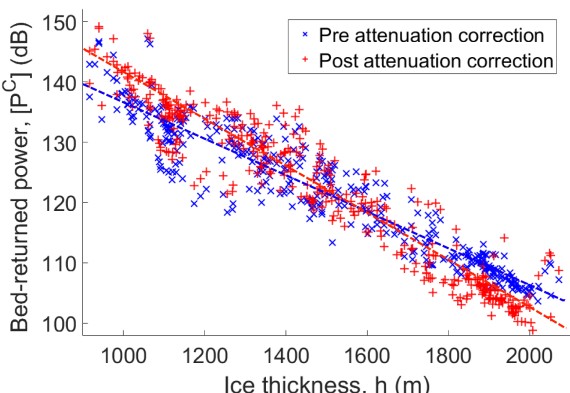

**Figure 8.** Bed-returned power versus ice thickness pre and post local attenuation correction, Eq. (9). The radar-inferred attenuation rate pre correction is $<B>$=15.4 dB km$^{-1}$ ($r^2$=0.56) and post correction is $<B>$=19.3 dB km$^{-1}$ ($r^2$=0.89). The central point of the sample region is 64.30° N, 43.82° W (100 km due South of the Dye 3 ice core) and has ice thickness 1604 m, and target window radi vectors: $R_1$=39 km, $R_2$=55 km, $R_3$=108 km, $R_4$=45 km.

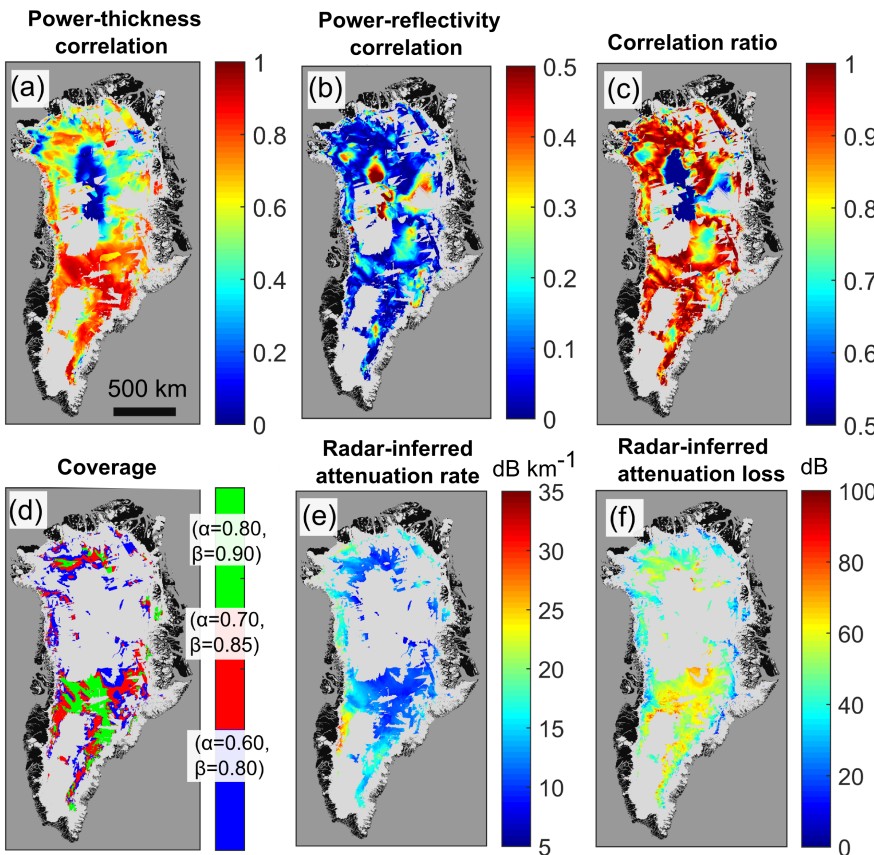

**Figure 9.** Ice sheet wide properties of the radar algorithm using the GISM temperature field. (**a**) Power-thickness correlation, $r^2_{[P^C]}$. (**b**) Arrhenius reflection coefficient-thickness correlation, $r^2_{[\hat{R}]}$. (**c**) Correlation ratio, $r^2_{\text{ratio}}$, Eq. (12). (**d**) Coverage for three thresholds (green is a subset of red and red is a subset of blue). (**e**) Radar-inferred attenuation rate, $< B(T_{\text{GISM}}) >$, for $(\alpha, \beta) = (0.60, 0.80)$. (**f**) Radar-inferred attenuation loss, $[L(T_{\text{GISM}})]$, for $(\alpha, \beta) = (0.60, 0.80)$.

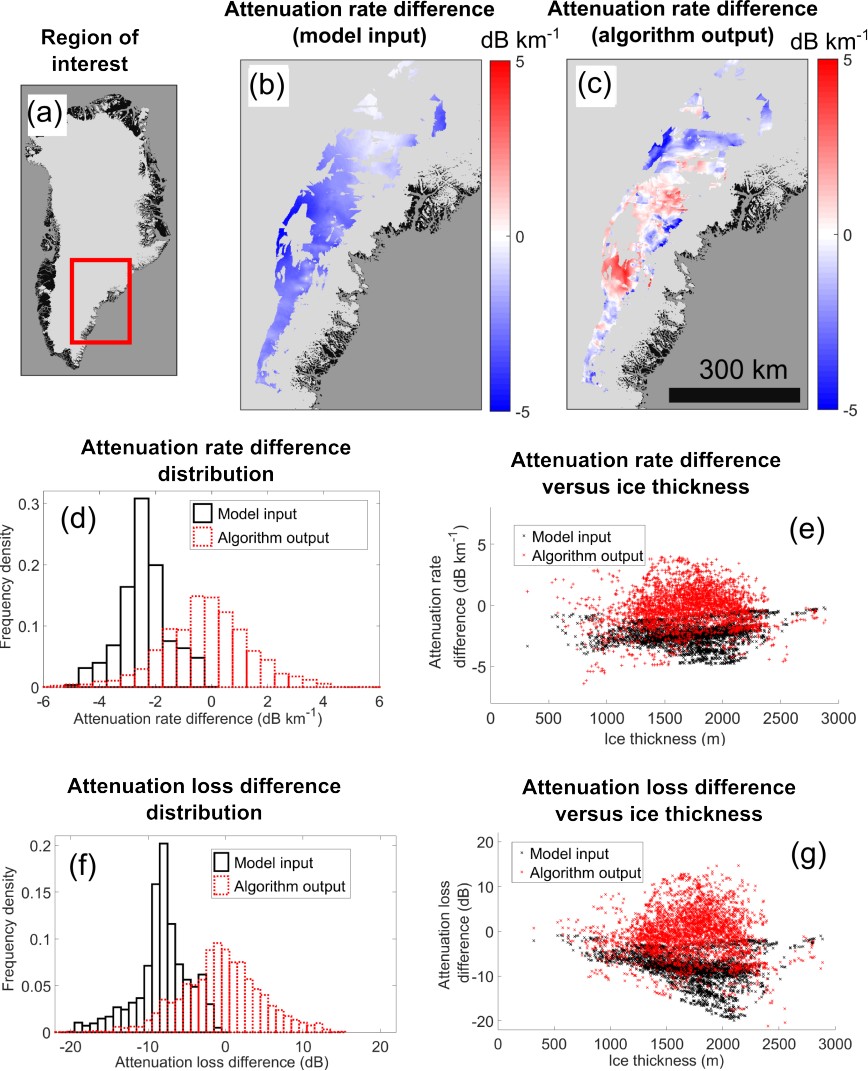

**Figure 10.** Attenuation solution convergence for the SE GrIS. (**a**) Region of interest. (**b**) Map for $< \hat{B}(T_{\mathrm{GISM}}) > - < \hat{B}(T_{\mathrm{SIC}}) >$ (Arrhenius model input). (**c**) Map for $< B(T_{\mathrm{GISM}}) > - < B(T_{\mathrm{SIC}}) >$ (algorithm output). (**d**) Difference distributions for (**b**) and (**c**). (**e**) Thickness dependence for plot (**d**). (**f**) Difference distributions for attenuation loss. (**g**) Thickness dependence for plot (**f**).

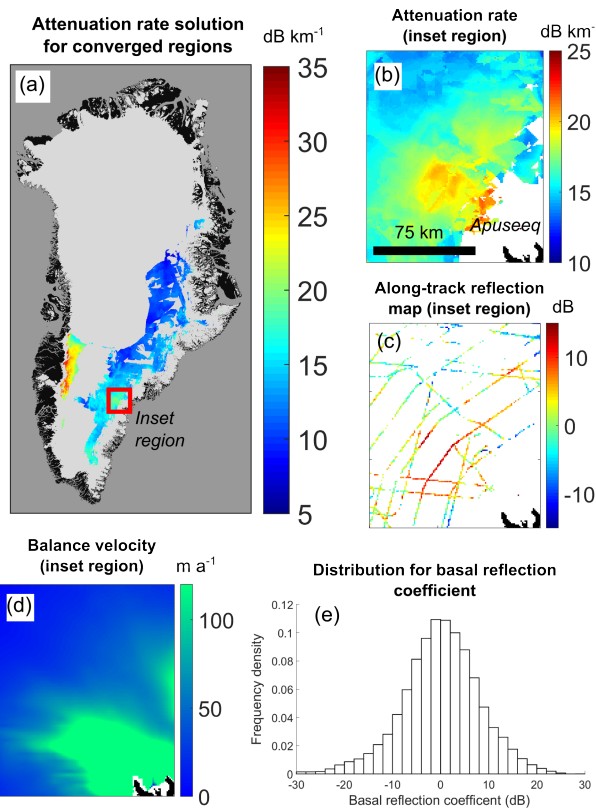

**Figure 11.** Attenuation solution and basal reflection. (**a**) Converged radar-inferred attenuation rate map, $< \bar{B} >$ (average for both input temperature fields). (**b**) Attenuation rate map for inset region. (**c**) Along-track map for basal reflection coefficient for inset region. (**d**) Balance velocities for inset region. (**e**) Probability distribution for basal reflection coefficient for entire coverage region in (**a**). The reflection coefficient is defined using the aggregated power for the basal echo.

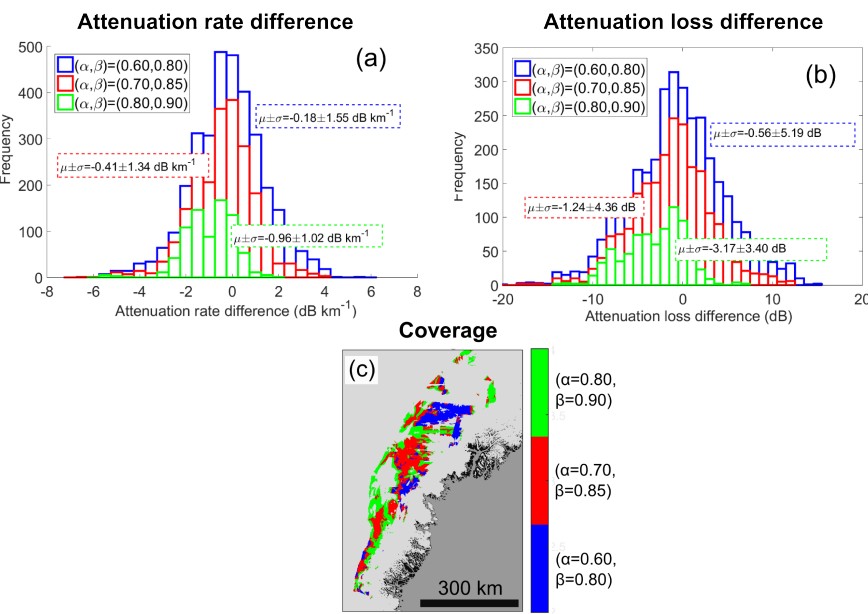

**Figure 12.** Relationship between algorithm coverage and uncertainty as measured by attenuation solution difference distributions. (**a**) Attenuation rate, $< B(T_{\mathrm{GISM}}) > - < B(T_{\mathrm{SIC}}) >$. (**b**) Attenuation loss, $[L(T_{\mathrm{GISM}})]$-$[L(T_{\mathrm{SIC}})]$. (**c**) Algorithm coverage. Green is a subset of red and red is a subset of blue. The region is the same as Fig. 10.

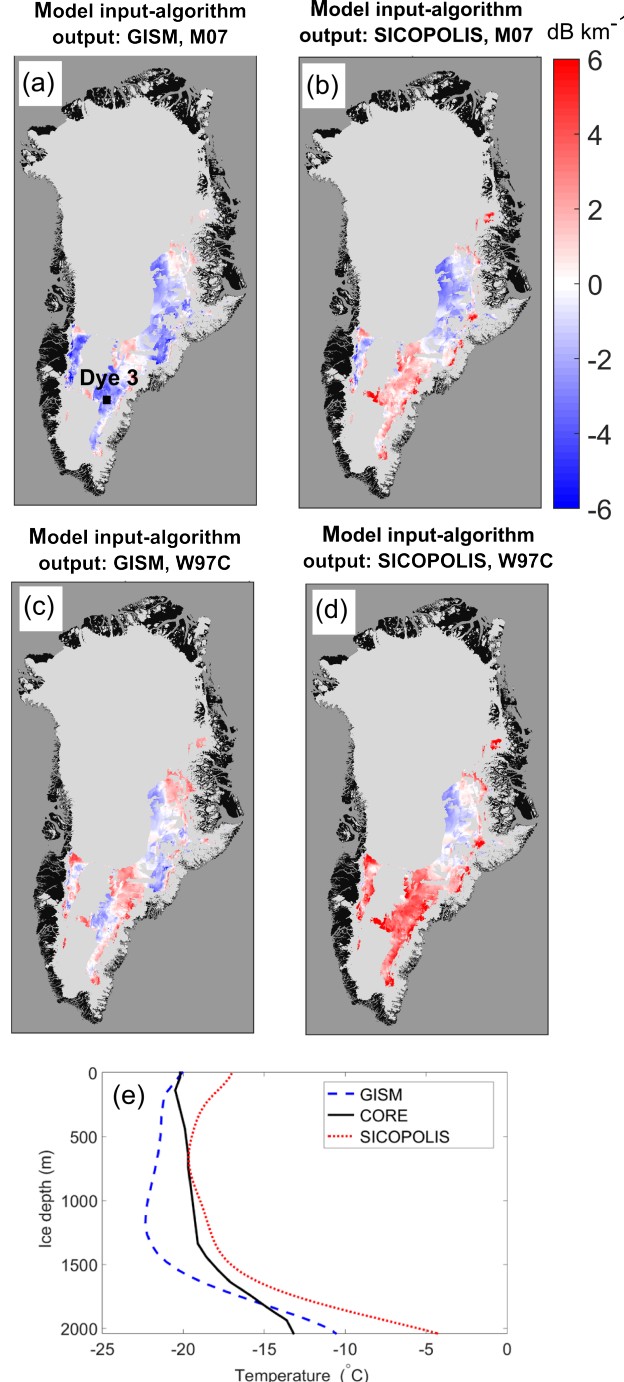

**Figure 13.** Evaluation of temperature bias for ice sheet models using attenuation rate differences. (**a**) $< \hat{B}(T_{\mathrm{GISM}}) > - < \bar{B} >$: M07. (**b**) $< \hat{B}(T_{\mathrm{SIC}}) > - < \bar{B} >$: M07. (**c**) $< \hat{B}(T_{\mathrm{GISM}}) > - < \bar{B} >$: W97C ($\sigma_{195\mathrm{MHz}}/\sigma_{300\mathrm{kHz}}$=1.7). (**d**) $< \hat{B}(T_{\mathrm{SIC}}) > - < \bar{B} >$: W97C ($\sigma_{195\mathrm{MHz}}/\sigma_{300\mathrm{kHz}}$=1.7). Red regions are suggestive of positive bias for depth-averaged temperature and blue regions are suggestive of negative bias. (**e**) Temperature profiles at Dye 3 core. The model temperature profiles are vertically rescaled using the ice core thickness (2038 m), and the core temperature profile is from (Gundestrup and Hansen, 1984).

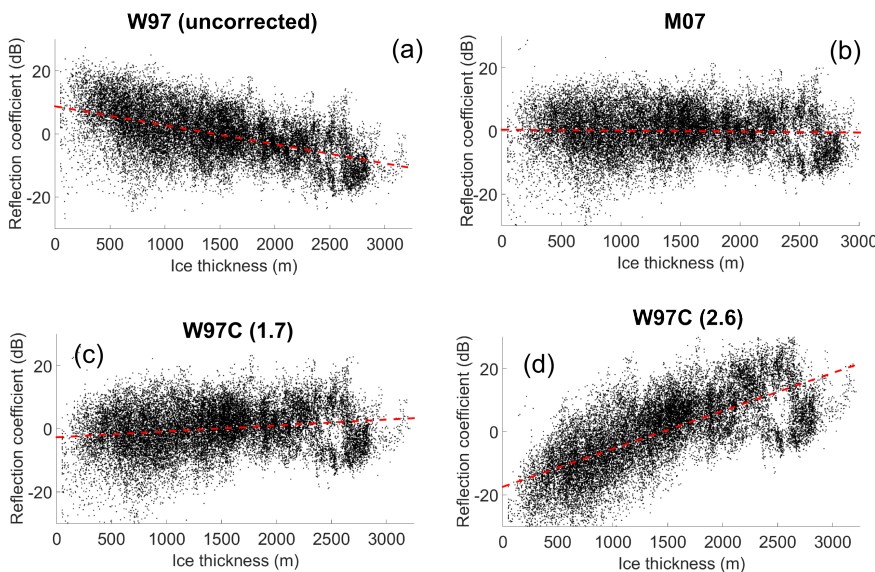

**Figure 14.** Estimated basal reflection coefficient, $[\hat{R}]$, versus ice thickness in northern Greenland for four different conductivity models: (**a**) W97, (**b**) M07, (**c**) W97C ($\sigma_{195\text{MHz}}/\sigma_{300\text{kHz}}$=1.7), (**d**) W97C ($\sigma_{195\text{MHz}}/\sigma_{300\text{kHz}}$=2.6). The negative and positive correlations in (**a**) and (**d**) are interpreted as underestimates/overestimates of the conductivity at the IPR frequency, whereas the near thickness-invariance in (**b**) and (**c**) are interpreted as good estimates of the conductivity. M07 is approximately equivalent to W97C with $\sigma_{195\text{MHz}}/\sigma_{300\text{kHz}}$=1.48.