# Peer review of "An ice sheet wide framework for englacial attenuation from ice penetrating radar data"

_The Cryosphere, 2016_

## Referee Comment (RC1) · J. MacGregor (Referee) · 11 Feb 2016

Review of "An ice-sheet wide framework for englacial attenuation and basal reflection from ice penetrating radar data" by T.M. Jordan et al.

Joseph A. MacGregor 11 February 2016

Summary

This manuscript describes a new method for inferring full-thickness depth-averaged attenuation rates from radar data. It leverages existing techniques and reasonable assumptions regarding the nature of attenuation variability to better understand the Greenland Ice Sheet's attenuation structure. The authors then compare their radar-

derived results in southeastern Greenland to two existing models to infer the spatial variation of their apparent temperature biases, which has inherent value for ice-flow modeling. The potential for this application to larger regions is considered.

The manuscript is novel, well structured, argued, illustrated and written. In those regards, I find little to fault. There are a few places in the method section where greater clarity is needed, and I think the nomenclature can be further simplified. I have a key specific concern regarding the attenuation model used, discussed below. While I wonder about the broader applicability of the method, given the breadth of data already available to the authors, the study as it stands is reasonably complete in terms of introducing and testing the method, and the well-considered Supplementary Material addresses some of these concerns. Repeatedly in the manuscript, where this is a question as to origin and motivation for approach, the authors present an effective justification and place it in context of their overarching goal. As it stands, the methodology presented is a clear advance over earlier techniques. It's not yet immediately applicable to all radar sounding data, but the authors do a superlative job of establishing present strengths and weaknesses. The comparison of temperature biases at DYE-3 and that inferred from the radar is particularly compelling.

The manuscript is clearly appropriate to the readership of The Cryosphere. The authors make a reasonable effort to extend the appeal of the manuscript beyond the relatively small audience interested in radar sounding analysis methods, by addressing the specific ice-sheet property (temperature) that can be constrained.

Major comments

Section 2.4: Following the nomenclature of MacGregor et al. (2015b), the authors select model M07 as their radar-attenuation model of choice. While I recognize that model as having a longer track record, MacGregor et al. (2015b) effectively deprecated that model in favor of a frequency-corrected version of the Wolff et al. (1997) model (W97-corrected). While reasonable people may disagree over the physical significance

of the applied correction to W97, based on Figure 6 of MacGregor et al. (2015b) it is indisputable the W97-corrected model better relates radar-inferred and borehole depth-averaged temperatures than model M07 for the Greenland Ice Sheet. Hence, when relating the spatial variation in depth-averaged radar-attenuation rate to temperature, regardless of chemistry, the W97-corrected model ought to be used.

Sections 2.5 (including Figure 5) and the latter half of 2.6 were the newest and un-surprisingly the hardest concepts to grasp. I got the gist and am comfortable with the approach (the consideration of anisotropy in the horizontal gradient is an important advance), but I recommend additional review of the text in this section for accessibility.

Section 2.8: The gridding method used is never discussed. Based on the distortions present in maps in Figure 6 onwards, I can reasonably guess that a bilinear or natural neighbor gridding was used. While the gridding method is not a critical element of this study, I strongly recommend that ordinary kriging be applied instead. As it stands, a sub-standard gridding perhaps unintentionally diminishes from the results presented. Regardless, clarify the method used.

13-4, 345-6, Section 3.5: These statements and this sub-section seem to imply that this is the first time that the temperature bias of a major ice-sheet model has been evaluated using radar-attenuation estimates, when in fact MacGregor et al. (2015b) did exactly that with the steady-state ISSM instance from Seroussi et al. (2013), albeit not for the entire ice column. Reword.

Minor comments

21: The use of the term "noticeable" here is somewhat odd. "substantial" would be better. The improved coverage is the result of the efforts of innumerable individuals over many years and is intentional.

40: Clarify whether linear or dB units are meant when referring to the range of variability of attenuation rate.

[Figure]

(11): How about defining and using \Delta B_\infty instead? What is _\infty meant to represent, anyway? It seems to originate on 167. I assume it references the high-frequency limit, but that's somewhat obscure and won't be obvious to most readers. I recommend to stick with nomenclature from earlier studies.

(12): What is the value of [S]? If dS/dh is assumed to be negligible, then it is irrelevant to dR_\infty dh, right?

(14): Come up with a more meaningful subscript title than "_{ratio}".

3. Results and discussion: It would be better to separate out results from discussion, as the current structure as labeled is uncommon. However, I find the text quite readable in this section so perhaps this is a non-issue.

503-4: Here I think "As was proposed by MacGregor et al. (2015b)," can be dropped, as the following statement is generally accepted regardless of that earlier study.

555: I don't understand what the qualifier "final" adds here. It implies that the future products that will be generated by this method will not need to be improved, which is a bit strong.

Appendix A: I don't mind this appendix but it closely hews to earlier studies. For brevity, it could be dropped.

580: Better to cite MacGregor et al. (2012) instead of MacGregor et al. (2007) here as the former provided the correct form of the equation shown.

585: M not uM for chemical impurity units, following the dimensions given for the other parameters.

Figures

For at least one Greenland-wide map, perhaps Figure 4c, it would be helpful to have a box representing the focused SE study area of Figure 9 onwards.

Figure 2. Use a legend rather than the caption to explain the coloring. I find this diagram helpful but would like it to be expanded, although I'm not exactly sure how. Perhaps including related figure numbers (thanks for including section numbers) and, more ambiguously, the question/challenge each algorithm step addresses.

Figure 7: Show best-fit lines also.

Figure 9: This figure's panel titles and legends are good examples of where the nomenclature needs some simplifying. Table 1 helps but is not quite enough.

Figures 10 and 11 could be combined into a 2x2 panel. Either way, identify the sub-region for Figure 11 in Figure 10.

Figure 11: Is the quantity shown in panel a actually reflectivity, as is in more common usage? A grayscale or some other color illustrating ice velocity beneath the bed values would be nice (e.g., Figure 1 of Jacobel et al., 2010, The Cryosphere).

Grammar, etc.

Regularly and particularly in the introduction, MacGregor et al. (2015a) is cited where I believe MacGregor et al. (2015b) is meant to be cited.

2: ice sheet 6: englacial 34: subglacial 90: missing parentheses around citations 55: tolerance 281: a hypothetical ice column 441: 63% 544: is present 547: a 'true' Table 1 title: principal

---

## Referee Comment (RC2) · M. Wolovick (Referee) · 18 Feb 2016

**Review:** "An ice-sheet wide framework for englacial attenuation and basal reflection from ice penetrating radar data"

T.M. Jordan, et al., *The Cryosphere*

Review by: Mike Wolovick

**Summary:**

This paper describes a semi-empirical method for estimating attenuation losses and bed reflectivity in radar data from continental ice sheets. The method uses a prior thermal model to estimate the spatial gradient in attenuation. Based on this spatial gradient, the method selects local regions that are expected to have broadly similar attenuation rates based on a segmentation approximation. Within each region, the thermal model is used to correct the observed bed-returned power for the local differences in attenuation relative to the mean for that region. The corrected bed-return power data are then fit with a least-squares linear best-fit representing the mean attenuation rate for that region. The residual to the fit represents the basal reflectivity.

The authors apply the method to the CReSIS radar dataset collected over the Greenland Ice Sheet for Operation IceBridge. They find that the method converges where ice thickness is variable and attenuation rates are high in the south and east of Greenland. In the north and west, as well as in the ice sheet interior, the method does not converge. They find a tradeoff between spatial coverage and precision, such that the area where the method is considered to have converged increases if one is willing to accept higher uncertainty. The output bed reflectivity estimates show a reduced spread consistent with the range of plausible subglacial materials. The authors also demonstrate that the estimated attenuation rates can be used to check the temperature bias in the original input models.

This manuscript is clearly relevant for *The Cryosphere*. The method developed by the authors represents an important advance in the integration of ice-penetrating radar data with ice sheet models. The authors demonstrate both how models can be used to guide the interpretation of radar data, and how the radar data can in turn be used to diagnose biases in those models. I have two major concerns, one relating to the inability of the model to converge in the ice sheet interior, and the other relating to the segmentation approximation. The first concern is actually an opportunity for the authors to expand

the scope of their results.  I believe that they have been too conservative, and that in fact they can constrain reflectivity in the interior of Greenland even if they cannot also constrain attenuation.  The second concern could be addressed by showing how the results respond to alternate means of choosing local sample regions.

**Major Comments:**

Ice Sheet Interior

The main weakness of the method developed by the authors is that it does not work in interior regions of Greenland where variations in ice thickness and attenuation rate are both low.  In other words, the method doesn't work where the problem is easy!  With little variability in ice thickness and relatively constant thermal structure, one could get good estimates of the basal reflectivity anomaly in interior Greenland by using no attenuation correction at all.

The reason why the authors' method does not converge in the interior of Greenland is one that the authors themselves identified:  the variability in ice thickness is too low.  Quantitatively, we can say that in order to get a good correlation between bed returned power and ice thickness, the following condition must hold:  $2B\Delta h > \Delta R$, where B is the regional average attenuation rate, $\Delta h$ is the standard deviation of ice thickness, and $\Delta R$ is the standard deviation of bed reflectivity.  When the variability in ice thickness is below this threshold, a local linear best-fit cannot constrain attenuation rate.

However, when the variability in ice thickness is low, the total attenuation losses should also be roughly constant.  Attenuation therefore becomes less important, and it should be possible to estimate basal reflectivity anomalies even if attenuation cannot be constrained.  Local variations in attenuation rate may still produce variability in total attenuation losses; however, the authors are already using a numerical model to estimate local variations in attenuation rate.  The authors make a good case that the local gradient in attenuation rate from the models is more reliable than the mean value, so they can simply continue using the model to correct for local variations in attenuation rate.  When the variability in ice thickness is low, the authors' method will be unable to constrain the regional mean attenuation rate, but it should still do a good job of estimating basal reflectivity, and the authors could present those reflectivity results in the interior of Greenland.

In order to capture regions where it is possible to constrain reflectivity but not attenuation, the

authors could introduce an alternate quality control check as a substitute for the $r^2_{[Pc]}$ check they introduce in Section 2.7. When the variability in ice thickness is low compared to the variability in basal reflectivity ($2B\Delta h < \Delta R$), the authors could check the standard deviation of bed reflectivity instead of $r^2_{[Pc]}$. The authors use the standard deviation of reflectivity as a check on the validity of their results anyway (Section 3.2), so it makes sense to formally add this metric to the quality control step. If the standard deviation of bed reflectivity is reasonable for subglacial materials, then the alternate quality control is passed and the authors can present results for reflectivity in that region.

Segmentation Approximation

The segmentation approximation seems needlessly complex. The purpose of the segmentation approximation (Section 2.5) is to define a local region in which attenuation rate is roughly constant. Why not simply define an oval-shaped region where the RMS variability in attenuation rate is less than some threshold? Or why not simply define an irregular contiguous region containing all grid cells where the difference in attenuation rate is less than the threshold? With an ellipse, the unknowns at this step of the problem would be reduced to three: the orientation of the elipse, the length of the major axis, and the length of the minor axis. With an irregular shape, no a priori assumptions about the nature of the ice sheet temperature field need to be made at all. Using an ellipse instead of the segmentation approximation would eliminate the sharp corners created along the segment boundaries in Figure 5f.

In addition, an ellipse or an irregular shape would drastically simplify Section 2.5. Nothing that the authors have presented indicates that the segmentation approximation is a particularly good representation of the ice sheet thermal structure. The segmentation approximation has no physical basis in ice dynamics or temperature that could justify the use of such a complex model. The only virtue of the segmentation approximation seems to be that it is capable of elongating perpendicular to the gradient of attenuation rate, but an ellipse or an irregular shape could do that too. In addition, the segmentation approximation is only capable of elongating at 45º angles, but an ellipse or irregular shape could elongate at any angle.

I do not believe that the awkwardness of the segmentation approximation invalidates the later results of this paper. It is likely that the authors would have achieved similar results with any reasonable method for selecting a local sample region based on the input thermal models. However, the unnecessary complexity of the segmentation approximation makes the paper harder to follow, has no realistic basis in ice sheet physics, and potentially contributes other artifacts into the results.

The supplemental material (Sections S1 and S2) explores the sensitivity of the sample regions produced by the segmentation approximation to the temperature model input (S1) and to the choice of RMS tolerance (S2). However, neither section addresses the sensitivity of later results to the segmentation approximation itself. I would like to see an exploration of how the results are affected by completely different means of choosing a local sample region. What happens if the segment boundaries are shifted by 22.5º (half a segment)? What happens if six segments are used instead of eight? What if an elliptical, a circular, or an irregular sample region is used? The authors need not address every single possible method of determining local sample regions, but I would like to see some exploration of the effects of using a segmentation approximation to choose local sample regions.

**Minor Comments:**

Line 4: "...which is an exponential function of temperature."
Attenuation is an Arrhenius function of temperature, not an exponential function.

Line 5 (and elsewhere): "stationarity"
I'm not sure I agree with the authors' use of the term "stationary". Typically, "stationarity" refers to a time series whose statistical properties are constant over time, and that concept could be generalized to spatial data whose statistical properties are constant over space. However, the authors use "stationarity in the attenuation rate" to mean a constant attenuation rate. A constant is stationary, but not all stationary data are constant. In this case, the authors could be both easier to understand and more accurate by saying "constant" when they mean constant.

Lines 16-31: Several places in this paragraph could benefit from adding additional (often older) references.
ice thickness: add [*Bailey et al.*, 1964; *Evans and Robin*, 1966; *Robin et al.*, 1969; *Jankowski and Drewry*, 1981]
basal material properties: add [*Oswald and Robin*, 1973; *Peters et al.*, 2005]
internal layer structure: add [*Robin et al.*, 1969; *Conway et al.*, 1999, 2002; *Vaughan et al.*, 1999; *Fahnestock et al.*, 2001; *Dahl-Jensen et al.*, 2003; *Ng and Conway*, 2004; *Tikku et al.*, 2004]

new data products for bed elevation and ice thickness: add [*Morlighem et al.*, 2014]

Additional uses of radar data that could be added to this paragraph:

ice rheology: [*Raymond*, 1983; *Hindmarsh et al.*, 2011; *Kingslake et al.*, 2014]

grounding line dynamics: [*Conway et al.*, 1999; *Catania et al.*, 2006; *Christianson et al.*, 2013]

basal melting or freezing: [*Fahnestock et al.*, 2001; *Catania et al.*, 2010; *Bell et al.*, 2011]

ice dynamic changes: [*Conway et al.*, 2002; *Bingham et al.*, 2015]

Line 33:

add [*Oswald and Robin*, 1973]

Line 38: "Arrhenius models where the attenuation rate is an exponential function of **inverse** temperature"

Line 45: "...make the implicit assumption..."

In some papers, the assumption is explicit. Maybe just say "...make the assumption..."

Line 45: "locally stationary"

See my comment above. "Locally stationary" is an oxymoron; stationarity implies that statistical properties are globally constant. Use "locally constant" here instead.

Line 53: "A central feature of our algorithm is the use of a prior Arrhenius model estimation of the attenuation rate as an initial condition."

The use of the phrase "initial condition" is incorrect here. Initial conditions apply to models that predict the evolution of some variable over time. A better term would be "first guess", "initial guess", or "initial estimate".

Lines 54-55: "Conceptually, the initial condition is used to estimate regions where the assumption of stationarity is valid within some specified tolerance."

Removing "stationarity" can clean up this sentence: "The initial estimate is used to determine regions where attenuation rate is approximately constant to within some specified tolerance."

Line 158: "...exponential dependence upon **inverse** temperature..."

Line 160, 164 (and possibly elsewhere):

Be careful about MacGregor et al., [2015a] versus MacGregor et al., [2015b]. I'm pretty sure you mean the second one in this context.

Lines 165-172:

Somewhere in here would be a good place to indicate that brackets <X> indicate the depth-averaged value of X.

Lines 173-181:

For completeness, it would be a good idea to state how big of an effect you expect to see from climate transitions in Greenland. The MacGregor et al., [2012] reference refers to East Antarctica, where climate transitions have both a smaller signal in ice chemistry and are more closely spaced in depth than in Greenland. In Greenland, the stratigraphic chemistry changes are dominated by the Holocene-LGM transition, which occurs at wildly different depths in southern and northern Greenland [*MacGregor et al.*, 2015]. The difference in depth of the Holocene-LGM transition with respect to the warm ice near the bed might be expected to produce a large difference in attenuation rate between northern Greenland and southern Greenland.

Lines 190-191:

Simplify this sentence to: "For the majority of the IPR data coverage region, GISM has lower temperature and therefore lower attenuation rate than SICOPOLIS (Fig. 4c).

Line 215

There is an extra parenthesis inside the square root sign.

Lines 216 and 217

The averaging brackets <> are in the wrong place. As written, the square root cancels the square and the expression reduces to the absolute value of the difference. The brackets should go outside of the squared difference. The expression inside the square root sign should be: $<(B_\infty(x,y)-B_\infty(x_0,y_0))^2>$ (and

likewise for the second expression).

The alternate possibility is that the averaging brackets represent column-averages, not areal averages. In that case, the expressions reduce to the absolute value of the difference.

Lines 208-217

This paragraph is very confusing. It sounds like the authors assume only radial dependence within each segment, but later, the sample region boundaries (Figure 5f) clearly show a dependence on angle even within each segment. This is because the region boundaries are interpolated from the central radius vector for each segment along a circular arc, in order to produce continuous (but not differentiable) window boundaries. It would be helpful to state somewhere in this paragraph that the ultimate goal is to produce a variable radial length of the target window by interpolating with respect to angle.

Equations 6 and 7:

Move the constant terms outside of the integral and simplify them to $2/R^2_n$.

As written, the RMS is also a function of $\theta_n$.

Lines 237-239:

See my major comment above. If the gradient of ice thickness with distance is small, it should be easy to estimate bed reflectivity, because the mean attenuation rate has little effect on total attenuation. Variations in total attenuation only arise from variations in the attenuation rate, and the method the authors develop relies on an a priori model to correct for local variations in depth-averaged attenuation anyway.

Lines 259-261:

Most readers probably know this already, but it still might be helpful to explicitly state that crevasse scattering is most likely to cause problems for radar analysis of basal conditions in fast-flowing regions near the ice sheet margin.

Lines 279-282:

Simplify this statement. This step corrects for the difference in attenuation rate between the

measurement point and the central point.

Lines 283-287:

Why are thinner ice columns warmer, and thicker ice columns colder? On the one hand, conductive cooling should tend to produce warmer conditions in thicker ice. Most people reading this paper would probably assume that thick=warm and thin=cold. On the other hand, the Peclet number ($Pe=ah/\kappa$) indicates that thicker ice columns should have a greater dominance of advection over diffusion, and therefore the cooling effect of surface accumulation should be greater in thick ice. In addition, ice must flow faster (with a higher driving stress) in thin regions, producing more shear heating. Finally, low surface elevations tend to have a higher surface temperature because of the atmospheric lapse rate, but this effect should not influence local temperature differences due to bed topography. Which one of these mechanisms is responsible for producing the thick=cold, thin=warm association?

Section 2.7

State the two criteria in words at the beginning of this section. Something like, "As a quality control check, we are looking for regions where (1) the correlation between ice thickness and bed-returned power is good, and (2) the correlation between ice thickness and bed reflectivity is poor."

Lines 349-350:

Why were the field seasons processed independently, if Lines 81-83 stated that power measurements from different field seasons could be combined?

Section 3.2:

State the definition of "convergence" up front. The reader has to wait until the last paragraph (Line 415) to learn that convergence means "a normally distributed difference centered on zero".

Lines 413-421:

The first sentence of this paragraph should state that the algorithm converges in southern and eastern Greenland, but not northern or western Greenland. The reader should not have to flip back and forth to the figures to determine where the basin numbers are located spatially.

Line 437: "A gridded map of the basal reflection coefficient...is shown in Fig. 11a."

Figure 11a does not look like a "gridded map". I realize that technically it is gridded at 1km cell size, but for all intents and purposes Figure 11a shows reflection coefficients along-track.

Lines 504-508:

It would be appropriate to mention here that the relationship between attenuation rate and temperature is highly nonlinear, so the difference in depth-averaged attenuation rate does not transfer neatly to a difference in depth-averaged temperature. $mean(x^2) \neq mean(x)^2$.

Section 4: Conclusions

This section should have a paragraph commenting on and interpreting the reflectivity results. From Figure 11, it appears that high reflectivity is concentrated in the approach to fast-flowing outlet glaciers. This is consistent with distributed hydrological networks or with saturated subglacial till, either of which would promote faster sliding.

Line 542-543: "We suggest that the converged radar algorithm attenuation solution is preferable to using a forward Arrhenius temperature model to calculate basal reflection coefficients."

Strengthen and clarify this conclusion: "We find that our data-based attenuation algorithm is superior to an attenuation correction calculated purely from an a priori temperature model."

Lines 546-548: "Notably, we demonstrated that even a small constant bias in the attenuation rate across a region; (this could be either with respect to a "true" value or another modelled value), leads to a thickness correlated bias in attenuation loss and therefore the basal reflection coefficients."

This sentence is awkward. Rephrase as: "We demonstrated that even a small regional bias in attenuation rate leads to thickness-correlated errors in attenuation losses and therefore the basal reflection coefficients. These thickness-correlated errors persist regardless of whether the regional bias is with respect to the 'true' value or to another modelled value."

Lines 562-564:

Is interpolation of bed reflectivity onto a regular grid even desirable, given that subglacial hydrology and geomorphology are likely to vary at scales much smaller than the grid spacing?

Lines 566-569: "Due to this lower spatial variability, (and despite the caveats in the paragraph above), these regions [ice sheet interiors] could potentially have their basal reflection values derived by using forward Arrhenius temperature model for the attenuation."

See my major comments above. When ice thickness has little variability, errors in the regional mean attenuation rate have little effect. Only the spatial gradients in attenuation rate matter, and as the authors point out earlier in the conclusion, the models do a better job representing these than they do at representing the mean value. The authors should have been able to take advantage of this fact to produce reflectivity estimates in the ice sheet interior.

**Figures:**

In general, the figures need better subplot titles and labeling. Symbols without words are inappropriate for subplot or axes labels because symbols are hard for readers to understand without flipping back and forth to the places where those symbols are defined. The subplot titles should express their meaning in words, and the corresponding symbols can be given in the caption if necessary. Many of the figures also need to be larger to permit more detail and wordier labels. Units should be placed on colorbar labels, not in subplot titles.

For most of the figures, I've given my suggestions for more descriptive titles and labels. The authors need not follow these specific suggestions, but all of the subplot titles should use descriptive words rather than symbols.

Figure 1
Subplot titles:
a) Flight Tracks
b) Drainage Basins
Put the numbers for the drainage basins in (b) arrayed around the coast of Greenland, rather than all together in the key. That way it is easier to tell at a glance which number refers to which basin. Also, it might be a good idea to circle or otherwise highlight the four basins in which the algorithm converges.

Figure 3

Subplot titles:

a) Good Return

b) Bad Return

Y-axis label: What does "linear units" mean, other than "not decibels"? Either convert to actual units of power (W or $Wm^{-2}$), express as a fraction of the transmit power, or normalize so that the peak in each plot is 1. Normalization may be the best option, so that the quality control check (decays to 2% of peak power) can be easily visualized.

State where the two examples were taken from in the caption.

Figure 4

Subplot titles:

a) Arrhenius Model for Attenuation Rate

b) Attenuation Rate from GISM

c) Difference between GISM and SICOPOLIS

The y-axis label in plot (a) should say the words "Attenuation Rate".

The colorbars should be labeled with their units (dB/km).

It might be appropriate to include a map for SICOPOLIS itself, in addition to the difference map.

Figure 5

subplots:

a) Model Estimated Attenuation Rate

b) Segments

c) Segment Approximation of Model Estimate

d) Difference from Central Value

e) Segment Approximation of Difference

f) Window Boundaries

The units should be given next to the colorbars, not in the subplot titles. The colorbars should have a larger font size as well.

Don't the square root and square cancel in plot (d)? Isn't that plot just showing the absolute value of the

difference?

The same comment that I made about lines 216 and 217 applies to the expressions in the caption. Either the averaging brackets are in the wrong place, or the expressions reduce to the absolute value of the difference.

Figure 6

Subplots:

a) Vector R1

b) Vector R2

c) Vector R3

d) Vector R4

Colorbar label should be "Length (km)"

Figure 7

Plot title: "Attenuation Difference Correction"

It is more accurate to refer to the process shown in this figure as the attenuation difference correction, rather than the attenuation correction. The step only corrects for the difference in attenuation rate between the data location and the central point.

Figure 8

Subplots:

a) Correlation Between Power and Ice Thickness ($r^2_{[Pc]}$)

b) Correlation Between Reflectivity and Ice Thickness ($r^2_{[R\infty]}$)

c) Correlation Ratio ($r^2_{[ratio]}$)

d) Coverage

e) Attenuation Rate

f) Attenuation Loss

Put units (dB, dB/km) in the colorbar labels.

Put plots a-c on the same color scale.

Note in caption whether high values or low values are good in a-c. In (a) and (c), high values indicate the algorithm converged, but in (b) low values indicate convergence.

Figure 9

Subplots:

a) Difference between Model Inputs

b) Difference between Algorithm Outputs

c) Attenuation Rate Difference Distribution

d) Attenuation Rate Difference vs Ice Thickness

e) Attenuation Loss Difference Distribution

f) Attenuation Loss Difference vs Ice Thickness

Add units to the colorbar.

Label important outlet glaciers in either (a) or (b). Helheim Glacier is in view here, for example.

Note in the caption what the reader should be looking for in terms of convergence: in plots (c) and (e), a normally distribution about zero indicates convergence, while in (d) and (f), a lack of systematic ice thickness dependence indicates convergence.

Figure 10

Plot title: Attenuation Rate

Label colorbar with the units.

As in Figure 9, label important outlet glaciers, including Helheim. Also put an inset box showing the area of detail in Figure 11.

Figure 11

Subplots:

a) fine as is

b) Reflectivity Distribution

c) Reflectivity vs Ice Thickness

Put units on the colorbar label. Label the outlet glacier(s) in the lower right of (a). Note in the caption that a range of approximately 20 dB in (b) is right for plausible subglacial materials, and that a lack of systematic ice thickness dependence in (c) indicates algorithm convergence.

Figure 12

Subplots:

a) Attenuation Rate Difference Distribution

b) Attenuation Loss Difference Distribution

c) fine as is

Note in the caption that green is a subset of red, which is a subset of blue.

Figure 13

Subplots:

a) Difference Between Prior Model and Radar Estimate (GISM)

b) Difference Between Prior Model and Radar Estimate (SICOPOLIS)

c) Comparison Between Models and Dye3 Ice Core

Add units to the colorbar labels.

**Supplemental Material:**

Line 33: "...the sample regions will contain individual ice columns..."

Replace "individual ice columns" with "grid cells".

Lines 43-45: "If this is rescaled by ice thickness for a sample region in the interior of the ice sheet (mean ice thickness ~2500m) this results in a desired attenuation rate accuracy ~1 dBkm$^{-1}$."

This explanation is very simple and should go in the main text.

Lines 69-70:

Mention that basins 3, 4, 5, and 6 are all located in south and east Greenland.

Figures:

Same comments as for the figures in the main text. All of these figures need better subplot titles and units labelled on the colorbars.

**References:**

Bailey, J. T., S. Evans, and G. de Q. Robin (1964), Radio echo sounding of polar ice sheets, *Nature*, *204*(4957), 420–421, doi:10.1038/204420a0.

Bell, R. E. et al. (2011), Widespread persistent thickening of the East Antarctic Ice Sheet by freezing from the base, *Science*, *331*(6024), 1592–1595, doi:10.1126/science.1200109.

Bingham, R. G., D. M. Rippin, N. B. Karlsson, H. F. J. Corr, F. Ferraccioli, T. A. Jordan, A. M. Le Brocq, K. C. Rose, N. Ross, and M. J. Siegert (2015), Ice-flow structure and ice dynamic changes in the Weddell Sea sector of West Antarctica from radar-imaged internal layering, *J. Geophys. Res. Earth Surf.*, *120*(4), 655–670, doi:10.1002/2014JF003291.

Catania, G., C. Hulbe, and H. Conway (2010), Grounding-line basal melt rates determined using radar-derived internal stratigraphy, *J. Glaciol.*, *56*(197), 545–554, doi:10.3189/002214310792447842.

Catania, G. A., H. Conway, C. F. Raymond, and T. A. Scambos (2006), Evidence for floatation or near floatation in the mouth of Kamb Ice Stream, West Antarctica, prior to stagnation, *J. Geophys. Res. Earth Surf.*, *111*(F1), F01005, doi:10.1029/2005JF000355.

Christianson, K., B. R. Parizek, R. B. Alley, H. J. Horgan, R. W. Jacobel, S. Anandakrishnan, B. A. Keisling, B. D. Craig, and A. Muto (2013), Ice sheet grounding zone stabilization due to till compaction, *Geophys. Res. Lett.*, *40*(20), 5406–5411, doi:10.1002/2013GL057447.

Conway, H., B. L. Hall, G. H. Denton, A. M. Gades, and E. D. Waddington (1999), Past and future grounding-line retreat of the West Antarctic Ice Sheet, *Science*, *286*(5438), 280–283, doi:10.1126/science.286.5438.280.

Conway, H., G. Catania, C. F. Raymond, A. M. Gades, T. A. Scambos, and H. Engelhardt (2002), Switch of flow direction in an Antarctic ice stream, *Nature*, *419*(6906), 465–467, doi:10.1038/nature01081.

Dahl-Jensen, D., N. Gundestrup, S. P. Gogineni, and H. Miller (2003), Basal melt at NorthGRIP modeled from borehole, ice-core and radio-echo sounder observations, *Ann. Glaciol.*, *37*(1), 207–212, doi:10.3189/172756403781815492.

Evans, S., and G. de Q. Robin (1966), Glacier depth-sounding from the air, *Nature*, *210*(5039), 883–885, doi:10.1038/210883a0.

Fahnestock, M., W. Abdalati, I. Joughin, J. Brozena, and P. Gogineni (2001), High geothermal heat flow, basal melt, and the origin of rapid ice flow in central Greenland, *Science*, *294*(5550), 2338–2342, doi:10.1126/science.1065370.

Hindmarsh, R. C. A., E. C. King, R. Mulvaney, H. F. J. Corr, G. Hiess, and F. Gillet-Chaulet (2011), Flow at ice-divide triple junctions: 2. Three-dimensional views of isochrone architecture from ice-penetrating radar surveys, *J. Geophys. Res. Earth Surf.*, *116*(F2), F02024, doi:10.1029/2009JF001622.

Jankowski, E. J., and D. J. Drewry (1981), The structure of West Antarctica from geophysical studies,

*Nature*, *291*(5810), 17–21, doi:10.1038/291017a0.

Kingslake, J., R. C. A. Hindmarsh, G. Adalgeirsdottir, H. Conway, H. F. J. Corr, F. Gillet-Chaulet, C. Martin, E. C. King, R. Mulvaney, and H. D. Pritchard (2014), Full-depth englacial vertical ice sheet velocities measured using phase-sensitive radar, *J. Geophys. Res. Earth Surf.*, *119*(12), 2604–2618, doi:10.1002/2014JF003275.

MacGregor, J. A., M. A. Fahnestock, G. A. Catania, J. D. Paden, S. Prasad Gogineni, S. K. Young, S. C. Rybarski, A. N. Mabrey, B. M. Wagman, and M. Morlighem (2015), Radiostratigraphy and age structure of the Greenland Ice Sheet, *J. Geophys. Res. Earth Surf.*, *120*(2), 212–241, doi:10.1002/2014JF003215.

Morlighem, M., E. Rignot, J. Mouginot, H. Seroussi, and E. Larour (2014), Deeply incised submarine glacial valleys beneath the Greenland ice sheet, *Nat. Geosci.*, *7*(6), 418–422, doi:10.1038/ngeo2167.

Ng, F., and H. Conway (2004), Fast-flow signature in the stagnated Kamb Ice Stream, West Antarctica, *Geology*, *32*(6), 481–484, doi:10.1130/G20317.1.

Oswald, G. K. A., and G. D. Q. Robin (1973), Lakes beneath Antarctic Ice Sheet, *Nature*, *245*(5423), 251–254, doi:10.1038/245251a0.

Peters, M. E., D. D. Blankenship, and D. L. Morse (2005), Analysis techniques for coherent airborne radar sounding: Application to West Antarctic ice streams, *J. Geophys. Res. Solid Earth*, *110*(B6), B06303, doi:10.1029/2004JB003222.

Raymond, C. (1983), Deformation in the vicinity of ice divides, *J. Glaciol.*, *29*(103), 357–373.

Robin, G. D. Q., S. Evans, and J. T. Bailey (1969), Interpretation of radio echo sounding in polar ice sheets, *Philos. Trans. R. Soc. Lond. Math. Phys. Eng. Sci.*, *265*(1166), 437–505, doi:10.1098/rsta.1969.0063.

Tikku, A. A., R. E. Bell, M. Studinger, and G. K. C. Clarke (2004), Ice flow field over Lake Vostok, East Antarctica inferred by structure tracking, *Earth Planet. Sci. Lett.*, *227*(3-4), 249–261, doi:10.1016/j.epsl.2004.09.021.

Vaughan, D. G., H. F. J. Corr, C. S. M. Doake, and E. D. Waddington (1999), Distortion of isochronous layers in ice revealed by ground-penetrating radar, *Nature*, *398*(6725), 323–326, doi:10.1038/18653.

---

## Author Comment (AC1) · 4 Apr 2016

**Author comments:** "An ice-sheet wide framework for englacial attenuation and basal reflection from ice penetrating radar data"

T.M. Jordan, et al., *The Cryosphere*

Review by: Joseph Macgregor

*We thank the reviewer for their very constructive and thoughtful comments. They have well understood both the motivation for our study and the methods developed. In order to address the reviewers' specific concern (about the choice of conductivity/attenuation model) we have completed a further investigation, and we describe this along with our proposed changes. We agree with the vast majority of the other suggestions and we provide detailed feedback to each point below, with our comments italicised in blue text.*

This manuscript describes a new method for inferring full-thickness depth-averaged attenuation rates from radar data. It leverages existing techniques and reasonable assumptions regarding the nature of attenuation variability to better understand the Greenland Ice Sheet's attenuation structure. The authors then compare their radar-inferred values in Southeastern Greenland to two existing models to infer the spatial variation of their apparent temperature biases, which has inherent value for ice-flow modeling. The potential for this application to larger regions is considered.

The manuscript is novel, well structured, argued, illustrated and written. In those regards, I find little to fault. There are a few places in the method section where greater clarity is needed, and I think the nomenclature can be further simplified. I have a key specific concern regarding the attenuation model used, discussed below. While I wonder about the broader applicability of the method, given the breadth of data already available to the authors, the study as it stands is reasonably complete in terms of introducing and testing the method, and the well-considered Supplementary Material addresses some of these concerns. Repeatedly in the manuscript, where this is a question as to origin and motivation for approach, the authors present an effective justification and place it in context of their overarching goal. As it stands, the methodology presented is a clear advance over earlier techniques. It's not yet immediately applicable to all radar sounding data, but the authors do a superlative job of establishing present strengths and weaknesses. The comparison of temperature biases at DYE-3 and that inferred from the radar is particularly compelling.

The manuscript is clearly appropriate to the readership of The Cryosphere. The authors make a reasonable effort to extend the appeal of the manuscript beyond the relatively small audience interested in radar sounding analysis methods, by addressing the specific ice-sheet property (temperature) that can be constrained
* * *
Section 2.4: Following the nomenclature of MacGregor et al. (2015b), the authors select model M07 as their radar-attenuation model of choice. While I recognize that model as having a longer track record, MacGregor et al. (2015b) effectively deprecated that model in favor of a frequency-corrected version of the Wolff et al. (1997) model (W97-corrected). While reasonable people may disagree over the physical significance of the applied correction to W97, based on Figure 6 of MacGregor et al. (2015b) it is indisputable the W97-corrected model better relates radar-inferred and borehole depth-averaged temperatures than model M07 for the Greenland Ice Sheet. Hence, when relating the spatial variation in depth-averaged radar-attenuation rate to temperature, regardless of chemistry, the W97-corrected model ought to be used.

*We agree that, in view of the results in Macgregor et al. (2015b), further investigation regarding the choice of conductivity/attenuation model is desirable to complete our study. Based upon our new results, which considers the W97-corrected model alongside the M07 model, we suggest the following changes to the manuscript.*

***(i) Revised Appendix A (choice of conductivity model).***

*Following the nomenclature of Macgregor et al. (2015b), we now outline the background to the W97 model (Wolf et al. 1997), and its frequency corrected form: W97-corrected. To the best of our knowledge, the W97-corrected model has not been used to calculate full ice column losses and basal reflection values. Subsequently, we propose a test for the validity of the different conductivity models, based upon the distribution for the Arrhenius model/estimated basal reflection values as a function of ice thickness. Strictly, this decibel range need not be thickness-invariant, but a strong positive/negative correlation would indicate a significant over/underestimate for the conductivity at the frequency of the IPR measurements.*

*We consider 4 conductivity models: (a) W97 (uncorrected), (b) M07, (c) W97-corrected (with empirical correction parameter=1.7), (d) W97-corrected (with empirical correction parameter=2.6). Model (c) is consistent with the frequency dependence observed in Paden et al. (2005), and model (d) is the best fit value from Macgregor et al. (2015b). Northern Greenland (basin 1 in Fig. 1b) is chosen to illustrate our results as the predicted spatial variation in attenuation rate is relatively low in this region, and therefore the effects of changing the conductivity model can be better isolated. The Arrhenius model estimates for the basal reflection coefficient as a function of ice thickness are shown below (Fig. A) using the GISM temperature field.*

[Figure]

Fig. A. Arrhenius model estimate for basal reflection coefficient versus ice thickness for four different conductivity models. (a) W97 (gradient=-6.03 dB/km, $r^2$=0.29), (b) M07 (gradient =-0.29 dB/km, $r^2$=0.0009), (c) W97-corrected (1.7) (gradient=1.86 dB/km1, $r^2$=0.03), (d) W97-corrected (2.6) (gradient=12.02 dB/km, $r^2$=0.49). The negative and positive correlations in (a) and (d) are interpreted as significant underestimates/overestimates for the conductivity at the IPR frequency, whereas the near thickness-invariance in (b) and (c) are interpreted as good estimates.

*Assuming a linear trend, the (uncorrected) W97 model has a negative correlation between modelled reflection and ice thickness. This is consistent with the conclusion in Macgregor et al. (2015b) that it significantly underestimates the conductivity/attenuation rate. Both the M07 model and W97-corrected (correction parameter=1.7), are near to being thickness invariant. We conclude that either of these models serve as suitable estimate for the attenuation rate/conductivity model used in our algorithm. W97-corrected (correction parameter=2.6) has a strong positive correlation, from which we infer that it represents a significant overestimate of the attenuation rate. Repeat analysis for other regions of the GrIS and using the SICOPOLIS temperature field confirm these conclusions, with the M07 model closest to overall thickness invariance.*

**(ii) Revised Sect. 3.5 (Temperature bias of ice sheet models)**

*We have now used the W97-corrected (correction parameter=1.7) model alongside the M07 model, when evaluating temperature bias. This enables us to highlight the sensitivity of the evaluation of temperature bias to the choice of conductivity model.*

*Our central conclusion, that our radar-inferred values are consistent with the model temperature bias at DYE 3, also holds for the W97-corrected(1.7) model alongside the M07 model. In recognition that temperature bias evaluation may be the aspect of our study that is of widest interest, we also now include an attenuation rate bias plot for the entirety of our converged region (basins 3,4,5,6), rather than just SE Greenland.*

*Additionally, following the investigation in Macgregor et al. 2015b, we also discuss how the inclusion of layer stratigraphy in the Arrhenius model could alter the result.*

Sections 2.5 (including Figure 5) and the latter half of 2.6 were the newest and unsurprisingly the hardest concepts to grasp. I got the gist and am comfortable with the approach (the consideration of anisotropy in the horizontal gradient is an important advance), but I recommend additional review of the text in this section for accessibility.

*We agree that Sect. 2.5 and Sect. 2.6 represent the two central method developments proposed in the paper. We propose that the following changes will improve their accessibility:*

*(i)    We will make it clearer in Sect. 2.2 (overview of algorithm), that Sect, 2.5 and Sect. 2.6 represent the major original contributions in our paper. We will also state explicitly what problems they address (see also later comments about Figure 2).*

*(ii)    We will move the more technical material/equations in Sect. 2.5 to an appendix (Appendix B). In the main text we now focus upon the key conceptual steps that enable us to constrain the anisotropic sample region.*

*(iii)    We will highlight in Section 2.6 that equation (11) (which acts to standardise power for local attenuation variation within the sample region), is our major modification to the basic-bed returned power method. Additionally, we will make it clear that, if there is a thermomechanical model available, this correction could also be used in future regional studies (i.e. the method proposed in Sect. 2.6 could be applied in a regional study even if the sample region windowing methods in Sect. 2.5 are not applied).*

Section 2.8: The gridding method used is never discussed. Based on the distortions present in maps in Figure 6 onwards, I can reasonably guess that a bilinear or natural neighbor gridding was used. While the gridding method is not a critical element of this study, I strongly recommend that ordinary kriging be applied instead. As it stands, a sub-standard gridding perhaps unintentionally diminishes from the results presented. Regardless, clarify the method used.

*The gridding in our paper, is a simple cell average (1 km resolution) and this has now been stated explicitly in Sect. 2.8. This is done primarily for the convenience of plotting raster data.*

*Just to be clear, no gridding (other than the along-track averaging) is initially applied to the power measurements when performing the thickness versus power linear regression (this maximises the available data density). However, the `gridding' of the radar-inferred depth-averaged attenuation rate arises naturally from the scan resolution of the central point moving target window (set to 1 km). These points have also been emphasised in Sect. 2.8.*

13-4, 345-6, Section 3.5: These statements and this sub-section seem to imply that this is the first time that the temperature bias of a major ice-sheet model has been evaluated using radar-attenuation estimates, when in fact MacGregor et al. (2015b) did exactly that with the steady-state ISSM instance from Seroussi et al. (2013), albeit not for the entire ice column. Reword

*We have rewritten Sect. 3.5, to better acknowledge this aspect of the prior study, and how it provides much of the groundwork for our own study. Seroussi et al. (2013) has also been added to the references.*

Minor comments
21: The use of the term "noticeable" here is somewhat odd. "substantial" would be better. The improved coverage is the result of the efforts of innumerable individuals over many years and is intentional.

*Changed as suggested.*

40: Clarify whether linear or dB units are meant when referring to the range of variability of attenuation rate.

*'range' has been replaced with 'decibel range ( ~5-40 dB).*

How about defining and using Delta B_infty instead? What is _infty meant to represent, anyway? It seems to originate on 167. I assume it references the highfrequency limit, but that's somewhat obscure and won't be obvious to most readers. I recommend to stick with nomenclature from earlier studies.

*We suggest that the use of Delta B_infty here may potentially lead to more confusion as we also consider other `differences' in attenuation rate for the temperature field biases later in the paper.*

*We agree that the infinity notation is a bit obscure, and potentially confusing. In the context of our paper, the key point is that we are dealing with a prior/estimated quantity (not that the variable is a consequence of the high frequency limit). In order to distinguish the prior value from the radar inferred value with have now used a `^' notation for the prior/estimated value. This parallels the use of prior notation Bayesian statistics.*

(12): What is the value of [S]? If dS/dh is assumed to be negligible, then it is irrelevant to dR_infty dh, right?

*That is correct: dS/dh is considered negligible and (under this assumption) is therefore irrelevant to dR/d_infty. In order not to introduce unnecessary notation we have removed [S] from equations (8), (9) and (12), but mentioned explicitly that our method is `based upon the assumption that the variation in radar system performance with respect to ice thickness (within a field season) is considered negligible.'*

(14): Come up with a more meaningful subscript title than "_{ratio}".

*After much consideration we cannot think of a clearer label. We have reinforced that strong relative correlation in dP/dh with respect dR/dh (which follows as a requirement for an accurate/unbiased solution of the radar equation) is represented by this `correlation ratio'*

3. Results and discussion: It would be better to separate out results from discussion, as the current structure as labelled is uncommon. However, I find the text quite readable in this section so perhaps this is a non-issue.

*In an earlier version of the manuscript, we did originally separate the Results and Discussion sections. However, we decided that an ongoing discussion of the results was the best way of describing our new algorithm.*

503-4: Here I think "As was proposed by MacGregor et al. (2015b)," can be dropped, as the following statement is generally accepted regardless of that earlier study.

*Changed as suggested.*

555: I don't understand what the qualifier "final" adds here. It implies that the future products that will be generated by this method will not need to be improved, which is a bit strong.

*We agree. ` Final' has been removed.*

Appendix A: I don't mind this appendix but it closely hews to earlier studies. For brevity, it could be dropped.

*We have now edited the existing content, and extended the content of Appendix A to discuss the different conductivity models (see earlier our response).*

580: Better to cite MacGregor et al. (2012) instead of MacGregor et al. (2007) here as the former provided the correct form of the equation shown.

*Done.*

585: M not uM for chemical impurity units, following the dimensions given for the other parameters.

*We agree. For brevity, (and as we have now expanded Appendix A to focus upon the conductivity models), we now reference the summary table for the dielectric parameters in Macgregor et al. (2015b).*

Figures

For at least one Greenland-wide map, perhaps Figure 4c, it would be helpful to have a box representing the focused SE study area of Figure 9 onwards

*An outline map with box has been added to Fig. 9 where SE Greenland is introduced as our example region.*

Figure 2. Use a legend rather than the caption to explain the coloring. I find this diagram helpful but would like it to be expanded, although I'm not exactly sure how. Perhaps including related figure numbers (thanks for including section numbers) and, more ambiguously, the question/challenge each algorithm step addresses.

*Colour coded boxes have now been added along with references to figures. The accompanying text that explains the steps of the algorithm (Sect 2.2), has also been made explicit to show the questions that we are addressing.*

Figure 7: Show best-fit lines also.

*Good suggestion. This adds clarity to the argument about the systematic underestimation in attenuation rate that is predicted to occur without the correction.*

Figure 9: This figure's panel titles and legends are good examples of where the nomenclature needs some simplifying. Table 1 helps but is not quite enough.

*We have now used variable descriptions in words rather than symbols for the figure titles (and listed the symbols in the caption).*

Figures 10 and 11 could be combined into a 2x2 panel. Either way, identify the subregion for Figure 11 in Figure 10.

*We agree. Figure 10 and 11 have been combined, identifying the sub-region.*

Figure 11: Is the quantity shown in panel actually reflectivity, as is in more common usage? A grayscale or some other color illustrating ice velocity beneath the bed values would be nice (e.g., Figure 1 of Jacobel et al., 2010, The Cryosphere). Grammar, etc.

*Strictly, the quantity is the reflectivity based upon the depth-integrated power, as described in lines 258-259 of our manuscript, and referred to as P_adjusted in Oswald and Gogineni (2008). This quantity represents the sum of specular and diffuse reflection/scattering components, and is therefore predicted to have a narrower dB range for reflectivity than the peak power definition. In order to make this explicit we have changed: `basal reflection coefficient' to `basal reflection coefficient using depth-integrated power' in the Fig. 11 caption. We have also added an extra sentence to Section 3.3 describing that the integrated measure is predicted to reduce the overall dB range.*

*We have now underlain the reflectivity map with balance velocity. We observe regions of faster flow where basal reflection values are higher.*

Regularly and particularly in the introduction, MacGregor et al. (2015a) is cited where I believe MacGregor et al. (2015b) is meant to be cited.

*Apologies; these citations have now been corrected.*

2: ice sheet 6: englacial 34: subglacial 90: missing parentheses around citations 55: tolerance 281: a hypothetical ice column 441: 63% 544: is present 547: a 'true' Table 1 title: principal

*All suggested changes have been made. 'ice-sheet' is now change to 'ice sheet' throughout the article.*

---

## Author Comment (AC2) · 4 Apr 2016

**Author comments:** "An ice-sheet wide framework for englacial attenuation and basal reflection from ice penetrating radar data"

T.M. Jordan, et al., *The Cryosphere*

Review by: Mike Wolovick

*We thank the reviewer for their exceptionally detailed feedback to our manuscript. They have well understood the methods developed, and have given many helpful suggestions regarding how we can improve the overall clarity of our presentation. We provide detailed feedback to their comments in blue italicised text. The summaries of our revisions which address the major comments are highlighted in bold text.*

**Summary:**

This paper describes a semi-empirical method for estimating attenuation losses and bed reflectivity in radar data from continental ice sheets. The method uses a prior thermal model to estimate the spatial gradient in attenuation. Based on this spatial gradient, the method selects local regions that are expected to have broadly similar attenuation rates based on a segmentation approximation. Within each region, the thermal model is used to correct the observed bed-returned power for the local differences in attenuation relative to the mean for that region. The corrected bed- return power data are then fit with a least-squares linear best-fit representing the mean attenuation rate for that region. The residual to the fit represents the basal reflectivity.

The authors apply the method to the CReSIS radar dataset collected over the Greenland Ice Sheet for Operation IceBridge. They find that the method converges where ice thickness is variable and attenuation rates are high in the south and east of Greenland. In the north and west, as well as in the ice sheet interior, the method does not converge. They find a tradeoff between spatial coverage and precision, such that the area where the method is considered to have converged increases if one is willing to accept higher uncertainty. The output bed reflectivity estimates show a reduced spread consistent with the range of plausible subglacial materials. The authors also demonstrate that the estimated attenuation rates can be used to check the temperature bias in the original input models.

This manuscript is clearly relevant for *The Cryosphere*. The method developed by the authors represents an important advance in the integration of ice-penetrating radar data with ice sheet models. The authors demonstrate both how models can be used to guide the interpretation of radar data, and how the radar data can in turn be used to diagnose biases in those models. I have two major concerns, one relating to the inability of the model to converge in the ice sheet interior, and the other relating to the segmentation approximation. The first concern is actually an opportunity for the authors to explain the scope of their results. I believe that they have been too conservative, and that in fact they can constrain reflectivity in the interior of Greenland even if they cannot also constrain attenuation. The second concern could be addressed by showing how the results respond to alternate means of choosing local sample regions.

**Major Comments:**

Ice Sheet Interior

The main weakness of the method developed by the authors is that it does not work in interior regions of Greenland where variations in ice thickness and attenuation rate are both low. In other words, the method doesn't work where the problem is easy! With little variability in ice thickness and relatively constant thermal

structure, one could get good estimates of the basal reflectivity anomaly in interior Greenland by using no attenuation correction at all.

The reason why the authors' method does not converge in the interior of Greenland is one that the authors themselves identified: the variability in ice thickness is too low. Quantitatively, we can say that in order to get a good correlation between bed returned power and ice thickness, the following condition must hold: $2B\Delta h > \Delta R$, where B is the regional average attenuation rate, $\Delta h$ is the standard deviation of ice thickness, and $\Delta R$ is the standard deviation of bed reflectivity. When the variability in ice thickness is below this threshold, a local linear best-fit cannot constrain attenuation rate.

However, when the variability in ice thickness is low, the total attenuation losses should also be roughly constant. Attenuation therefore becomes less important, and it should be possible to estimate basal reflectivity anomalies even if attenuation cannot be constrained. Local variations in attenuation rate may still produce variability in total attenuation losses; however, the authors are already using a numerical model to estimate local variations in attenuation rate. The authors make a good case that the local gradient in attenuation rate from the models is more reliable than the mean value, so they can simply continue using the model to correct for local variations in attenuation rate. When the variability in ice thickness is low, the authors' method will be unable to constrain the regional mean attenuation rate, but it should still do a good job of estimating basal reflectivity, and the authors could present those reflectivity results in the interior of Greenland.

In order to capture regions where it is possible to constrain reflectivity but not attenuation, the authors could introduce an alternate quality control check as a substitute for the $r^2[Pc]$ check they introduce in Section 2.7. When the variability in ice thickness is low compared to the variability in basal reflectivity ($2B\Delta h < \Delta R$), the authors could check the standard deviation of bed reflectivity instead of $r^2[Pc]$. The authors use the standard deviation of reflectivity as a check on the validity of their results anyway (Section 3.2), so it makes sense to formally add this metric to the quality control step. If the standard deviation of bed reflectivity is reasonable for subglacial materials, then the alternate quality control is passed and the authors can present results for reflectivity in that region.

*The reviewer has made the case that it is possible to constrain relative reflection (reflection anomalies) in regions where it is not possible to constrain attenuation. They have correctly implied that our quality control measures, described in Sect. 2.7, are designed with attenuation solution accuracy in mind, and therefore fail to identify some regions where relative reflection can be constrained (primarily the northern interior). Additionally, they have suggested that we could reformulate our algorithm using the standard deviation of relative reflection as a quality control measure.*

*We appreciate that we should have been more explicit about the findings of previous studies, Oswald and Goginenni (2008, 2012), that have already addressed exactly the problem that the reviewer raises: constraining relative reflection in the interior of Greenland. Specifically, Oswald and Goginenni (2008, 2012) demonstrated that, assuming a simple (approximately constant) attenuation rate model, relative reflection values for the interior have a decibel range that is near-invariant with ice thickness, with an approximate bimodal distribution that they associate with wet and dry beds.*

*As described in the conclusions of our paper, in future work we aim to use a similar approach when producing a gridded reflection data product for the interior, (however probably in conjunction with a forward Arrhenius attenuation model, rather than the specific attenuation model in Oswald and Goginenni (2008, 2012)). We therefore believe that to focus to on a reformulation of the problem (i.e. `quality control in terms of the distribution of relative reflection') would detract from the important step forward that we have made in our paper: that we have developed a robust, automated, method of inferring full ice column attenuation values toward the margins of*

*ice-sheets (i.e. where the assumption made regarding the attenuation model in Oswald and Goginenni (2008, 2012) breaks down, and where their method cannot be applied). Additionally, whilst our study deals with relative reflection, an ongoing goal is to incorporate radar system performance ([S] in equation (8)), and therefore constrain absolute basal reflection rather than basal reflection anomalies. This approach would require englacial attenuation to be known (subject to an estimated uncertainty bound), which is consistent with the approach we have taken in our paper.*

*We envisage the following problems using the standard deviation of reflection as an automated control measure:*

*(i) It makes the underlying assumption that the distribution for basal reflection is unimodal. Whilst this appears to be is the case for the coverage region in SE Greenland (Fig. 11), it is not required as an a priori assumption of our existing method. As mentioned above, Oswald and Goginenni (2008, 2012) demonstrated that the distribution of basal reflection values for the interior of Greenland is approximately bimodal.*

*(ii) Automatically selecting regions on based upon a minimising the spread of the distribution in basal reflection values is not necessarily desirable. Subject to this control measure, the algorithm would preferentially select regions that are homogenous, and fail to select sharp transition regions.*

***In summary, we thank the reviewer for adding some true clarity to the problem which we address and we suggest the following changes to the manuscript:***

(i)   ***We will make it clear in the introduction that Oswald and Goginenni (2008, 2012) concluded that relative reflection/reflection anomalies can be constrained in the interior of Greenland where attenuation rate variation is low. However, due to both the higher spatial variation and higher absolute values in attenuation rate (as predicted by Arrhenius models), the same is not true toward the margins.***

(ii)   ***We will state explicitly that the algorithm quality control measures, equations (13) and (14), are specifically designed with attenuation rate/loss accuracy in mind, (rather than constraining the distribution of relative reflection). Given the valuable second use of radar attenuation to constrain temperature, we believe that this scientific problem requires a full investigation as outlined in our paper.***

(iii)   ***In view of point (ii) we will revise the title of the manuscript title to 'An ice-sheet wide framework for englaical attenuation from ice penetrating radar data'. The introduction/abstract will now better focus on the `dual role' for an IPR-derived attenuation solution (i.e. constraining basal reflection and temperature).***

(iv)   ***Finally, as part of our feedback to the other reviewer's comments, we will present a reflection map for the interior of Greenland using a forward Arrhenius model. This backs up the conclusion in Oswald and Goginenni (2008, 2012) that when attenuation variation is low, the reflection distribution is well constrained and near thickness-invariant.***

**Segmentation Approximation**

The segmentation approximation seems needlessly complex.  The purpose of the segmentation approximation (Section 2.5) is to define a local region in which attenuation rate is roughly constant. Why not simply define an oval-shaped region where the RMS variability in attenuation rate is less than some threshold?  Or why not simply define an irregular contiguous region containing all grid cells where the difference in attenuation rate is less than the threshold? With an ellipse, the unknowns at this step of the problem would be reduced to three: the orientation of the elipse, the length of the major axis, and the length of the minor axis.  With an irregular

shape, no a priori assumptions about the nature of the ice sheet temperature field need to be made at all. Using an ellipse instead of the segmentation approximation would eliminate the sharp corners created along the segment boundaries in Figure 5f.

In addition, an ellipse or an irregular shape would drastically simplify Section 2.5. Nothing that the authors have presented indicates that the segmentation approximation is a particularly good representation of the ice sheet thermal structure. The segmentation approximation has no physical basis in ice dynamics or temperature that could justify the use of such a complex model. The only virtue of the segmentation approximation seems to be that it is capable of elongating perpendicular to the gradient of attenuation rate, but an ellipse or an irregular shape could do that too. In addition, the segmentation approximation is only capable of elongating at 45° angles, but an ellipse or irregular shape could elongate at any angle.

I do not believe that the awkwardness of the segmentation approximation invalidates the later results of this paper. It is likely that the authors would have achieved similar results with any reasonable method for selecting a local sample region based on the input thermal models. However, the unnecessary complexity of the segmentation approximation makes the paper harder to follow, has no realistic basis in ice sheet physics, and potentially

The supplemental material (Sections S1 and S2) explores the sensitivity of the sample regions produced by the segmentation approximation to the temperature model input (S1) and to the choice of RMS tolerance (S2). However, neither section addresses the sensitivity of later results to the segmentation approximation itself. I would like to see an exploration of how the results are affected by completely different means of choosing a local sample region. What happens if the segment boundaries are shifted by 22.5° (half a segment)? What happens if six segments are used instead of eight? What if an elliptical, a circular, or an irregular sample region is used? The authors need not address every single possible method of determining local sample regions, but I would like to see some exploration of the effects of using a segmentation approximation to choose local sample regions.

*The reason why the segmentation approximation uses the `compass directions' to define anisotropy is analogous to why finite difference methods for differential operators do: it is the most computationally practical method for a gridded data structure. Conceptually, the `local difference' terms, $ - $, are analogous to the numerator of a finite difference derivative. (As an aside, we use local differences rather than the finite derivatives, due the derivative being much nosier and having sharp discontinuities present. Additionally, our use of `8 compass directions' captures greater information than the `4 compass directions' that would occur when using a standard horizontal gradient operator.) Ultimately, the geometric parameters of either and an oval or ellipse would also be conditioned by the horizontal gradient/difference terms, and therefore would also be subject to similar limitations (i.e. being conditioned by 4 or 8 axes) and have similar artifacts present.*

*What we perhaps did not make as clear as we should have done in the manuscript, is how the `complexity' of the method arises due the more obvious approaches (such as those suggested by the reviewer and experimented with by ourselves during the method development) having practical difficulty in their implementation. Below we deal with specific points made by the reviewer.*

Re: Why not simply define an oval-shaped region where the RMS variability in attenuation rate is less than some threshold?

*As mentioned in the manuscript we did originally experiment with a more simple RMS measure of window tolerance for the segments line 220). We concluded that this approach produces sharp discontinuities in the spatial dependence of the target window dimensions (i.e. the `window radi' in Fig. 6), and therefore very sharp*

*discontinuities in radar-inferred attenuation rate/IPR data that is sampled. Hence our use of the integrated RMS tolerance measure*

Re: why not simply define an irregular contiguous region containing all grid cells where the difference in attenuation rate is less than the threshold?

*Whilst this is a simple question to ask, this is a substantially more computationally demanding approach than our current method. At our chosen resolution each grid cell (~10^6 in total) would have an associated `sample region mask' to be defined (~10^4-10^5 cells in total). Additionally, a specific algorithm would have to be developed to define the contiguous region.  We do, however, agree that this approach provides the best estimate of the thermal/attenuation structure of the GrIS and this has been added to Section 2.5.*

Re*: What happens if the segment boundaries are shifted by 22.5º (half a segment)?  What happens if six segments are used instead of eight?*

*Again, these suggestions are significantly more complex to implement than our existing method for a local difference measure on a rectangular grid.*

Re*: What if an elliptical, a circular, or an irregular sample region is used?*

*See above comments regarding elliptical and irregular shape regions. We originally investigated circular (isotropic) regions, where the radius of the circle is a function of the magnitude of the horizontal gradient in the depth-averaged attenuation rate. However, all other things being equal, this resulted in more pronounced systematic biases for cross-over measurements of accuracy (different temperature fields and field seasons).*

***In summary, given that the reviewer does not believe that `the segmentation invalidates the later results of the paper' we suggest that it not truly a major concern/comment, and that their suggestions for how one could potentially reformulate procedure are best placed in the context of future modifications. We suggest the following revisions:***

> ***(i) Clearly stating the segmentation approximation is just one possible representation of the anisotropy of the estimated $$ field,  list the other possibilities that could be considered, and state that an irregular contiguous region is the most desirable, but computationally expensive, approach.***

> ***(ii) A rewrite of Section 2.5 where the conceptual arguments are made more explicit, whilst moving the more technical details/equations regarding the segmentation approximation to a new appendix (Appendix B). This is in correspondence with our response to the other reviewer.***

> ***(ii) State that the `8 compass directions' used in the segment approach, arises through analogy with numerical schemes for finite difference derivatives, (and that the local differences that we use are much smoother and more tractable than simple application of a finite derivative operator.)***

**Minor Comments:**

Line 4:  "...which is an exponential function of temperature."

Attenuation is an Arrhenius function of temperature, not an exponential function.

*We have changed `exponential function of temperature' to `Arrhenius function of temperature'.*

Line 5 (and elsewhere):  "stationarity"

I'm not sure I agree with the authors' use of the term "stationary". Typically, "stationarity" refers to a time series whose statistical properties are constant over time, and that concept could be generalized to spatial data whose statistical properties are constant over space. However, the authors use "stationarity in the attenuation rate" to mean a constant attenuation rate. A constant is stationary, but not all stationary data are constant. In this case, the authors could be both easier to understand and more accurate by saying "constant" when they mean constant.

*The reviewer is correct and, with regards to the attenuation rate field, we exclusively use `stationary' to mean `constant'. This a very sensible suggestion we have changed all usage of stationary to constant.*

Lines 16-31: Several places in this paragraph could benefit from adding additional (often older)

references.

ice thickness: add [*Bailey et al.*, 1964; *Evans and Robin*, 1966; *Robin et al.*, 1969; *Jankowski and*

*Drewry*, 1981]

basal material properties: add [*Oswald and Robin*, 1973; *Peters et al.*, 2005]

internal layer structure: add [*Robin et al.*, 1969; *Conway et al.*, 1999, 2002; *Vaughan et al.*, 1999;

*Fahnestock et al.*, 2001; *Dahl-Jensen et al.*, 2003; *Ng and Conway*, 2004; *Tikku e* new data
products for bed elevation and ice thickness: add [*Morlighem et al.*, 2014] Additional uses of
radar data that could be added to this paragraph:

ice rheology: [*Raymond*, 1983; *Hindmarsh et al.*, 2011; *Kingslake et al.*, 2014]

grounding line dynamics: [*Conway et al.*, 1999; *Catania et al.*, 2006; *Christianson et al.*, 2013]

basal melting or freezing: [*Fahnestock et al.*, 2001; *Catania et al.*, 2010; *Bell et al.*, 2011]

ice dynamic changes: [*Conway et al.*, 2002; *Bingham et al.*, 2015]

*We deliberately added 'e.g.' when listing references in this introductory section, to represent that our reference list was non-exhaustive. However, as a major motivation for our work is to develop new data products, we have added Morlighem et al. (2014). Additionally, as basal melting or freezing is potentially very relevant to our work on basal reflection we have added [Fahnestock et al., 2001; Catania et al., 2010; Bell et al., 2011].*

Line 33:

 add [*Oswald and Robin*, 1973]

*Done.*

Line 38: "Arrhenius models where the attenuation rate is an exponential function of **inverse** temperature"

*Done.*

Line 45: "...make the implicit assumption..."

In some papers, the assumption is explicit. Maybe just say "...make the assumption..."

*Done.*

Line 45: "locally stationary"

See my comment above. "Locally stationary" is an oxymoron; stationarity implies that statistical properties are globally constant. Use "locally constant" here instead.

*Done.*

Line 53: "A central feature of our algorithm is the use of a prior Arrhenius model estimation of the attenuation rate as an initial condition."

The use of the phrase "initial condition" is incorrect here. Initial conditions apply to models that predict the evolution of some variable over time. A better term would be "first guess", "initial guess", or "initial estimate".

*We agree, the term `initial condition' is normally used in the context of a dynamical system and we do not want to confuse the reader here. We have replaced 'initial condition' with `initial estimate'.*

Lines 54-55: "Conceptually, the initial condition is used to estimate regions where the assumption of stationarity is valid within some specified tolerance."

Removing "stationarity" can clean up this sentence: "The initial estimate is used to determine regions where attenuation rate is approximately constant to within some specified tolerance."

*Done – good suggestion.*

Line 158: "...exponential dependence upon **inverse** temperature

*Done*

Line 160, 164 (and possibly elsewhere):

Be careful about MacGregor et al., [2015a] versus MacGregor et al., [2015b]. I'm pretty sure you mean the second one in this context.

*Done. This mistake was also raised by the other reviewer.*

Lines 165-172:

Somewhere in here would be a good place to indicate that brackets <X> indicate the depth-averaged value of X.

*Done. We agree that it is very important to make this clear*

Lines 173-181:

For completeness, it would be a good idea to state how big of an effect you expect to see from climate transitions in Greenland. The MacGregor et al., [2012] reference refers to East Antarctica, where climate transitions have both a smaller signal in ice chemistry and are more closely spaced in depth than in Greenland. In Greenland, the stratigraphic chemistry changes are dominated by the Holocene- LGM transition, which occurs at wildly different depths in southern and northern Greenland [*MacGregor et al.*, 2015]. The difference

in depth of the Holocene-LGM transition with respect to the warm ice near the bed might be expected to produce a large difference in attenuation rate between northern Greenland and southern Greenland.

*This is an interesting point. We have now incorporated this discussion into Sect. 3.5 where the Arrhenius model is used to determine temperature bias. We specifically note that: (i) the depth-averaging approximation using GRIP core values generally underestimates attenuation loss relative to using layer stratigraphy, (ii) this underestimation is greater in South and West Greenland where there is a greater proportion of Holocene ice.*

Lines 190-191:

Simplify this sentence to: "For the majority of the IPR data coverage region, GISM has lower temperature and therefore lower attenuation rate than SICOPOLIS (Fig. 4c).

*Done – we agree this is clearer.*

Line 215

There is an extra parenthesis inside the square root sign.

*Done.*

Lines 216 and 217

The averaging brackets $<>$ are in the wrong place. As written, the square root cancels the square and the expression reduces to the absolute value of the difference. The brackets should go outside of the squared difference. The expression inside the square root sign should be: $<(B\infty(x,y)-B\infty(x0,y0))^2>$ (and likewise for the second expression). The alternate possibility is that the averaging brackets represent column-averages, not real averages. In that case, the expressions reduce to the absolute value of the difference.

*The second interpretation is correct and the averaging brackets represent column/depth averages, and they are therefore **not** in the wrong place. This should be clear, since throughout the paper $<>$ corresponds to column/depth rather than statistical averages.*

*Yes we agree, the expression does reduce to an absolute difference. However, due to the later development of our integrated tolerance measure (where the square root is taken outside of the integral), we prefer the equivalent squared/square root notation rather than the modulus.*

Lines 208-217

This paragraph is very confusing. It sounds like the authors assume only radial dependence within each segment, but later, the sample region boundaries (Figure 5f) clearly show a dependence on angle even within each segment. This is because the region boundaries are interpolated from the central radius vector for each segment along a circular arc, in order to produce continuous (but not differentiable) window boundaries. It would be helpful to state somewhere in this paragraph that the ultimate goal is to produce a variable radial length of the target window by interpolating with respect to angle.

*See our response to the major comment on the segmentation approximation*

Equations 6 and 7:

Move the constant terms outside of the integral and simplify them to $2/R^2_n$.

*Agreed. This is better practice.*

As written, the RMS is also a function of $\theta_n$.

*We agree that the RMS measure is also a function of theta. However, theta is fixed for each integral, where Rn is the `target variable' to be solved for.*

Lines 237-239:

See my major comment above. If the gradient of ice thickness with distance is small, it should be easy to estimate bed reflectivity, because the mean attenuation rate has little effect on total attenuation. Variations in total attenuation only arise from variations in the attenuation rate, and the method the authors develop relies on an a priori model to correct for local variations in depth-averaged attenuation anyway.

*See our earlier response to the reviewer's first major comment.*

Lines 259-261:

Most readers probably know this already, but it still might be helpful to explicitly state that crevasse scattering is most likely to cause problems for radar analysis of basal conditions in fast-flowing regions near the ice sheet margin.

*Good suggestion. We have added this point.*

Lines 279-282:

Simplify this statement. This step corrects for the difference in attenuation rate between the measurement point and the central point.

*We respectfully disagree with the reviewer on this point as line 279 describes exactly what equation (11) represents.*

Lines 283-287:

Why are thinner ice columns warmer, and thicker ice columns colder? On the one hand, conductive cooling should tend to produce warmer conditions in thicker ice. Most people reading this paper would probably assume that thick=warm and thin=cold. On the other hand, the Peclet number ($Pe=ah/\kappa$) indicates that thicker ice columns should have a greater dominance of advection over diffusion, and therefore the cooling effect of surface accumulation should be greater in thick ice. In addition, ice must flow faster (with a higher driving stress) in thin regions, producing more shear heating. Finally, low surface elevations tend to have a higher surface temperature because of the atmospheric lapse rate, but this effect should not influence local temperature differences due to bed topography. Which one of these mechanisms is responsible for producing the thick=cold, thin=warm association?

*As discussed in Section 3.5 of our paper, and Macgregor et al. (2015b), the depth-averaged attenuation rate and the depth-averaged temperature are proxy variables for each other, and it is in this sense we use the terms*

*`warm' and `cold'. An estimate for the spatial variation in the depth-averaged attenuation rate over the Greenland ice sheet is shown in Fig. 4(b). It is clear that, as a first approximation, the depth-averaged attenuation rate is proportional to ice thickness (e.g. Bamber et al. (2013), Fig. 3), and it is lower in the interior of the ice sheet where the ice is thickest. This suggests that surface temperature (and its dependence upon elevation), is the dominant `mechanism' that governs the spatial distribution of depth-averaged attenuation rate. This supports our general `thick=cold, thin=warm' association. Finally, it is clear that this association holds over the spatial scale of our sample regions, (refer to Fig. 6 for the window vector plot).*

State the two criteria in words at the beginning of this section. Something like, "As a quality control check, we are looking for regions where (1) the correlation between ice thickness and bed-returned power is good, and (2) the correlation between ice thickness and bed reflectivity is poor."

*This is a sensible suggestion that improves the clarity of the section. We would, however, argue that point (2) should be stated as `the correlation between ice thickness and bed reflectivity is poor relative to the correlation between ice thickness and bed-returned power'. As an aside, we did initially experiment with using the thickness correlation for [R∞] (the Arrhenius model estimate of the relative basal reflection coefficient), as a quality control measure, but we found the correlation ratio (14) to be more robust (in the sense that we have greater coverage for given solution accuracy).*

Lines 349-350:

Why were the field seasons processed independently, if Lines 81-83 stated that power measurements from different field seasons could be combined?

*This was done as we wanted to test if our attenuation algorithm was repeatable for different field campaign data/flight tracks (as described in lines 453-456) and the Supplementary Material. We have now stated this explicitly.*

Section 3.2:

State the definition of "convergence" up front. The reader has to wait until the last paragraph (Line 415) to learn that convergence means "a normally distributed difference centered on zero".

*We have changed `convergence' to `convergence (defined here as a normally distributed difference centered on zero)'.*

Lines 413-421:

The first sentence of this paragraph should state that the algorithm converges in southern and eastern Greenland, but not northern or western Greenland. The reader should not have to flip back and forth to the figures to determine where the basin numbers are located spatially.

*Done.*

Line 437: "A gridded map of the basal reflection coefficient...is shown in Fig. 11a." Figure 11a does not look like a "gridded map". I realize that technically it is gridded at 1km cell size, but for all intents and purposes Figure 11a shows reflection coefficients along-track.

*We have replace gridded map with `map for relative reflection along flight tracks'.*

Lines 504-508:

It would be appropriate to mention here that the relationship between attenuation rate and temperature is highly nonlinear, so the difference in depth-averaged attenuation rate does not transfer neatly to a difference in depth-averaged temperature. $\text{mean}(x^2) \neq \text{mean}(x)^2$.

*We agree, the complicating effects of non-linearity should be emphasised here. We have replaced `non-unique' with `non-linear and therefore non-unique'*

Section 4: Conclusions

This section should have a paragraph commenting on and interpreting the reflectivity results. From Figure 11, it appears that high reflectivity is concentrated in the approach to fast-flowing outlet glaciers. This is consistent with distributed hydrological networks or with saturated subglacial till, either of which would promote faster sliding.

*This suggestion was also made by our other reviewer and we have expanded Fig. 10 and Fig. 11 and Sect 3.3 to have more geophysical interpretation. In particular, we now compare the reflection map to a velocity map.*

Line 542-543: "We suggest that the converged radar algorithm attenuation solution is preferable to using a forward Arrhenius temperature model to calculate basal reflection coefficients."

Strengthen and clarify this conclusion: "We find that our data-based attenuation algorithm is superior to an attenuation correction calculated purely from an a priori temperature model."

*We agree; this is a conclusion that we draw. Our evidence for this is Section 3.2, where we show that the converged radar-inferred solution significantly reduces the thickness correlated bias for the depth-averaegd attenuation rate. However, as our algorithm has incomplete coverage (whereas the Arrhenius model solution has complete coverage), we have now restated the conclusion as:*

*'The converged radar algorithm attenuation solution provides a means of assessing the bias of forward Arrhenius temperature models. Where temperature fields are poorly constrained, and where the algorithm has good coverage, we suggest that it is preferable to using a prior Arrhenius model. This is due…'*

Lines 546-548: "Notably, we demonstrated that even a small constant bias in the attenuation rate across a region; (this could be either with respect to a "true" value or another modelled value), leads to a thickness correlated bias in attenuation loss and therefore the basal reflection coefficients."

This sentence is awkward. Rephrase as: "We demonstrated that even a small regional bias in attenuation rate leads to thickness-correlated errors in attenuation losses and therefore the basal reflection coefficients. These thickness-correlated errors persist regardless of whether the regional bias is with respect to the 'true' value or to another modelled value."

*Done – we thank the reviewer for making this point clearer.*

Lines 562-564:

Is interpolation of bed reflectivity onto a regular grid even desirable, given that subglacial hydrology and geomorphology are likely to vary at scales much smaller than the grid spacing?

*We agree that bed reflectivity will have sub-grid variability. However, the same is also true for ice thickness, and gridded basal topography data products are of widespread utility for ice-sheet modelling. Our hope is that a `coarse grained' bed-reflectivity data product (and the relationship to basal traction/basal sliding) could help to define the lower-boundary condition for ice-sheet models (see line 29 in the introduction).*

Lines 566-569: "Due to this lower spatial variability, (and despite the caveats in the paragraph above), these regions [ice sheet interiors] could potentially have their basal reflection values derived by using forward Arrhenius temperature model for the attenuation."

See my major comments above. When ice thickness has little variability, errors in the regional mean attenuation rate have little effect. Only the spatial gradients in attenuation rate matter, and as the authors point out earlier in the conclusion, the models do a better job representing these than they do at representing the mean value. The authors should have been able to take advantage of this fact to produce reflectivity estimates in the ice sheet interior.

*See major comments and the results of a forward Arrhenius model in the revised Appendix A.*

**Figures:**

In general, the figures need better subplot titles and labeling. Symbols without words are inappropriate for subplot or axes labels because symbols are hard for readers to understand without flipping back and forth to the places where those symbols are defined. The subplot titles should express their meaning in words, and the corresponding symbols can be given in the caption if necessary. Many of the figures also need to be larger to permit more detail and wordier labels. Units should be placed on colorbar labels, not in subplot titles.

For most of the figures, I've given my suggestions for more descriptive titles and labels. The authors need not follow these specific suggestions, but all of the subplot titles should use descriptive words rather than symbols.

*We again thank the reviewer for their detailed feedback, and have made the majority of the suggested changes. In particular:*

- *Moving units above the color bars*
- *Using more informative labeling.*

Figure1

Subplot titles:

a) Flight Tracks

b) Drainage Basins

Put the numbers for the drainage basins in (b) arrayed around the coast of Greenland, rather than all together in the key. That way it is easier to tell at a glance which number refers to which basin. Also, it might be a good idea to circle or otherwise highlight the four basins in which the algorithm converges.

Figure 3

Subplot titles:

Y-axis label: What does "linear units" mean, other than "not decibels"? Either convert to actual units of power (W or $Wm^{-2}$), express as a fraction of the transmit power, or normalize so that the peak in each plot is 1. Normalization may be the best option, so that the quality control check (decays to 2% of peak power) can be easily visualized.

State where the two examples were taken from in the caption.

*Units have been changed to `relative received power W' (which follows the description in the CreSIS L1B data product.*

Figure 4

Subplot titles:

a) Arrhenius Model for Attenuation Rate
b) Attenuation Rate from GISM
c) Difference between GISM and SICOPOLIS

The y-axis label in plot (a) should say the words "Attenuation Rate". The colorbars should be labeled with their units (dB/km).

It might be appropriate to include a map for SICOPOLIS itself, in addition to the difference map.

Figure 5

subplots:
a) Model Estimated Attenuation Rate
b) Segments
c) Segment Approximation of Model Estimate
d) Difference from Central Value
e) Segment Approximation of Difference
f) Window Boundaries
The units should be given next to the colorbars, not in the subplot titles. The colorbars should have a larger font size as well.

The units should be given next to the colorbars, not in the subplot titles. The colorbars should have a larger font size as well.

Don't the square root and square cancel in plot (d)? Isn't that plot just showing the absolute value of the difference? The same comment that I made about lines 216 and 217 applies to the expressions in the caption. Either the averaging brackets are in the wrong place, or the expressions reduce to the absolute value of the difference.

*See previous comment regarding lines 216/217*

Figure 6

Subplots:

a) Vector R1

b) Vector R2
c) Vector R3
d) Vector R4

Colorbar label should be "Length (km)"

Figure7

Plot title: "Attenuation Difference Correction"

It is more accurate to refer to the process shown in this figure as the attenuation difference correction, rather than the attenuation correction. The step only corrects for the difference in attenuation rate between the data location and the central point.

Figure 8

Subplots:

a) Correlation Between Power and Ice Thickness
b) Correlation Between Reflectivity and Ice Thickness
c) Correlation Ratio
d) Coverage
e) Attenuation Rate
f) Attenuation Loss
Put units (dB, dB/km) in the colorbar labels.
Put plots a-c on the same color scale.

Note in caption whether high values or low values are good in a-c. In (a) and (c), high values indicate the algorithm converged, but in (b) low values indicate convergence.

*Changed as suggested.*

Figure 9

Subplots:

a) Difference between Model Inputs

b) Difference between Algorithm Outputs

c) Attenuation Rate Difference Distribution

d) Attenuation Rate Difference vs Ice Thickness

e) Attenuation Loss Difference Distribution

f) Attenuation Loss Difference vs Ice Thickness

Add units to the colorbar.

Label important outlet glaciers in either (a) or (b). Helheim Glacier is in view here, for example.

*See revised section 3.3 and Figure 10 and 11 in the other reviewers' comments.*

Note in the caption what the reader should be looking for in terms of convergence: in plots (c) and (e), a normally distribution about zero indicates convergence, while in (d) and (f), a lack of systematic ice thickness dependence indicates convergence.

*Done.*

Figure10

Plot title: Attenuation Rate

Label colorbar with the units.

As in Figure 9, label important outlet glaciers, including Helheim. Also put an inset box showing the area of detail in Figure 11.

*We have labeled Helheim and Apuseeq glaciers (corresponding to the inset region for the reflectivity map).*

Figure11

Subplots:

a) fine as is

b) Reflectivity Distribution

c) Reflectivity vs Ice Thickness

Put units on the colorbar label. Label the outlet glacier(s) in the lower right of (a). Note in the caption that a range of approximately 20 dB in (b) is right for plausible subglacial materials, and that a lack of systematic ice thickness dependence in (c) indicates algorithm convergence.

Figure 12

Subplots:

a) Attenuation Rate Difference Distribution b) Attenuation Loss Difference Distribution c) fine as is.

Note in the caption that green is a subset of red, which is a subset of blue.

*Done.*

Figure 13

Subplots:

a) Difference Between Prior Model and Radar Estimate (GISM)

b) Difference Between Prior Model and Radar Estimate (SICOPOLIS)

c) Comparison Between Models and Dye3 Ice Core

Add units to the colorbar labels.

*This figure has now been substantially revised to incorporate two conductivity modelsfollowing the other reviewer's comments.*

**Supplemental Material:**

Line 33: "...the sample regions will contain individual ice columns..." Replace "individual ice columns" with "grid cells".

*Ice column is our preferred term, as the linear regression procedure applies to each (along-track averaged) measurement*

Lines 43-45: "If this is rescaled by ice thickness for a sample region in the interior of the ice sheet (mean ice thickness ~2500m) this results in a desired attenuation rate accuracy ~1 dBkm$^{-1}$." This explanation is very simple and should go in the main text.

*Agreed. This has been moved to main text.*

Lines 69-70:

Mention that basins 3, 4, 5, and 6 are all located in south and east Greenland.

*Done.*

Figures:

Same comments as for the figures in the main text. All of these figures need better subplot titles and units labelled on the colorbars.

*See our response to the main article figures.*

**References**

Bailey, J. T., S. Evans, and G. de Q. Robin (1964), Radio echo sounding of polar ice sheets, *Nature*, *204*(4957), 420–421, doi:10.1038/204420a0.

Bell, R. E. et al. (2011), Widespread persistent thickening of the East Antarctic Ice Sheet by freezing from the base, *Science*, *331*(6024), 1592–1595, doi:10.1126/science.1200109.

Bingham, R. G., D. M. Rippin, N. B. Karlsson, H. F. J. Corr, F. Ferraccioli, T. A. Jordan, A. M. Le Brocq, K. C. Rose, N. Ross, and M. J. Siegert (2015), Ice-flow structure and ice dynamic changes in the Weddell Sea sector of West Antarctica from radar-imaged internal layering, *J. Geophys. Res. Earth Surf.*, *120*(4), 655–670, doi:10.1002/2014JF003291.

Catania, G., C. Hulbe, and H. Conway (2010), Grounding-line basal melt rates determined using radar- derived internal stratigraphy, *J. Glaciol.*, *56*(197), 545–554, doi:10.3189/002214310792447842.

Catania, G. A., H. Conway, C. F. Raymond, and T. A. Scambos (2006), Evidence for floatation or near floatation in the mouth of Kamb Ice Stream, West Antarctica, prior to stagnation, *J. Geophys. Res. Earth Surf.*, *111*(F1), F01005, doi:10.1029/2005JF000355.

Christianson, K., B. R. Parizek, R. B. Alley, H. J. Horgan, R. W. Jacobel, S. Anandakrishnan, B. A. Keisling, B. D. Craig, and A. Muto (2013), Ice sheet grounding zone stabilization due to till compaction, *Geophys. Res. Lett.*, *40*(20), 5406–5411, doi:10.1002/2013GL057447.

Conway, H., B. L. Hall, G. H. Denton, A. M. Gades, and E. D. Waddington (1999), Past and future grounding-line retreat of the West Antarctic Ice Sheet, *Science*, *286*(5438), 280–283, doi:10.1126/science.286.5438.280.

Conway, H., G. Catania, C. F. Raymond, A. M. Gades, T. A. Scambos, and H. Engelhardt (2002), Switch of flow direction in an Antarctic ice stream, *Nature*, *419*(6906), 465–467, doi:10.1038/nature01081.

Dahl-Jensen, D., N. Gundestrup, S. P. Gogineni, and H. Miller (2003), Basal melt at NorthGRIP modeled from borehole, ice-core and radio-echo sounder observations, *Ann. Glaciol.*, *37*(1), 207–212, doi:10.3189/172756403781815492.

Evans, S., and G. de Q. Robin (1966), Glacier depth-sounding from the air, *Nature*, *210*(5039), 883–885, doi:10.1038/210883a0.

Fahnestock, M., W. Abdalati, I. Joughin, J. Brozena, and P. Gogineni (2001), High geothermal heat flow, basal melt, and the origin of rapid ice flow in central Greenland, *Science*, *294*(5550), 2338–2342, doi:10.1126/science.1065370.

Hindmarsh, R. C. A., E. C. King, R. Mulvaney, H. F. J. Corr, G. Hiess, and F. Gillet-Chaulet (2011), Flow at ice-divide triple junctions: 2. Three-dimensional views of isochrone architecture from ice-penetrating radar surveys, *J. Geophys. Res. Earth Surf.*, *116*(F2), F02024, doi:10.1029/2009JF001622.

Jankowski, E. J., and D. J. Drewry (1981), The structure of West Antarctica from geophysical studies, *Nature*, *291*(5810), 17–21, doi:10.1038/291017a0.

Kingslake, J., R. C. A. Hindmarsh, G. Adalgeirsdottir, H. Conway, H. F. J. Corr, F. Gillet-Chaulet, C. Martin, E. C. King, R. Mulvaney, and H. D. Pritchard (2014), Full-depth englacial vertical ice sheet velocities measured using phase-sensitive radar, *J. Geophys. Res. Earth Surf.*, *119*(12),

2604–2618, doi:10.1002/2014JF003275.

MacGregor, J. A., M. A. Fahnestock, G. A. Catania, J. D. Paden, S. Prasad Gogineni, S. K. Young, S. C. Rybarski, A. N. Mabrey, B. M. Wagman, and M. Morlighem (2015), Radiostratigraphy and age structure of the Greenland Ice Sheet, *J. Geophys. Res. Earth Surf.*, *120*(2), 212–241, doi:10.1002/2014JF003215.

Morlighem, M., E. Rignot, J. Mouginot, H. Seroussi, and E. Larour (2014), Deeply incised submarine glacial valleys beneath the Greenland ice sheet, *Nat. Geosci.*, *7*(6), 418–422, doi:10.1038/ngeo2167.

Ng, F., and H. Conway (2004), Fast-flow signature in the stagnated Kamb Ice Stream, West Antarctica, *Geology*, *32*(6), 481–484, doi:10.1130/G20317.1.

Oswald, G. K. A., and G. D. Q. Robin (1973), Lakes beneath Antarctic Ice Sheet, *Nature*, *245*(5423), 251–254, doi:10.1038/245251a0.

Peters, M. E., D. D. Blankenship, and D. L. Morse (2005), Analysis techniques for coherent airborne radar sounding: Application to West Antarctic ice streams, *J. Geophys. Res. Solid Earth*, *110*(B6), B06303, doi:10.1029/2004JB003222.

Raymond, C. (1983), Deformation in the vicinity of ice divides, *J. Glaciol.*, *29*(103), 357–373. Robin, G. D. Q., S. Evans, and J. T. Bailey (1969), Interpretation of radio echo sounding in polar ice sheets, *Philos. Trans. R. Soc. Lond. Math. Phys. Eng. Sci.*, *265*(1166), 437–505, doi:10.1098/rsta.1969.0063.

Tikku, A. A., R. E. Bell, M. Studinger, and G. K. C. Clarke (2004), Ice flow field over Lake Vostok, East Antarctica inferred by structure tracking, *Earth Planet. Sci. Lett.*, *227*(3-4), 249–261, doi:10.1016/j.epsl.2004.09.021.

Vaughan, D. G., H. F. J. Corr, C. S. M. Doake, and E. D. Waddington (1999), Distortion of isochronous layers in ice revealed by ground-penetrating radar, *Nature*, *398*(6725), 323–326,

---

## Author Response (AR1)

Manuscript prepared for The Cryosphere
with version 2015/04/24 7.83 Copernicus papers of the LaTeX class copernicus.cls.
Date: 29 April 2016

**An ice sheet wide framework for englacial attenuation from ice penetrating radar data**

T. M. Jordan[1], J. L. Bamber[1], C. N. Williams[1], J. D. Paden[2], M. J. Siegert[3], P. Huybrechts[4], O. Gagliardini[5], and F. Gillet-Chaulet[6]

[1]Bristol Glaciology Centre, School of Geographical Sciences, University of Bristol, Bristol, UK.
[2]Center for Remote Sensing of Ice Sheets, University of Kansas, Lawrence, USA.
[3]Grantham Institute and Earth Science and Engineering, Imperial College, University of London, London, UK.
[4]Department of Geography, Vrije Universiteit Brussels, Brussels, Belgium.
[5]Le Laboratoire de Glaciologie et Géophysique de l'Environnement, University Grenoble Alpes, Grenoble, France.
[6]Le Laboratoire de Glaciologie et Géophysique de l'Environnement, Centre National de la Recherche Scientifique, Grenoble, France.

*Correspondence to:* T. M. Jordan (tom.jordan@bris.ac.uk) or J. L. Bamber (j.bamber@bris.ac.uk)

*The abstract has been revised to reflect that our algorithm is designed with attenuation solution accuracy in mind (rather than constraining reflection in absence of attenuation estimates)*

**Abstract.** Radar-inference of the bulk properties of glacier beds, most notably identifying basal melting, is, in general, derived from the basal reflection coefficient. On the scale of an ice sheet, unambiguous determination of basal reflection is primarily limited by uncertainty in the englacial attenuation of the radio wave, which is an Arrhenius function of temperature.  Existing bed-returned power algorithms for deriving attenuation assume that the attenuation rate is regionally constant which is not feasible at an ice sheet wide scale. Here we introduce a new semi-empirical framework for deriving englacial attenuation, and, to demonstrate its efficacy, we apply it to the Greenland Ice Sheet. A central feature is the use of a prior Arrhenius temperature model to estimate the spatial variation in englacial attenuation as a first guess input for the radar algorithm. We demonstrate regions of solution convergence for two input temperature fields, and for independently analysed field campaigns. The coverage achieved is a trade-off with uncertainty and we propose that the algorithm can be 'tuned' for discrimination of basal melt (attenuation loss uncertainty $\sim 5$ dB). This is supported by our physically realistic ($\sim 20$ dB) range for the basal reflection coefficient. Finally, we show that the attenuation solution can be used to predict the temperature bias of thermomechanical ice sheet models, and is in agreement with known model temperature biases at the Dye 3 ice core.

**1 Introduction**

*Following our author comments the following revisions have been made to the introduction: (i) we make it clear that Oswald and Goginenni (2008, 2012) concluded that relative reflection/reflection anomalies can be constrained in the interior of Greenland where attenuation rate variation is low, (ii) we make a greater emphasis regarding how our attenuation solution is useful for evaluation of englacial temperature, (iii) we note that there is potentially an additional uncertainty in the Arrhenius model due to the frequency dependence of attenuation.*

[revised manuscript text omitted]

*Following our author comments we have now substantially rewritten this section. Major changes include: (i) introducing the W97 model alongside the M07 model, (ii) outlining the empirical frequency correction in Macgregor et al. 2015b, (iii) referencing the results of Appendix A (where we conclude that the M07 model is a good estimate for attenuation at the radar frequency. (iv) Better introducing our notation for the modeled/estimated variables.*

[revised manuscript text omitted]

**2.5 Constraining the algorithm sample region**

*Following our author comments we have now substantially revised this section, moving the more*
240 *technical details to Appendix B.*

[revised manuscript text omitted]

*We have now revised this section. In particular: (i) we now combine analysis of reflection/attenuation/velocity in a single figure, (ii) we map the attenuation solution for all the converged regions*

[revised manuscript text omitted]

*Following our author comments we now fully rewritten this section. Key changes include: using the W97C alongside the M07 model to illustrate sensitivity/robustness, and extending the attenuation/temperature bias plots to include all of our converged region.*

[revised manuscript text omitted]

We have yet to consider an explicit classification of the subglacial materials and quantification of regions of basal melting. In future work, we aim to combine IPR data from preceding CReSIS field campaigns to produce a gridded data product for  basal reflection values and basal melt. It is anticipated that, as outlined by Oswald and Gogineni (2008, 2012); Schroeder et al. (2013), the specularity properties of the basal waveform, and how this relates to basal melt detection, could also be incorporated in this analysis. As the regions of algorithm coverage are sensitive to uncertainty, we suggest that these data products could have spatially varying uncertainty incorporated. Additionally, for the basal reflection and basal melt data sets, uncertainty in the measurements of $[P^C]$ will have to be incorporated in the uncertainty estimate for $[R]$. Establishing a procedure for the interpolation of these data sets where either: (i) the algorithm coverage is poor due to low attenuation solution accuracy, or (ii) the IPR data are sparse, will form part of this framework. Regions of lower solution accuracy, generally correspond to the interior of the ice sheet where spatial variation in the attenuation rate is much less pronounced (primarily the northern interior). Due to this lower spatial variability, (and despite the caveats in the paragraph above), these regions could potentially have their basal reflection values derived by using a forward Arrhenius temperature model for the attenuation.

Finally, we envisage that the framework introduced in this paper could be used for radar-inference of radar-attenuation, basal reflection and basal melt for the Antarctic Ice Sheet. Given that for high solution accuracy the radar algorithm requires high topographic roughness and relatively warm ice we suggest that IPR data in rougher regions toward the margins should be analysed first (refer to Siegert et al. (2005) for an overview of topographic roughness in East Antarctica). Additionally, the prediction of the model temperature field bias using the attenuation rate solution could be extended to the Antarctic Ice Sheet.

*Following our author comments we now have included: (i) A revised Appendix A that focuses on the choice of conductivity model, (ii) A new Appendix B (which incorporates the more technical details from Sect. 2.5*

[revised manuscript text omitted]

*We have implemented both reviewers suggestions for the figures, revising the captions where necessary. Major changes include: (i) Titles in words (rather than symbols), (ii) the Arrhenius model plot (plot 4 in original submission) now is split into 2 separate plots, (iii) we have merged the reflection map and attenuation rate map (10 and 11 in original submission). Our attenuation map is now over all of the converged region, and likewise for the reflection distribution, (iv) The temperature bias is also over the converged region and also considers the W97C model, (v) We have an extra plot for Appendix A, which reflection-thickness plots for the different conductivity models.*

[revised manuscript text omitted]

---

## Referee Report (RR1)

**Review:** "An ice-sheet wide framework for englacial attenuation and basal reflection from ice penetrating radar data" . Second Review.

T.M. Jordan, et al., The Cryosphere

Review by: Mike Wolovick

**Summary of Changes:**

The authors have changed the emphasis of the paper to focus on englacial attenuation in addition to basal reflectivity. They have improved the clarity of the figures and made other minor changes.

**Response to Major Comments:**

In my previous review, I had two major comments. The first was that the authors could extend the scope of their results to include basal reflectivity anomalies in the ice sheet interior. I suggested that, because the interior of the Greenland Ice Sheet has little variability in attenuation loss, the authors could constrain basal reflectivity anomalies even if they could not also constrain attenuation rate. Rather than expand the scope of their results, the authors refocused the emphasis of the paper onto constraining attenuation rate, and therefore diagnosing the biases of ice temperature models. This response is acceptable, if unsatisfying. There are few independent constraints on ice sheet thermal structure, other than sparsely distributed ice cores. The addition of full-depth temperature constraints from widely distributed radar data greatly expands the area over which we can constrain ice sheet thermal structure. As such, the advance presented by the authors is worth publishing on it's own, even if I would have liked to see them present reflectivity results in the ice sheet interior and compare those results with previous studies that did not use prior thermal models.

The second major comment I made was to express concern about the segmentation approximation the authors used to select a local sample area. I felt that the segmentation approximation was arbitrary and overly complex, and I suggested that the authors explore the sensitivity of their method to other means of selecting local sample areas, although I also stated that it was likely that any reasonable method of selecting a local sample area would produce similar results. In response, the authors included language stating that an irregular contiguous region would be preferable to a segmentation approximation, but declined to implement such a method because of

computational constraints.

 I am inclined to allow the authors' manuscript through on this round, but only because I attempted to write a script that computed irregular contiguous sample regions myself. I found that the algorithm itself is extremely simple; however, the difficulty arises because the resulting irregular regions are *too* irregular, to the point that they would probably produce unreliable results if implemented in the authors' method. The key bit of code that actually computes the irregular region (lines 76-100 in the attached script) is only 13 lines of Matlab code (excluding comments and without using ellipses to break up one long line of code into multiple lines). If you have access to the image processing toolbox, "imfill" does the same task in a single command. However, computation time is about 5 seconds per grid cell, which is unacceptable when there are ~$10^6$ active grid cells. In addition, the resulting sample regions often have highly unusual shapes and tend to be elongated parallel to the coast. If I did not impose a maximum area, some of the sample regions would form rings around the whole ice sheet margin, because attenuation rate tends to be higher around the edges of the ice sheet. Obviously, it is not glaciologically reasonable to include opposite sides of the ice sheet in the same sample region. I still do not believe the the segmentation approximation is the optimal solution to this problem, but given the difficulties I encountered when attempting to implement an alternative, I now think that the authors' method is better than I originally gave them credit for.

**Minor Comments:**

Line 18:

I still think that these two references should be replaced by older ones. I realize that the "e.g." is meant to imply additional uncited references, but radioglaciology did not start measuring ice thickness during the Obama administration. This place in the introduction is where you should give the audience a sense of the broader historical context of your work. Bailey et al., [1964] and Evans and Robin [1966] are more appropriate here. Besides, Fretwell et al., [2013] and Bamber et al., [2013] are referenced later in the paragraph.

Line 25:

The reference to Morlighem et al., [2014] is still incomplete.

Line 47:

dB should be dB/km

Lines 312-313:

I am glad that you took my suggestion to state the quality control criteria at the beginning of this section.  However, this sentence is still very unclear.  It can be clarified be using words in addition to symbols: say "(i) a strong correlation between bed-returned power and ice thickness ($d[P^C]/dh$) and (ii) a weak correlation of reflectivity and ice thickness ($d[R]/dh$) relative to the correlation between power and ice thickness ($d[P^C]/dh$)."  Using only symbols makes this sentence extremely opaque.

Line 395: "(defined here as...)"
Clarify the wording in the parentheses by saying "(convergence is defined here as...)".

Lines 454-458:
I am glad you have added geophysical interpretation to your results section.

Line 610:
I'm not sure I agree that the roughest topography in Antarctica is found around the margins.  The Siegert et al. paper was published before the Gamburtsev Mountains were surveyed in detail, for example.

Figures:
I appreciate the improved titles and labeling on all of the figures.  I would have liked it if Helheim was labeled in addition to Apuseeq, as a much higher percentage of the audience will have heard of Helheim.

Supplement Lines 35-36:
These lines still have "stationary" instead of "constant" (although I'm glad you made the change in the main text).

Supplement Line 76: "Greenlan"

---

## Editor Decision (ED1)

Dear Tom,

Thank you for your work to improve the original manuscript. Now I have received reviews from Joe MacGregor and Mike Wolovick who provided insightful reviews to an earlier version of your manuscript. They both suggested accepting your manuscript only with technical corrections. Below, please find my own comments to your revised manuscript. I request minor revision at this stage. I expect no major disputes on the issues brought by the reviewers and by myself, so I will not send the revised one to the reviewers, but review it by myself.

Thanks for choosing The Cryosphere to publish your valuable work.

Best regards,

Kenny Matsuoka

===

General issues.

G1: The manuscript is well written but I am afraid that many people may confuse differences in for example  and . Please add such symbol to all labels of the figures, when appropriate (e.g. Figure 4's ordinate, labels for Fig. 5a and 5b, label of Fig 6a, Fig. 9, Fig. 10).

G2: GISM and SICOPOLIS models do not predict ice thickness accurately. I assume that the authors ignore the difference between observed and predicted ice thicknesses, and used observed ice thickness and model-predicted vertical profile of the temperature for relative depth (i.e. fraction of the local ice thickness). Is this understanding correct? Anyway, please add a paragraph to explain this point, and if possible to present the difference of GISM-, SICOPOLIS-modeled ice thickness to the observed ice thickness.

Line 30-35: I also recommend including pioneer work of radar to measure ice thickness. Fretwell's and Bamber's work are cited lines below in the context of new bed DEMs of Greenland and Antarctica. I cannot clearly see the difference between work done for basal material properties and for basal melting or freezing. At lines 42-43, basal melting is considered as a part of basal material properties. I think that everyone has different opinions which work is most significant for these sub disciplines in radioglaciology, but I would suggest considering to cite following work as well.

For internal layer structure, I suggest Fujita et al. (1999), which is away more significant than my own work in 2010 that you cited.

Fujita, S., Maeno, H., Uratsuka, S., Furukawa, T., Mae, S., Fujii, Y., & Watanabe, O. (1999). Nature of radio echo layering in the Antarctic ice sheet detected by a two-frequency experiment. *Journal of Geophysical Research-Solid Earth, 104*(B6), 13013-13024.

Also, Bentley et al. (1998) and Peters et al. (2005) made milestones.

Bentley, C. R., Lord, N., & Liu, C. (1998). Radar reflections reveal a wet bed beneath stagnant Ice Stream C and a frozen bed beneath ridge BC, West Antarctica. *Journal of Glaciology, 44*(146), 149-156.
Peters, M. E., Blankenship, D. D., & Morse, D. L. (2005). Analysis techniques for coherent airborne radar sounding: Application to West Antarctic ice streams. *Journal of Geophysical Research-Solid Earth, 110*(B6), doi:10.1029/2004JB003222. doi:10.1029/2004jb003222|issn 0148-0227

Line 50: Peters 2005 should be Peters et al. (2005). Correct the reference list as well.

Line 88: remove the end parenthesis ")".

Line 113: missing figure number. It should be Figure 2.

Line 133: "surface roughness" → "bed roughness"

Line 193: "electrical conductivity" → "dielectric conductivity"

Line 199: Define M. In the current form, M = micro mol/L. It is probably better to define M = mol/L so that $CH+ = 0.8$ micro M.

L205: [L hat] is defined as the total loss, but it is two-way attenuation. Total loss sounds like that it includes surface transmission loss, volume scattering due to crevasses etc as well.

L212: "electrical" → "dielectric" (you said "dielectric properties" at line 209).

L217: "electrical" → "dielectric"

L218: "radar system frequency" → "radio-wave frequency (or radar frequency)"

Line 225: I assume that the authors calculated depth series of (in-situ) attenuation rates using depth series of ice temperatures predicted by the two models, integrated the in-situ attenuation rates over the full ice column and then divided it by the ice thickness. Please briefly explain this process around this line. I often see that people first calculate depth-averaged temperature to estimate the depth-averaged attenuation rate, which is wrong due to the Arrhenius relationship between them.

Line 228: I think all other depth-normalized values are in the unit of per kilometers, not per meters.

Line 233: I cannot see this point clearly in Figures 1a and 5c. Because data density is highly variable over the GrIS, majority of the data and majority of the data covered region are quite different.

Line 250: Matsuoka (2011, GRL) demonstrated that even if everything is equal but only ice thickness varies, the depth-averaged attenuation can vary. In other words, even if the sampling region is small enough to avoid any variable SMB, geothermal flux, or such, the empirical method to estimate the attenuation rate from the depth variations of the returned power is inherently not robust (see Figure 3b and Figure 4 in Matsuoka, 2011). I accept the approach the authors took but this point should be mentioned here to clarify the limitation of the proposed method. Depth-averaged attenuation derived in this way is hardly consistent with the attenuation rates estimated with temperature models (Fig. 3b in Matsuoka, 2011).

Line 281: "slowing varying" → "slowly varying"?

Line 285: "in the supplementary material (Figure S2)"

Line 295: Matsuoka et al. (2012b) analyzed depth dependence of the returned power but it is to demonstrate how the classical analysis is not robust. So, it is not appropriate to cite Matsuoka et al. (2012b) in this context.

Line 298-299: [S] is not defined. The current Equation (6) includes [S] so it does include the instrumental factors (I assume that [S] represents instrumental factors, such as transmission power).

Line 308: "and if d[S]/dh = 0" When d[R]/dh is large, it is usually caused by tilted bed.

Line 314: [S] is not well constrained in many cases, so usually only spatial variations of [R], not the absolute value of [R], is discussed.

Line 317: It is probably helpful to cite Matsuoka (2011).

Line 328: Figure 8 shows that corrections are typically more than zero for thinner ice, whereas the corrections are less than zero for thicker ice. However, in theory, thinner ice is colder (not warmer) and then the attenuation rate is predicted smaller than the thicker ice (Fig. 3b of Matsuoka, 2011). So, I don't know whether this depth dependent features are really from the ice temperature or from a combination of many factors. Can you demonstrate how this depth dependence is generally vaid over the GrIS?

Line354: Equation (6) is defined as [Pc] = [R] - [L] + [S], so the Equation (10) should be [R] = [L] + [Pc] - [S] = 2h + [P^c] - [S].

Line 357: Please define [R hat] clearly so that the difference between [R] and [R hat] will be clearer. My understanding is that [R] = 2h + [P^c] – [S], so only one difference between [R] and [R hat] is whether Arrhenius-model-based or Radar-inferred attenuation rates are used. Is this correct?

Line 423: what do you want to say with "radar-inferred attenuation rate/loss"? Is it rate or total loss? And loss could include for example volume scattering from crevasses. I think that it is better to say "attenuation rate" and "two-way attenuation".

Line 457: sigma is used to define the dielectric conductivity. I don't really see a need to define mean and standard deviation here using symbols.

Line 464: "wet" → "thawed"?

Line 485:  is already defined with Equation 8, so please use a different symbol, such as . ("mean" can be a bar over "B").

Line 502: "region region"

Line 503: "near-continuous"

Line 507: "the frequency distribution" → "the probably distribution (or probability function"? "Frequency" is confusing with this context.

Line 578: please rewrite "For our temperature fied-conditioned, bed –returned power, method this is not…"

Line 591: "The difference between Arrhenius-model and Radar-inferred attenuation rates averaged over the ice thickness"?

Line 594: "electrical" → "dielectric"

Line 596: I assume that Figure 13e shows modeled temperature at fractions of the modeled ice thickness in terms of observed ice thickness. E.g. modeled ice temperature 10% of the model-predicted ice thickness below the surface is shown here as the modeled temperature 10% of the observed ice thickness below the surface. Please see general comments G2.

Line 614: please add a reference for the acidity argument.

Line 640: "Attenuation rate/loss". See my comment above.

Line 695: "impurities" → "constituents"

Line 709: "electrical" → "dielectric"

Line 730: "non-specular, volume scattering"?

Table 1: Table 1 says that [R hat] is defined with Equation (12) but it is defined with Equation (10), not (12). [R] is said that it is defined with Equation (8), but I cannot see an equation that defines [R].

Figure 3: Change the ordinate so that the radar returned power is shown in the logarithm (dB) scale. All other figures show radar data in the decibel scale. Also consider using the depth instead of depth index for the abscissa.

Figure 4: "Arrhenius model **M07** in MacGregor et al. (2007)"

Figure 8: please add the lengths of the targeted window.

Figure 9: [L] is defined total loss in the main text so it is inconsistent (but I proposed to call it "two-way attenuation", not "loss").

Figure 12: (here and elsewhere) [L] should be called more consistently. "Attneuation" and "loss" are used in interchangeable manner. I recommend to call [L] as two-way attenuation.

Figure 14: Bold (a) and (d) in the caption.

Supplemental document (SD) Line 7: "Reproducibility"?

---

## Author Response (AR2)

We again thank both the reviewers and the editor for their constructive comments, and the time taken to review our manuscript. In this document we provide responses to the second reviews and the editor comments, with our responses in blue italic text, followed by a marked up version of the manuscript.

Review of "An ice sheet wide framework for englacial attenuation from ice penetrating radar data" by T.M. Jordan et al.

Joseph A. MacGregor 5 June 2016

This manuscript continues to impress. I have only a few minor editorial comments and suggested edits.

20: A paper that the authors don't cite but is somewhat related to their undertaking is Schroeder et al. [2016, Geophysics, 81(1), WA35-WA43]. The present authors' is arguably more sophisticated, but Schroeder et al. [2016] also undertook an analogous strategy of leveraging the reasonably predicted behavior of the attenuation rate to better constrain basal conditions

Schroeder et al. 2016 has now been added here.

25: Mor -> Morlighem et al. [2014], I assume

**Correct. Changed as suggested.**

42: expected to be low

Changed as suggested.

54: uniform: same in space; constant: same in time. Here "uniform" makes more sense than "constant"

We would like to retain use of the term constant (which was suggested by the other reviewer).

55: "constancy" adds nothing here

We disagree. As demonstrated in section 2.6 if `uncorrected' for, the constancy assumption leads to a pronounced systematic bias in the linear regression slope estimates.

59-60: "A central feature of our algorithm is to use an Arrhenius model to estimate the attenuation rate." Expand upon this statement, because in its present form it does not distinguish the present study clearly from many other studies.

We have now been more explicit and replaced the sentence with: A central feature of our approach is to firstly estimate the spatial variation in attenuation rate using an Arrhenius model, which enables us to modify the empirical bed-returned power method'

308: 20 measurements

**Done.**

527-9: Here and elsewhere in the manuscript, there is a mild confusion regarding the nature of the variable depth range sampled by MacGregor et al. [2015b]. While we reported depth-averaged temperatures, we made no attempt to restrict our analysis to the approximately isothermal portion of the ice sheet. We simply considered our inferred values to represent the mean temperature within the sampled depth range, even at borehole intersections where the temperature is known not to be isothermal within that range.

We appreciate this subtlety (hence our use of `approximately isothermal') and apologise for any apparent confusion. We do, however, think that it is important to note that (on average) greater variation in temperature occurs over the full ice column, than the region where internal layers are present/detectable. We have replaced line 527 with: 'We leave this problem, which is potentially more complex for the full ice column than the depth section where internal layers are present (which is closer to being isothermal), for future work'.

652-9: I think this is fair description of the problem and the route selected by the authors is a wise one (evaluation of bed-reflectivity/ice-thickness correlation for each model in Figure 14). While MacGregor et al. [2015b] expected the inferred frequency dependence to be somewhat lower than they reported, I am surprised at how well 1.7 does vs. 2.6 in Figure 14. I cannot reconcile why 2.6 works for the partial-thickness borehole-radar comparison in MacGregor et al. [2015b], but not for the entire ice column as considered here. To do so would require discounting the authors' otherwise reasonable assumption. Note that, from the perspective of the lab measurements of Stillman et al. [2013] and their interpretation by MacGregor et al. [2015b], the M07 model is "right for the wrong reasons". While I do not expect it as a revision, I would have preferred for W97C-1.7 to be used with the 3-D temperature models and subsequent analysis, rather than M07.

We thank the reviewer for their comments here. The study, MacGregor et al. [2015b], was a major inspiration for our work, and hopefully future work will reconcile our results.

681: Here I believe W97 is meant, not W97C

Done.

**Review:** "An ice-sheet wide framework for englacial attenuation and basal reflection from ice penetrating radar data" . Second Review. T.M. Jordan, et al., The Cryosphere Review by: Mike Wolovick

**Summary of Changes:**

The authors have changed the emphasis of the paper to focus on englacial attenuation in addition to basal reflectivity. They have improved the clarity of the figures and made other minor changes.

**Response to Major Comments:**

In my previous review, I had two major comments. The first was that the authors could extend the scope of their results to include basal reflectivity anomalies in the ice sheet interior. I suggested that, because the interior of the Greenland Ice Sheet has little variability in attenuation loss, the authors could constrain basal reflectivity anomalies even if they could not also constrain attenuation rate. Rather than expand the scope of their results, the authors refocused the emphasis of the paper onto constraining attenuation rate, and therefore diagnosing the biases of ice temperature models. This response is acceptable, if unsatisfying. There are few independent constraints on ice sheet thermal structure, other than sparsely distributed ice cores. The addition of full-depth temperature constrain ice sheet thermal structure. As such, the advance presented by the authors is worth publishing on it's own, even if I would have liked to see them present reflectivity results in the ice sheet interior and compare those results with previous studies that did not use prior thermal models.

The second major comment I made was to express concern about the segmentation approximation the authors used to select a local sample area. I felt that the segmentation approximation was arbitrary and overly complex, and I suggested that the authors explore the sensitivity of their method to other means of selecting local sample areas, although I also stated that it was likely that any reasonable method of selecting a local sample area would produce similar results. In response, the authors included language stating that an irregular contiguous region would be preferable to a segmentation approximation, but declined to implement such a method because of computational constraints.

I am inclined to allow the authors' manuscript through on this round, but only because I attempted to write a script that computed irregular contiguous sample regions myself. I found that the algorithm itself is extremely simple; however, the difficulty arises because the resulting irregular regions are *too* irregular, to the point that they would probably produce unreliable results if implemented in the authors' method. The key bit of code that actually computes the irregular region (lines 76-100 in the attached script) is only 13 lines of Matlab code (excluding comments and without using ellipses to break up one long line of code into multiple lines). If you have access to the image processing toolbox, "imfill" does the same task in a single command. However, computation time is about 5 seconds per grid cell, which is unacceptable when there are ~10^6 active grid cells. In addition, the resulting sample regions often have highly unusual shapes and tend to be elongated parallel to the coast. If I did not impose a maximum area, some of the sample regions would form rings

around the whole ice sheet margin, because attenuation rate tends to be higher around the edges of the ice sheet.

Obviously, it is not glaciologically reasonable to include opposite sides of the ice sheet in the same sample region. I still do not believe the segmentation approximation is the optimal solution to this problem, but given the difficulties I encountered when attempting to implement an alternative, I now think that the authors' method is better than I originally gave them credit for.

We thank the reviewer for their exemplary level of scrutiny and their openness in providing scripts. As stated in our previous comments, we fully believe that this problem (constraining an anisotropic sample region based upon a local tolerance), is much more difficult to implement in a robust manner than it at first appears. We would also like to add that our use of an `integrated tolerance measure' neatly circumnavigates the need for an additional maximal area constraint.

**Minor Comments:**

Line 18: I still think that these two references should be replaced by older ones. I realize that the "e.g." is meant to imply additional uncited references, but radioglaciology did not start measuring ice thickness during the Obama administration. This place in the introduction is where you should give the audience a sense of the broader historical context of your work. Bailey et al., [1964] and Evans and Robin [1966] are more appropriate here.

**Done.**

Besides, Fretwell et al., [2013] and Bamber et al., [2013] are referenced later in the paragraph.

Line 25: The reference to Morlighem et al., [2014] is still incomplete.

**Done.**

Line 47: dB should be dB/km

**Done.**

Lines 312-313:

I am glad that you took my suggestion to state the quality control criteria at the beginning of this section. However, this sentence is still very unclear. It can be clarified be using words in addition to symbols: say "(i) a strong correlation between bed-returned power and ice thickness (d[PC]/dh) and (ii) a weak correlation of reflectivity and ice thickness (d[R]/dh) relative to the correlation between power and ice thickness (d[PC]/dh)." Using only symbols makes this sentence extremely opaque.

**Done.**

Line 395: "(defined here as...)" Clarify the wording in the parentheses by saying "(convergence is defined here as...)".

**Done.**

Lines 454-458: I am glad you have added geophysical interpretation to your results section.

Line 610:

I'm not sure I agree that the roughest topography in Antarctica is found around the margins. The Siegert et al. paper was published before the Gamburtsev Mountains were surveyed in detail, for example.

We would like to leave this reference/sentence as it is. This is because we are making the point that our method requires `rough topography and warm ice' for high solution accuracy (i.e. we need a pronounced slope in the power-thickness plots).

Figures:

I appreciate the improved titles and labeling on all of the figures. I would have liked it if Helheim was labeled in addition to Apuseeq, as a much higher percentage of the audience will have heard of Helheim.

Supplement Lines 35-36:

These lines still have "stationary" instead of "constant" (although I'm glad you made the change in the main text).

Done.

Supplement Line 76: "Greenlan"

Done.

Editor comments

**General notes:**

1. We have gone through the manuscript and made sure that: (i) `loss' is always preceded by attenuation, (ii) All references to `total loss' are removed. I have also defined [L] explicitly in table 1 as `two-way attenuation loss'.

2. One of my co-authors (Phillipe Huybrechts) has recommend a few additional changes which I have also made/marked up:

(i) His author affiliation has now been changed to: Earth System Science and Departement Geografie, Vrije Universiteit Brussel, Brussels, Belgium

(ii) The GISM Temperature field is not strictly steady state. I have therefore change `steady-state temperature field' to `temperature field' throughout.

(iii) He recommended two extra references when introducing the GISM temperature field which I have added (Shapiro and Ritzwoller 2004, Goelzer et al. 2013)

G1: The manuscript is well written but I am afraid that many people may confuse differences in for example  and . Please add such symbol to all labels of the figures, when appropriate (e.g. Figure 4's ordinate, labels for Fig. 5a and 5b, label of Fig 6a, Fig. 9, Fig. 10).

The use of words (as opposed to symbols) as figure labels/axes was explicitly asked for by Mike Wolovick, and we made his changes as suggested (we had originally used symbolic notation in our first submission). As you can see, we have included all relevant symbols in the accompanying captions (including , ). We therefore believe that our current presentation strikes the best balance between `accessibility to a general glaciological audience' (stating the titles in words), and `precise clarity to a RES specialist' (providing an explicit statement of the variables/symbols in the caption), and would like to request to leave the figures/captions as they are.

G2: GISM and SICOPOLIS models do not predict ice thickness accurately. I assume that the authors ignore the difference between observed and predicted ice thicknesses, and used observed ice thickness and model-predicted vertical profile of the temperature for relative depth (i.e. fraction of the local ice thickness). Is this understanding correct? Anyway, please add a paragraph to explain this point, and if possible to present the difference of GISM-, SICOPOLIS-modeled ice thickness to the observed ice thickness.

Yes; this is the correct interpretation. We use ice thickness observations: either the Bamber et al. 2013 1 km thickness data product (or in the case of the ice core plots we use the core profile thickness.) In order to make this point clear we have added to Section 2.4:

`Both the GISM and SICOPOLIS models provide temperature profiles as a function of relative depth, and these were vertically scaled using the Greenland Bedmap 2013 ice thickness data product'

The temperature profile plot (Fig 13) also notes that the core thickness is used in this instance.

Unfortunately I do not readily have the GISM and SICOPOLIS ice thickness fields to hand, so this is not possible.

Line 30-35: I also recommend including pioneer work of radar to measure ice thickness. Fretwell's and Bamber's work are cited lines below in the context of new bed DEMs of Greenland and Antarctica.

**We have now followed Mike Wolovicks' suggestions for ice thickness references.**

I cannot clearly see the difference between work done for basal material properties and for basal melting or freezing. At lines 42-43, basal melting is considered as a part of basal material properties.

We agree. However, the basal melting/freezing category was added at the suggestion of Mike Wolovick, so we have left this unchanged.

I think that everyone has different opinions which work is most significant for these sub disciplines in radioglaciology, but I would suggest considering to cite following work as well.

For internal layer structure, I suggest Fujita et al. (1999), which is away more significant than my own work in 2010 that you cited.

Fujita, S., Maeno, H., Uratsuka, S., Furukawa, T., Mae, S., Fujii, Y., & Watanabe, O. (1999). Nature of radio echo layering in the Antarctic ice sheet detected by a two-frequency experiment. *Journal of Geophysical Research-Solid Earth, 104*(B6), 13013-13024.

Also, Bentley et al. (1998) and Peters et al. (2005) made milestones.

Bentley, C. R., Lord, N., & Liu, C. (1998). Radar reflections reveal a wet bed beneath stagnant Ice Stream C and a frozen bed beneath ridge BC, West Antarctica. *Journal of Glaciology, 44*(146), 149-156.

Peters, M. E., Blankenship, D. D., & Morse, D. L. (2005). Analysis techniques for coherent airborne radar sounding: Application to West Antarctic ice streams. *Journal of Geophysical Research-Solid Earth, 110*(B6), doi:10.1029/2004JB003222. doi:10.1029/2004jb003222|issn 0148-0227

All references have been added. Thanks for the recommendations.

Line 50: Peters 2005 should be Peters et al. (2005). Correct the reference list as well.

**Done.**

Line 88: remove the end parenthesis ")".

**Done**

Line 113: missing figure number. It should be Figure 2. *Done*

Line 133: "surface roughness"

"""

"bed roughness"

Apologies. I was using `surface' in the sense of an EM surface/interface, but I now realise bed is clearer to glaciologists

Apologies; this was a careless mistake (I appreciate that the dielectric conductivity is the correct term as it incorporates dispersion).

Line 199: Define M. In the current form, M = micro mol/L. It is probably better to define M = mol/L so that CH = 0.8 micro M.

**Changed as suggested.**

L205: [L hat] is defined as the total loss, but it is two-way attenuation. Total loss sounds like that it includes surface transmission loss, volume scattering due to crevasses etc as well.

`Total loss' has been changed to `two-way attenuation loss'

Done.

L217: "electrical"

"

"
dielectric"

**Done**

L218: "radar system frequency"

"
"
radio-wave frequency (or radar frequency)"

**Changed to radar frequency.**

Line 225: I assume that the authors calculated depth series of (in-situ) attenuation rates using depth series of ice temperatures predicted by the two models, integrated the in-situ attenuation rates over the full ice column and then divided it by the ice thickness. Please briefly explain this process around this line. I often see that people first calculate depth-averaged temperature to estimate the depth-averaged attenuation rate, which is wrong due to the Arrhenius relationship between them.

This is correct. We are fully explicit about these steps in Appendix A (including equations) which is referenced in this section.

Line 228: I think all other depth-normalized values are in the unit of per kilometers, not per meters.

**Changed to km.**

Line 233: I cannot see this point clearly in Figures 1a and 5c. Because data density is highly variable over the GrIS, majority of the data and majority of the data covered region are quite different.

**Changed 'For the majority of the IPR data coverage region' to 'Toward the ice sheet margins'**

Line 250: Matsuoka (2011, GRL) demonstrated that even if everything is equal but only ice thickness varies, the depth-averaged attenuation can vary. In other words, even if the sampling region is small enough to avoid any variable SMB, geothermal flux, or such, the empirical method to estimate the attenuation rate from the depth variations of the returned power is inherently not robust (see Figure 3b and Figure 4 in Matsuoka, 2011). I accept the approach the authors took but this point should be mentioned here to clarify the limitation of the proposed method. Depth-averaged attenuation derived in this way is hardly consistent with the attenuation rates estimated with temperature models (Fig. 3b in Matsuoka, 2011).

We agree that there are problems with the basic empirical/bed returned power method, which is precisely why we introduce the local attenuation correction in section 2.6. We have now referenced Matsuoka 2011 explicitly in Section 2.6, as motivating our approach (which acts to reduce systematic bias due to local variation in attenuation, when performing slope estimates).

Done

Line 285: "in the supplementary material (Figure S2)"

Done

Line 295: Matsuoka et al. (2012b) analyzed depth dependence of the returned power but it is to demonstrate how the classical analysis is not robust. So, it is not appropriate to cite Matsuoka et al. (2012b) in this context.

**Done -apologies.**

Line 298-299: [S] is not defined. The current Equation (6) includes [S] so it does include the instrumental factors (I assume that [S] represents instrumental factors, such as transmission power).

We initially included instrumental factors, [S] in the first submission, but this was removed at the request of one of the reviewers. I think the rationale here is that for the recent CReSIS data [S] can be well approximated as a constant for each field season (and hence, if attenuation can be well constrained so can relative reflection).

Line 308: "and if d[S]/dh = 0" When d[R]/dh is large, it is usually caused by tilted bed.

Line 314: [S] is not well constrained in many cases, so usually only spatial variations of [R], not the absolute value of [R], is discussed.

Line 317: It is probably helpful to cite Matsuoka (2011).

Done.

Line 328: Figure 8 shows that corrections are typically more than zero for thinner ice, whereas the corrections are less than zero for thicker ice. However, in theory, thinner ice is colder (not warmer) and then the attenuation rate is predicted smaller than the thicker ice (Fig. 3b of Matsuoka, 2011). So, I don't know whether this depth dependent features are really from the ice temperature or from a combination of many factors. Can you demonstrate how this depth dependence is generally vaid over the GrIS?

This point was also raised as being (initially) counter-intuitive by Mike Wolovick. Here is our response:

'As discussed in Section 3.5 of our paper, and Macgregor et al. (2015b), the depth-averaged attenuation rate and the depth-averaged temperature are proxy variables for each other, and it is in this sense we use the terms 'warm' and 'cold'. An estimate for the spatial variation in the depth-averaged attenuation rate over the Greenland ice sheet is shown in Fig. 4(b). It is clear that, as a first approximation, the depth-averaged attenuation rate is proportional to ice thickness (e.g. Bamber et al. (2013), Fig. 3), and it is lower in the interior of the ice sheet where the ice is thickest. This suggests that surface temperature (and its dependence upon surface elevation), is the dominant 'mechanism' that governs the spatial distribution of depth-averaged attenuation rate. This supports our general 'thick=cold, thin=warm' association. Finally, it is clear that this association holds over the spatial scale of our sample regions, (refer to Fig. 6 for the window vector plot).'

However, in view that the behaviour in Figure 8. may not hold everywhere. 'Typically' has been replaced with `In this case' in line 328, and '/warmer' and '/colder' has been deleted. Finally, our power correction does not assume that 'thick=cold, thin=warm' must hold (I introduced it as a mental model, which aided in the interpretation of Fig. 8) so I would argue that we do not need to demonstrate this.

Line354: Equation (6) is defined as [Pc] = [R] - [L] + [S], so the Equation (10) should be  $[R] = [L] + [Pc] - [S] = 2 < B > h + [P^c] - [S]$ .

**As stated before, [S] has been removed from the equations.**

Line 357: Please define [R hat] clearly so that the difference between [R] and [R hat] will be clearer. My understanding is that  $[R] = 2 < B > h + [P^c] - [S]$ , so only one difference between [R] and [R hat] is whether Arrhenius-model-based or Radar-inferred attenuation rates are used. Is this correct?

This is the correct interpretation. We define [R hat] in equation (10) a few lines before so this should all be explicit. (I think maybe (10) was misread here?)

Line 423: what do you want to say with "radar-inferred attenuation rate/loss"? Is it rate or total loss? And loss could include for example volume scattering from crevasses. I think that it is better to say "attenuation rate" and "two-way attenuation".

**Done**

Line 457: sigma is used to define the dielectric conductivity. I don't really see a need to define mean and standard deviation here using symbols.

We agree; it is not strictly necessary to have used symbols here. However, in this context, use of mu and sigma is standard practice. Sigma is also distinguishable from sigma\_infty (the HF dielectric conductivity), as it the conductivity has a subscript.)

Line 464: "wet" □□"thawed"?

**Done.**

Line 485:  is already defined with Equation 8, so please use a different symbol, such as . ("mean" can be a bar over "B").

We agree - this is much clearer.

Line 502: "region region" *Done.*

Line 503: "near-continuous"

Done.

Line 507: "the frequency distribution"  $\Box \Box$  "the probably distribution (or probability function"? "Frequency" is confusing with this context.

Done.

Line 578: please rewrite "For our temperature field-conditioned, bed –returned power, method this is not..."

Replaced with: `For our method, which uses ice sheet model temperature fields as an input'

Line 591: "The difference between Arrhenius-model and Radar-inferred attenuation rates averaged over the ice thickness"?

I think, given our explicit use of depth-averaged notation in the rest of the paragraph it should be clear that we mean the solution differences in the modelled and radar-inferred depth-averaged attenuation rates?

**Done**

Line 596: I assume that Figure 13e shows modeled temperature at fractions of the modeled ice thickness in terms of observed ice thickness. E.g. modeled ice temperature 10% of the model-predicted ice thickness below the surface is shown here as the modeled temperature 10% of the observed ice thickness below the surface. Please see general comments G2.

Correct. We have added 'The model temperature profiles are vertically rescaled using the core ice core thickness (2038 m)' to the figure caption

Line 614: please add a reference for the acidity argument.

Macgregor et al. 2015a (which demonstrates that the Holocene- LGM transition occurs at different depths in southern and northern Greenland) has been added here.

Line 640: "Attenuation rate/loss". See my comment above.

See response above

Done

Line 730: "non-specular, volume scattering"?

We have change this to 'non-specularity of internal reflections, volume scattering". Hopefully this is now clear.

Table 1: Table 1 says that [R hat] is defined with Equation (12) but it is defined with Equation (10), not (12). [R] is said that it is defined with Equation (8), but I cannot see an equation that defines [R].

This has now been corrected (this mistake happened because we moved some equations to the appendix between manuscript iterations). We have also added that 'total loss' is ' (two-way) attenuation loss').

Figure 3: Change the ordinate so that the radar returned power is shown in the logarithm (dB) scale. All other figures show radar data in the decibel scale. Also consider using the depth instead of depth index for the abscissa.

If possible we would prefer to use the linear scale. This is because: (i) we impose WF quality control in the linear scale, (ii) our linear plot can be well compared with the plots in Oswald and Gogineii (2008), upon which our method is based. The depth-index follows the notation of equation (2).

Figure 4: "Arrhenius model M07 in MacGregor et al. (2007)"

Done.

Figure 8: please add the lengths of the targeted window.

**Done – good suggestion.**

Figure 9: [L] is defined total loss in the main text so it is inconsistent (but I proposed to call it "two-way attenuation", not "loss").

**See general note.**

Figure 12: (here and elsewhere) [L] should be called more consistently. "Attneuation" and "loss" are used in interchangeable manner. I recommend to call [L] as two-way attenuation.

See general note

Figure 14: Bold (a) and (d) in the caption.

**Done**

Supplemental document (SD) Line 7: "Reproducibility"?

Done.

Manuscript prepared for The Cryosphere with version 2015/04/24 7.83 Copernicus papers of the LATEX class copernicus.cls. Date: 23 June 2016

**An ice sheet wide framework for englacial attenuation from ice penetrating radar data**

T. M. Jordan1, J. L. Bamber1, C. N. Williams1, J. D. Paden2, M. J. Siegert3, P. Huybrechts4, O. Gagliardini5, and F. Gillet-Chaulet6

[revised manuscript text omitted]